# KnowledgeSmith: Uncovering Knowledge Updating in LLMs with Model Editing and Unlearning

**Yinyi Luo**[1,2]* **Zhexian Zhou**[1], **Hao Chen**[1], **Kai Qiu**[1], **Marios Savvides**[1], **Sharon Li**[3], **Jindong Wang**[2†]
[1]Carnegie Mellon University    [2]William & Mary    [3]University of Wisconsin-Madison

## Abstract

Knowledge editing and machine unlearning are two popular approaches for large language models (LLMs) to stay up-to-date. However, the knowledge updating mechanism of LLMs remains largely unexplored due to insufficient, isolated, and small-scale evaluation. For instance, are LLMs similar to humans in modifying certain knowledge? What differs editing and unlearning as training data increases? This paper proposes KnowledgeSmith, a unified framework to systematically understand the updating mechanism of LLMs. We first cast editing and unlearning as instances of one constrained optimization problem. Then, we propose an automatic dataset generator that provides structured interventions across multiple graph levels and data scales, enabling controlled studies of how different modification strategies propagate through model knowledge. Extensive experiments demonstrate nuanced insights over knowledge propagation, plasticity scaling, consistency, and robustness. For instance, our results show that LLMs do not exhibit similar updating as humans for different levels of knowledge, and there exists consistency-capacity trade-off. We hope our findings can offer suggestions to the design of more reliable and scalable strategies. Code: https://github.com/AIFrontierLab/KnowledgeSmith.

## 1 Introduction

Human knowledge is not stored as isolated facts but as a vast, interconnected web (Liu et al., 2024). From early encyclopedias to modern knowledge graphs, we represent knowledge as structured relations (Yang et al., 2025): concepts (nodes) linked by semantic or causal connections (edges). This networked organization enables humans to reason flexibly (Mark et al., 2020), update beliefs (Paulheim, 2016) when new evidence arises, and propagate changes across related domains (Flouris et al., 2008). For instance, when scientists revised the classification of Pluto from a planet to a dwarf planet, the update did not merely alter one fact but cascaded through textbooks, curricula, and related scientific explanations.

Do Large language models (LLMs) exhibit similar properties? Zhang et al. (2024) showed that they store and retrieve information at scale, generating answers that span diverse domains; Yet, unlike human knowledge graphs, the internal structure of LLM knowledge remains opaque (Zhang et al., 2023). Fine-tuning can overwrite large swaths of parameters but is resource-intensive and imprecise (Balne et al., 2024; Gekhman et al., 2024), often introducing instability or hallucinations (Khan et al., 2025; Ovadia et al., 2024). Researchers have recently shifted attention toward knowledge editing (Wei et al., 2024; Markowitz et al., 2025; Wang et al., 2024) and unlearning (Yao et al., 2024; Pawelczyk et al., 2024; Hong et al., 2024), where editing offers targeted modifications and unlearning aims to broadly remove specific information. Both are valuable, yet they are typically studied in isolation and without grounding in structured knowledge representations.

How to understand the knowledge updating mechanism in LLMs? Recent efforts show that editing techniques can be adapted for forgetting by redirecting or suppressing knowledge representations (Li et al., 2025b; Jung et al., 2025), while unlearning methods sometimes resemble coarse-grained editing at the dataset level (Guo et al., 2019). Other works investigate continual or compositional

---

*Work done with Prof. Jindong Wang at William & Mary.
†Contact: yinyil@andrew.cmu.edu; Corresponding author: jdw@wm.edu.

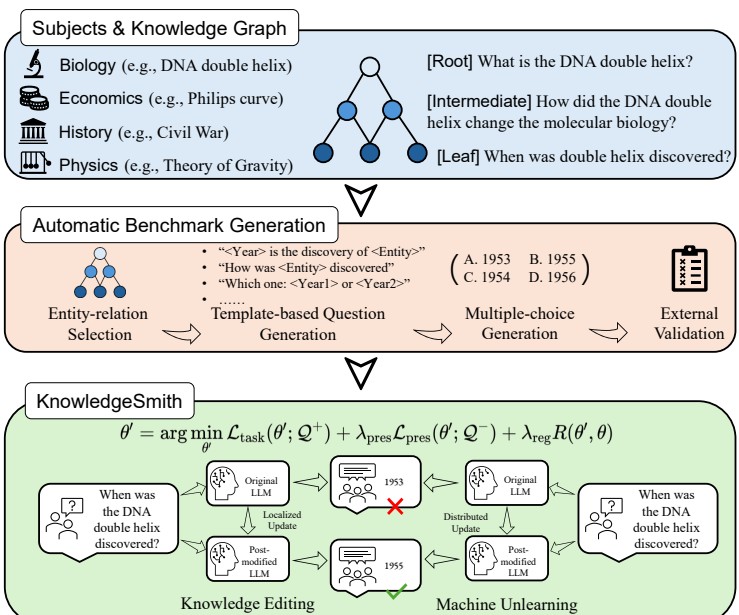

Figure 1: KnowledgeSmith pipeline. Starting from static KG, we generate dynamic probes at root, intermediate, and leaf levels, enabling evaluation of direct and propagated effects.

settings, where localized edits may interfere with broader forgetting objectives or vice versa (Gupta et al., 2024; Chen et al., 2024). A parallel strand examines the tension between specificity and generalization: editing often prioritizes precision but risks side effects, whereas unlearning emphasizes removal but may fail to incorporate new or corrected knowledge (Yao et al., 2023a).

Despite recent progress, there are still three critical challenges. First, most evaluations target isolated facts, neglecting the structured and interconnected nature of real-world knowledge (Thede et al., 2025). For example, if we update the fact that "Lyon is the capital of France" instead of Paris, a coherent system should also adjust related knowledge such as "the Eiffel Tower is located in France's capital," which otherwise becomes inconsistent. Second, the role of data scale in editing vs. unlearning remains unclear, with small data often sufficing for edits but not for forgetting(Zhong et al., 2023; Meng et al., 2022a). Third, there is no unified framework to jointly understand editing and unlearning, leaving their trade-offs in propagation, stability, and generalization unclear.

In this paper, we introduce **KnowledgeSmith** (Figure 1), a unified framework to understand the knowledge updating mechanisms in LLMs.[1] Theoretically, our framework casts editing and unlearning as complementary forms of constrained optimization. Empirically, building on the intuition that human knowledge is naturally structured as knowledge graphs (KGs), our framework can automatically transform any existing KG-related dataset into a benchmark for knowledge intervention evaluation, enabling systematic and scalable assessment without the need for hand-crafted test sets. For instance, more insights can be gained through interventions across hierarchical levels (root, intermediate, leaf) and data scales (from single instances to millions). Then, we conduct an extensive evaluation of editing and unlearning on different LLM families to explore knowledge propagation, scaling laws, representation shifts, and robustness under stress tests. Our key findings are:

1. **Propagation Asymmetry and Plasticity Limits:** Editing can over-spread(unintentionally altering related nodes), especially at higher nodes, while unlearning mostly under-spreads(forgetting failing to propagate beyond the target node). Hierarchical branch structure imposes intrinsic ceilings on update effectiveness, with higher or more central nodes limiting achievable knowledge modifications(§5.2.1,§5.2.2).

2. **Consistency–Capacity Tradeoff and Subject-Dependent Update:** Increasing data can trigger consistency collapse, where local updates contradict other knowledge; editing prioritizes local enforcement, unlearning preserves broader consistency. Some domains, like history, resist updates more than others, highlighting the need for subject-aware evaluation (§5.2.3,§5.2.4).

---

[1]Other approaches can also update knowledge in LLMs; we focus on editing and unlearning in this paper.

3. **Model Robustness:** Editing improves in-domain accuracy but harms OOD and adversarial stability, while unlearning preserves global robustness at the cost of weaker local gains(§5.3).

4. **Method-level Trade-offs:** Editing balances integration and preservation with strong low-data efficiency, unlearning is conservative but stable, while LoRA fine-tuning is unstable and prone to drift, making it unreliable for continual updates (§5.4).

5. **Unified Failure Modes and Stress Testing:** By observing model behavior on open-ended questions, we identify six main failure modes and find that unlearning preserves general task integrity better, whereas editing is more aggressive but effective in low-data regimes (§5.5).

**Contributions.** (1) We introduce KnowledgeSmith as a unified framework to understand knowledge updating in LLMs with editing and unlearning. (2) We present automatic data generation pipeline for LLM evaluation with scalable KG-structured interventions. (3) Our experiments demonstrate several insightful findings towards LLM knowledge updating that could inspire future research.

## 2 RELATED WORK

Other than fine-tuning which is expensive and requires large amount of training data, knowledge editing and machine unlearning are two popular and effective approaches to update LLMs' knowledge. Knowledge editing modifies LLMs' internal parameters to update its predictions on specific factual associations while ideally preserving unrelated knowledge (Yao et al., 2023b; Cao et al., 2021; Sinitsin et al., 2020). Existing approaches include gradient-based fine-tuning (Sinitsin et al., 2020; Zhu et al., 2020), localized weight modifications such as ROME (Meng et al., 2022a), MEMIT (Meng et al., 2022b), and SERAC (Mitchell et al., 2021), and memory-augmented methods that externalize edits (Mitchell et al., 2022). However, most prior evaluations are restricted to small benchmarks (Levy et al., 2017; Meng et al., 2022a) and do not examine how edits propagate through structured knowledge dependencies.

On the other hand, motivated by ethical, legal, or safety considerations, machine unlearning seeks to selectively erase information linked to a dataset, (Izzo et al., 2021; Thudi et al., 2022; Xu et al., 2025). Methods include retraining-based approaches (Ginart et al., 2019), negative-gradient fine-tuning (Thudi et al., 2022), regularization-based constraints (Golatkar et al., 2020), and approximate removal via influence functions or Fisher-weighted updates (Guo et al., 2019; Baumhauer et al., 2022). Yet, unlearning has largely been studied in isolation from editing, without systematic comparisons or evaluation in structured knowledge contexts.

In short, existing research highlights strong methodological advances but leaves two key gaps: (1) editing and unlearning are often treated as disjoint problems despite their conceptual overlap, and (2) evaluations rely on narrow datasets that fail to capture scaling behavior or structured propagation effects. Our work tries to establish a unified view of them and present an extensive analysis towards understanding LLM knowledge updating.

## 3 KNOWLEDGESMITH

In this section, we propose **KnowledgeSmith**, a unified framework to view editing and unlearning as complementary interventions.

### 3.1 PROBLEM DEFINITION

Let $f_\theta$ denote a language model parameterized by $\theta$, defining a conditional distribution $p_\theta(y \mid x)$ over output $y$ given input $x$. We study targeted interventions that modify or remove specific knowledge while preserving the model's general behavior.

An *update request* is given by an item $e$ (e.g., a factual triple, a prompt–response pair, or a small dataset), optionally accompanied by a scope $c$ that defines locality or related probes. For example, if $e$ is the fact "Paris is the capital of France", $c$ could include all prompts asking about European capitals such as "What is the capital of France?" or "Name the capital of European countries" while excluding unrelated prompts like "Who is the president of the United States?", ensuring that only related knowledge is affected while leaving unrelated knowledge untouched. Applying an update

operator $\mathcal{T}$ (e.g., editing or unlearning) yields updated parameters:

$$\theta' = \mathcal{T}(\theta; e, c), \qquad \Delta = \theta' - \theta, \tag{1}$$

where $\Delta$ is the parameter update.

The objective is therefore to update the targeted knowledge while preserving unrelated knowledge. To facilitate analysis, we define two probe sets: (1) *Positive probes* $\mathcal{Q}^+$ are inputs where the model's predictions should change; and (2) *Preservation probes* $\mathcal{Q}^-$ are inputs where predictions should remain unchanged. Formally, for an input $x$, denote $p_\theta(\cdot \mid x)$ and $p_{\theta'}(\cdot \mid x)$ as the output distribution of the model before and after KnowledgeSmith intervention, respectively, we have:

$$
\begin{aligned}
d\big(p_{\theta'}(\cdot \mid x), q_{\text{target}}(\cdot \mid x)\big) &\leq \eta^+, \quad \forall x \in \mathcal{Q}^+, \\
d\big(p_{\theta'}(\cdot \mid x), p_\theta(\cdot \mid x)\big) &\leq \varepsilon, \qquad \forall x \in \mathcal{Q}^-,
\end{aligned}
\tag{2}
$$

where $d(\cdot, \cdot)$ is a divergence or distance measure between distributions (e.g., KL divergence, cross-entropy, or $\ell_2$ distance over logits), $q_{\text{target}}(\cdot \mid x)$ is the desired post-intervention distribution on positive probes, the constant $\eta^+$ specifies a tolerance threshold for successful edits, reflecting that editing algorithms may only approximate the target distribution rather than match it exactly, and $\varepsilon$ is a stability threshold controlling how much drift is allowed on $\mathcal{Q}^-$.

## 3.2 A Unified Framework for Analyzing Editing and Unlearning

While Equation (2) formalizes the objectives using tolerance thresholds $\eta^+$ and $\varepsilon$, in practice we implement these constraints by relaxing them into loss terms over probes. Specifically, $\mathcal{L}_{\text{task}}(\theta'; \mathcal{Q}^+)$ penalizes deviations from the target distribution on $\mathcal{Q}^+$, $\mathcal{L}_{\text{pres}}(\theta'; \mathcal{Q}^-)$ penalizes drift on $\mathcal{Q}^-$, and $R(\theta', \theta)$ regularizes the overall update. Thus, both model editing and unlearning can be cast as a constrained optimization over model parameters:

$$\theta' = \arg\min_{\theta'} \ \mathcal{L}_{\text{task}}(\theta'; \mathcal{Q}^+) + \lambda_{\text{pres}} \mathcal{L}_{\text{pres}}(\theta'; \mathcal{Q}^-) + \lambda_{\text{reg}} R(\theta', \theta), \tag{3}$$

where $\mathcal{L}_{\text{task}}$ enforces the desired behavior on $\mathcal{Q}^+$, $\mathcal{L}_{\text{pres}}$ penalizes drift on $\mathcal{Q}^-$, and $R(\theta', \theta)$ regularizes the update (e.g., $\|\Delta\|_2^2$ (Ng, 2004), Fisher norm (Gu et al., 2012), or others (Hu et al., 2022)).

**Editing as targeted alignment.** Knowledge editing can be viewed as minimizing $\mathcal{L}_{\text{task}}$ toward a distribution $q_{\text{target}}$ that encodes corrected knowledge. For example, ROME (Meng et al., 2022a) and MEMIT (Meng et al., 2022b) locate and modify specific MLP weights to enforce new facts, while MEND (Mitchell et al., 2021) trains an auxiliary retriever–classifier to redirect predictions on edited queries. Other approaches apply gradient-based updates on $\mathcal{Q}^+$ while regularizing drift, such as GRACE (Hartvigsen et al., 2023). Even parameter-efficient methods like LoRA-based editing (Hu et al., 2022; Zheng et al., 2023) fit this form, with $R(\theta', \theta)$ enforcing low-rank adaptation.

**Unlearning as neutral alignment.** Unlearning corresponds to the same objective but with $q_{\text{target}}$ chosen as a *neutral distribution* $q_{\text{neutral}}$ that suppresses unwanted associations. This captures approaches that erase knowledge through gradient descent (Thudi et al., 2022), influence-function–based forgetting (Golatkar et al., 2020; Guo et al., 2019), or certified removal in convex models (Ginart et al., 2019). Recent work on unlearning in deep networks (Jagielski et al., 2022) also fits: their objectives penalize predictive alignment with sensitive data while constraining performance on $\mathcal{Q}^-$, exactly corresponding to the $\mathcal{L}_{\text{pres}}$ and $R(\theta', \theta)$ terms above.

**A unifying lens.** In this view, the distinction between editing and unlearning reduces to the choice of $q_{\text{target}}$: Editing: $q_{\text{target}}$ encodes a factual correction (e.g., "Paris is the capital of Germany"). Unlearning: $q_{\text{target}}$ is neutral, erasing prior associations (e.g., "Paris is the capital of [MASK]"). This framework subsumes methods across the spectrum: localized weight modifications (Meng et al., 2022b;a), memory-based editors (Mitchell et al., 2021), parameter-efficient adaptations (Hu et al., 2022; Zheng et al., 2023), influence-based forgetting (Golatkar et al., 2020), and certified removal (Ginart et al., 2019). Despite methodological differences, all can be interpreted as solving the same constrained optimization problem with different instantiations of $\mathcal{L}_{\text{task}}$, $\mathcal{L}_{\text{pres}}$, and $R(\theta', \theta)$.

Our formulation provides a principled and generalized lens for analyzing parameter modifications in LLMs, enabling fair comparison of editing and unlearning on their trade-offs in plasticity, stability, and generalization. However, to rigorously measure these effects in practice, we need benchmarks

that capture hierarchical dependencies, e.g., local versus global changes, and multilevel propagation of updates, which are largely missing from existing datasets. This motivates our automated benchmark construction in the following.

# 4 CONSTRUCTING EVALUATION BENCHMARK

Existing benchmarks (Meng et al., 2022a; Levy et al., 2017) for knowledge intervention evaluation suffer from two major limitations. First, they are largely *static*, testing only isolated facts without accounting for how updates might affect related knowledge. Second, they fail to capture *dependencies across facts*, which are crucial for understanding how changes propagate through the model and for revealing trade-offs between editing and unlearning.

We leverage *knowledge graphs (KGs)* to address these gaps, which dynamically encode hierarchical and relational dependencies among facts. Anchoring probes in a curated KG enables us to generate both local edits and their downstream consequences, transforming a single KG into a dynamic benchmark. Specifically, by targeting *root*, *intermediate*, and *leaf* nodes, our framework systematically tests how interventions propagate across multiple levels of dependency, thus providing a rigorous way to evaluate whether models can coherently update, forget, or preserve knowledge while maintaining global consistency. Concretely speaking, our data generation method can automatically transform any existing knowledge-related benchmarks such as MMLU (Hendrycks et al., 2021) into new ones, providing domain coverage and a standardized multiple-choice QA format for easy evaluation. Our pipeline consists of three stages (Figure 1), ensuring both quality and flexibility:

1. **Entity–Relation Selection:** We begin by prompting GPT-4o to generate a KG where entities and relations are organized hierarchically. The model is then asked to categorize nodes into three levels: *root* (broad, domain-level concepts), *intermediate* (mid-level categories or subtopics), and *leaf* (specific entities or instances). Sampling nodes from all three categories preserves the KG's hierarchical structure, ensuring evaluation goes beyond isolated facts to capture how edits or deletions propagate across different levels of related knowledge.

2. **Template-Based Question Generation:** Multiple question forms are generated for each triple, varying in directness and context. All templates are manually verified for grammaticality and factual alignment, preserving unambiguous mapping back to the KG. Six categories of probes are constructed (direct, reverse, conflict, multi-hop, comparison and contextual), each tied to a different aspect of model behavior under intervention.

3. **Multiple-Choice Construction:** Each probe is cast as a four-choice QA item, consistent with the MMLU-inspired format, ensuring that evaluation reflects true knowledge states rather than guesswork. Entity substitution and paraphrasing yield over one million samples across domains. All items are validated against the KG, with manual spot checks for quality assurance.

**Connection to KG-Based Evaluation.** Our generation pipeline is organized around two complementary families of probes: (1) *Positive probes* $\mathcal{Q}^+$, which directly test the edited or redirected knowledge, including its hierarchical propagation across root, intermediate, and leaf nodes. (2) *Preservation probes* $\mathcal{Q}^-$, which ensure that unrelated or out-of-scope knowledge remains intact, guarding against collateral damage.

To operationalize these two families, we instantiate six probe types. *Direct probes* ($\mathcal{Q}^+$) test whether the target fact itself is recalled or updated at different hierarchical levels. *Reverse probes* ($\mathcal{Q}^+$) examine whether knowledge updates preserve relation directionality. *Conflict probes* ($\mathcal{Q}^+/\mathcal{Q}^-$) expose residual beliefs and adversarial robustness by checking for contradictions after intervention. *Multi-hop probes* ($\mathcal{Q}^+$) evaluate whether interventions correctly propagate through chained relations in the KG. *Comparison probes* ($\mathcal{Q}^+$) assess whether the updated knowledge is consistently preferred when contrasted with alternatives or distractors. Finally, *Contextual probes* ($\mathcal{Q}^-$) test whether unrelated in-domain or OOD knowledge remains preserved in naturalistic settings. This design aligns directly with our experimental analyses: By explicitly embedding these probe types into the KG's hierarchical structure, the benchmark enables analyses that go beyond isolated fact checking, revealing whether interventions cascade consistently across levels of related knowledge.

**Generated Benchmark Dataset.** Our method allows flexible data generation across domains. In this paper, we instantiate the benchmark in four domains: economics, physics, history, and biology.

We restricted our evaluation to four domains to balance diversity and feasibility.[2] Each domain yields paired *pre-edit* and *post-edit* datasets that preserve entities but differ in factual content. Probes span root, intermediate, and leaf nodes, with conflict, propagation, comparative, and reverse variants, and include multiple paraphrased realizations. For each branch within every domain, we generate $10,000$ samples each for editing and unlearning, plus 100 evaluation probe sets, leading to $360,000$ training samples in total. This design creates a benchmark that is both large-scale and structurally sensitive, allowing systematic evaluation of edits and unlearning not just at the point of intervention but throughout the knowledge hierarchy. Dataset examples are in Appendix A.

## 5 EXPERIMENTS

### 5.1 SETUP

**Models.** Our evaluation covers 6 families of LLMs with 1B to 123B parameters, leading to a total of 13 models: LLaMA-3 (1B, 3B, 8B, 70B) (Meta, 2024), Qwen-3 (1.7B, 14B, 32B) (Team, 2025b), QwQ-32B (Team, 2025a), Mistral (24B, 123B) (Jiang et al., 2023), Gemma (2B, 7B) (Team, 2024), and DeepSeek-R1-0528-Qwen3-8B (DeepSeek-AI, 2025). This broad coverage enables us to study whether scaling behaviors and editing/unlearning performance generalize across architectures.

**Implementation Details.** We adopted AlphaEdit (Fang et al., 2025) and ReLearn (Xu et al., 2025).[3] AlphaEdit is a state-of-the-art editor that has been shown to outperform prior methods such as MEMIT(Meng et al., 2022b) and ROME(Meng et al., 2022a) in editing tasks, while ReLearn represents a leading approach to unlearning. Importantly, our framework is method-agnostic and directly extensible to other baselines, making it straightforward to integrate additional methods. Unlike traditional unlearning approaches where the retain set corresponds to the original knowledge, in our redirection-based setup the retain set is defined as the post-updated knowledge, ensuring that the model preserves the rewritten fact rather than reverting to its prior belief. This redirection-based formulation aligns better with real-world scenarios where knowledge is updated rather than erased. Editing and unlearning were applied separately to leaf, intermediate, and root nodes of the knowledge graph, with training data sizes ranging from 1 to $10,000$ samples. This setup allowed us to systematically analyze the effect of both hierarchy depth and data scale on the success of editing and unlearning. For evaluation, since each knowledge probing question is multiple-choice, we report accuracy as the proportion of questions for which the model selects the correct choice. This metric directly reflects the model's correctness in retrieving or updating the intended knowledge.

### 5.2 COMPARATIVE ANALYSIS OF EDITING AND UNLEARNING

#### 5.2.1 PROPAGATION ASYMMETRY: OVER- VS. UNDER-SPREADING

Human learners expect hierarchical consistency: updating a root concept should cascade to its descendants, while modifying a leaf should remain localized. We evaluate this in LLMs by applying editing or unlearning at three hierarchy levels (root, intermediate, leaf) and measuring performance on both targeted and structurally related nodes. We quantify these effects using *direct vs. multi-hop accu-*

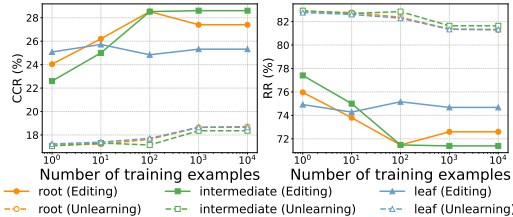

Figure 2: Propagation asymmetry metrics.

*racy* (Figure 2) as a proxy for propagation metrics: the *Collateral Change Ratio (CCR)* captures over-spreading for editing, and the *Residual Retention (RR)* captures under-spreading for unlearning (For the complete definitions of CCR and RR, see Appendix B).

Our results reveal a clear asymmetry: **editing tends to over-spread**, unintentionally altering related nodes, especially in lower hierarchy levels, whereas **unlearning often under-spreads**, failing to propagate forgetting beyond the target. These simple, interpretable metrics allow us to visualize propagation behavior across hierarchical branches.

---

[2]These subjects span both STEM and humanities, offering a representative testbed. Our pipeline is directly extensible to other domains such as law and medicine.

[3]We also conduct experiments on other methods with similar performance. Reported at Appendix E.

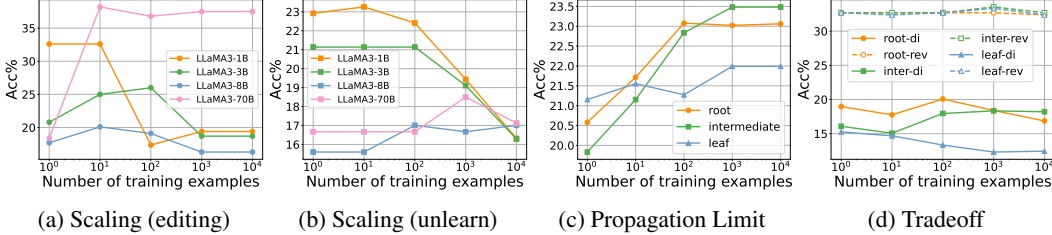

Figure 3: Plasticity scaling of the LLaMA3 family under (a) editing and (b) unlearning. (c) Propagation limits across three branches. (d) Consistency capacity tradeoff.

### 5.2.2 PLASTICITY SCALING AND BRANCH-DEPENDENT LIMITS

Plasticity captures how readily a model can update knowledge in response to limited training data, balancing the optimization of $\mathcal{L}_{task}$ on positive probes $\mathcal{Q}^+$ against preservation constraints $\mathcal{L}_{pres}$ on $\mathcal{Q}^-$. We extend this notion to *plasticity scaling*, examining systematically how model size, data scale, and hierarchical branch jointly influence the effectiveness of editing and unlearning.

Our main observations are as follows. First, as shown in Figures 3a and 3b, **smaller models exhibit higher immediate plasticity**, rapidly adapting to few-shot interventions and achieving strong in-domain performance on $\mathcal{Q}^+$, but their changes are often unstable, leading to degraded preservation on $\mathcal{Q}^-$. **Larger models require more data to register updates, reflecting lower short-term plasticity**, yet once modified they maintain stronger out-of-domain consistency, indicating more reliable preservation. Second, **branch-dependent upper bounds.** As shown in Figure 3c, different hierarchical branches exhibit distinct ceilings for achievable accuracy. Root-level edits/unlearning face a lower ceiling due to structural complexity and the need for coherent propagation across descendants. Intermediate-level branches achieve moderate ceilings. Leaf-level edits/unlearning can reach near-perfect in-domain accuracy with fewer examples, reflecting minimal propagation constraints. This reveals the effectiveness of updates is not uniform across the hierarchy: higher or more central nodes constrain achievable plasticity, while lower nodes allow maximal update with limited data.

### 5.2.3 CONSISTENCY–CAPACITY TRADE-OFF

Most prior work (Zhong et al., 2023; Park et al., 2025; Shi et al., 2024; Li et al., 2025a) primarily assess whether the target fact is updated successfully, without probing inverse relations. To our knowledge, no prior work explicitly quantifies this type of cross-relation or hierarchical consistency. In this work, we define consistency as the model's ability to maintain logical coherence across related knowledge after an intervention. Specifically, we test consistency by probing both the direct relation (e.g., "Paris is the capital of France") and the inverse or complementary relation (e.g., "France has capital Paris"), as well as across hierarchical or semantically related branches. A consistent update should correctly modify the target knowledge while preserving these related facts.

We uncover a new phenomenon: **consistency collapses once data scale surpasses the model capacity**. We term this the *consistency–capacity trade-off*, observed both in relation–inverse relation pairs (e.g., *capital-of* vs. *has-capital*) and across hierarchical branches. As shown in Figure 3d, direct probes initially respond to interventions but plateau or degrade as training scale grows, whereas reverse probes remain stably high, indicating preservation of contradictory knowledge. The divergence defines a *consistency collapse point*, occuring earlier in lower branches (intermediate, leaf) than root. Editing typically achieves stronger local updates but triggers earlier global inconsistency; unlearning preserves broader consistency but rarely removes the targeted knowledge completely.

Table 1: Similarity scores for each model are independently normalized via a log–min–max transformation: a small positive offset $\epsilon$ is added, $\log_{10}$ is applied, and the resulting values are linearly scaled to the $[0, 1]$ range.

| Metric | Setting | 1 | 10 | 100 | 1000 | 10000 |
|--------|---------|-------|-------|-------|-------|-------|
| KL | Unlearn | 0.014 | 0.392 | 0.805 | 0.838 | 0.883 |
| | Edit | 0.140 | 0.522 | 0.606 | 0.647 | 0.652 |
| L2 | Unlearn | 0.013 | 0.286 | 0.647 | 0.758 | 0.948 |
| | Edit | 0.054 | 0.368 | 0.507 | 0.628 | 0.633 |
| Fisher | Unlearn | 0.014 | 0.352 | 0.781 | 0.847 | 0.919 |
| | Edit | 0.101 | 0.438 | 0.552 | 0.641 | 0.647 |
| CKA | Unlearn | 0.917 | 0.861 | 0.566 | 0.576 | 0.692 |
| | Edit | 0.958 | 0.852 | 0.801 | 0.714 | 0.714 |

**Representation and Efficiency.** Table 1 shows the analysis of internal representations via Centered Kernel Alignment (CKA) (Kornblith et al., 2019), KL divergence, L2 distance and Fisher score (Zhang et al., 2022). The results show that unlearning exhibits abrupt phase transitions beyond a critical data scale, while editing induces smoother, localized adjustments (details in Ap-

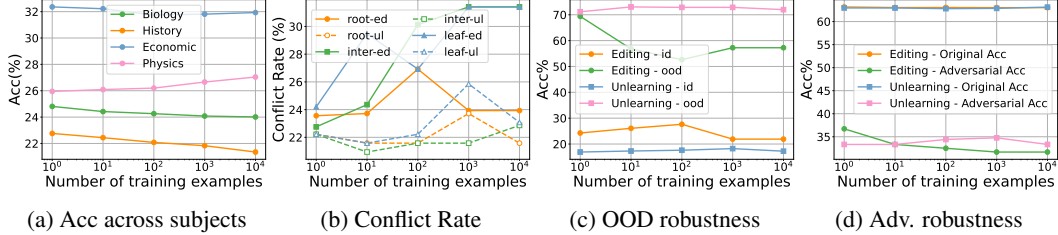

| (a) Acc across subjects | (b) Conflict Rate | (c) OOD robustness | (d) Adv. robustness |

Figure 4: Robustness evaluation under multiple stress tests. (a) Out-of-distribution (OOD) vs. in-domain accuracy. (b) Adversarial robustness relative to original accuracy. (c) Instruction-following accuracy in free generation, judged by an LLM. (d) Hallucination tendency across interventions.

pendix H). Computationally, unlearning is faster (e.g., ∼0.2h vs ∼6h for editing on 1,000 samples on an NVIDIA H100), reflecting its focus on stability over precise enforcement.

Consistency collapse is not only evident in output accuracy but also mirrored in representation dynamics and computational cost: editing maximizes factual enforcement at the expense of broader consistency and resources, whereas unlearning prioritizes stability and efficiency.

### 5.2.4 SUBJECT-DEPENDENT KNOWLEDGE UPDATE

At the subject level, Figure 4a reveal that **knowledge updating is strongly subject-dependent**. Among the four subjects (biology, economics, history, and physics), **history consistently exhibits the lowest update accuracy**, sometimes remaining nearly unchanged even with large numbers of training examples. Other subjects update, in contrast, propagate more efficiently. This highlights a critical insight: evaluation benchmarks must account for **subject-specific difficulty**. Standard datasets (e.g., CounterFact (Meng et al., 2022a), ZsRE (Levy et al., 2017)) treat all domains equivalently, but our results indicate that certain knowledge domains, such as history, are significantly more resistant to modification. Consequently, subject-aware evaluation is essential for accurately assessing editing and unlearning performance in LLMs.

### 5.2.5 CONTRADICTIONS AND CONFLICT RATE

While residual belief (Elidan et al., 2012) is commonly used to evaluate whether interventions succeed in suppressing prior knowledge, it does not capture a critical failure mode: the emergence of *contradictions*. We therefore introduce a complementary metric, *conflict rate*, which measures the proportion of queries where the model simultaneously supports mutually inconsistent statements after intervention. For instance, a model may assert both "Paris is the capital of Germany" and "Paris is the capital of France" under different contexts. Figure 4b shows this metric exposes patterns that residual belief alone cannot: **editing often leads to higher conflict in related branches (over-spreading), whereas unlearning tends to leave contradictions unresolved in upstream nodes (under-spreading).** By explicitly quantifying such inconsistencies, conflict rate provides a fuller view of hidden instabilities and unintended side effects.

### 5.3 ANALYSIS ON ROBUSTNESS

OOD robustness is tested using MMLU (Hendrycks et al., 2021). In the unified framework, in-domain probes $\mathcal{Q}^+$ consist of questions from the same subject (e.g., updating facts about geography using geography questions), reflecting alignment with $q_{\text{target}}$. In contrast, out-of-domain (OOD) probes $\mathcal{Q}^-$ are drawn from unrelated subjects (e.g., updating geography facts but measuring performance on economics, history, or law), testing the model's ability to preserve unrelated knowledge after the intervention. As shown in Figure 4c, **these objectives often conflict**. Unlearning preserves strong OOD accuracy (63–82%) but yields modest in-domain gains (≤30%), while editing substantially boosts in-domain accuracy (up to 50–60% in economics) at the cost of OOD stability, especially in mid-sized models. Larger models reduce but do not eliminate this trade-off. Increasing training examples improves in-domain performance until gains plateau, and disciplines vary, with economics generalizing better and history proving more resistant. This trade-off reflects the balance between $\mathcal{L}_{\text{task}}$ and $\mathcal{L}_{\text{pres}}$: stronger enforcement on $\mathcal{Q}^+$ tends to destabilize preservation on $\mathcal{Q}^-$, highlighting the challenge of achieving both local fidelity and global robustness together.

We then measure adversarial robustness by exposing the model to misleading or deceptive inputs, such as probes combining unrelated concepts (Figure 4d). This assesses whether the optimization constraints maintain stability on preservation probes $\mathcal{Q}^-$ under stress (details in Appendix D.1).

## 5.4 ANALYSIS ON FINE-TUNING

We further compare editing and unlearning with LoRA fine-tuning on Llama3-8B-Instruct to isolate method-level tradeoffs. Figure 6a shows LoRA yields unstable ID accuracy, sometimes dropping to $12.5\%$ at $k = 1000$. Scarce data lead to poor enforcement of target updates ($\mathcal{Q}^+$) while undermining preservation ($\mathcal{Q}^-$). Figure 6b shows OOD accuracy declining from $63.0\%$ ($k = 1$) to $61.6\%$ ($k = 1000$), indicating drift risks. Un-

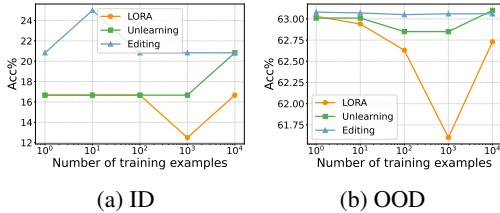

(a) ID        (b) OOD

Figure 6: LoRA, Editing and Unlearning.

learning remains stable around $63\%$, preserving prior knowledge but limiting target success. Editing combines stability with low-data efficiency, boosting ID accuracy to $25\%$ at $k = 10$ compared to $16.7\%$ for LoRA and unlearning. In summary, editing balances new knowledge integration and preservation, LoRA risks drift, and unlearning is conservative but stable, explaining why we prefer editing/unlearning for continual updates.

## 5.5 FAILURE MODE AND STRESS TESTING

Existing studies describe errors such as incomplete forgetting or knowledge pollution in a fragmented way, without systematically characterizing the underlying mechanisms. Through our experiments on **open-ended question answering**, we observed that models fail for different reasons under editing and unlearning interventions. To capture these patterns, we propose a **Unified Failure Mode Taxonomy** that organizes observed errors into six categories (examples of each type in Appendix D.2):

Table 2: Percentage (%) of observed failures in editing and unlearning.

| Failure Mode | Editing | Unlearning |
|---|---|---|
| Under-forgetting (RR) | 20 | 35 |
| Over-spreading (CCR) | 35 | 15 |
| Conflict emergence | 30 | 12 |
| Knowledge drift | 18 | 10 |
| Instruction-following drop | 22 | 18 |
| Hallucination increase | 5 | 4 |

under-forgetting (RR), over-spreading (CCR), conflict emergence (contradictions between updated and related knowledge), knowledge drift (performance degradation on unrelated tasks), instruction-following drop (reduced ability to follow complex instructions), and hallucination increase.

Stress-testing evaluates the failure modes with open generation tasks, making the model show practical robustness and use gpt-4o to evaluate. Our results show that hallucination (evaluated on TruthfulQA (Lin et al., 2022)) remains stable, instruction-following (open generation) drops moderately, and CoT reasoning can improve edit generalization but may increase residual knowledge, complicating unlearning (details in Appendix C).

Sequential update experiments, reported in Appendix F, further illustrate how multiple consecutive edits affect these behaviors and highlight potential cumulative effects on residual knowledge.

## 5.6 THEORETICAL ANALYSIS

Our theoretical perspective connects the observed behaviors to their geometric effects on model representations. Let $W \in \mathbb{R}^{m \times n}$ denote a parameter matrix (e.g., attention or MLP projection), with singular value decomposition $W = U\Sigma V^\top$. An intervention updates $W$ to $W' = U'\Sigma'V'^\top$. The difference between $W$ and $W'$ can be decomposed into two interpretable components:

- **Scaling effects.** Changes in singular values $\Sigma'/\Sigma$ indicate amplification or attenuation of certain representational directions.
- **Rotational effects.** Differences in subspaces $\text{span}(U, V)$ vs. $\text{span}(U', V')$ reflect reorientation of features while preserving their magnitude.

**Editing as local rotation with mild rescaling.** As shown in Figure 7a, editing primarily induces moderate rescaling of singular values while maintaining high orthogonal similarity between $(U, V)$ and $(U', V')$ across layers. This implies that editing preserves most of the representational geometry, redirecting specific factual directions through controlled rotations. Consequently, editing behaves

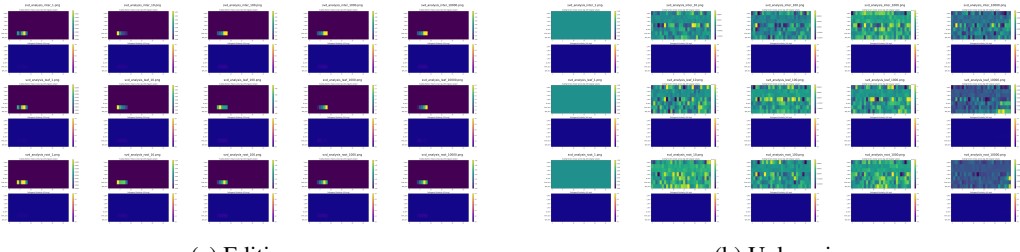

|                          |                          |
| :----------------------: | :----------------------: |
| (a) Editing              | (b) Unlearning           |

Figure 7: SVD-based geometric analysis of interventions. (a) Editing adjusts knowledge by gently rotating and slightly rescaling the representation space, preserving overall geometry while redirecting specific directions. (b) Unlearning, in contrast, acts by shrinking certain dimensions more aggressively, reducing the model's capacity in those directions rather than rotating them.

like a *rotation-plus-scaling operator*: it reallocates emphasis toward new factual associations while retaining global coherence. This explains why editing achieves strong local enforcement but often *over-spreads* changes to nearby branches (high CCR in Section 5.2).

**Unlearning as anisotropic scaling.** By contrast, Figure 7b shows that unlearning produces sharper downscaling of singular values with less stable alignment of $U, V$ across layers. This indicates suppression of capacity in certain subspaces rather than a simple rotation. Thus, unlearning resembles an *attenuation operator*: it removes the ability to encode certain directions but does not reliably rotate them into new ones. This mechanism aligns with the observed *under-spreading* behavior (high RR in Section 5.2), where forgetting is localized and fails to propagate fully across related nodes.

**Hierarchy-dependent dynamics.** Leaf-level interventions concentrate changes in later layers, supporting near-perfect local adaptation. Root-level interventions require distributed rotations and scalings across the network, introducing stricter ceilings on achievable accuracy. Intermediate nodes combine aspects of both. These theoretical patterns mirror our empirical findings on branch-dependent plasticity limits (Section 5.2.2).

## 5.7 DISCUSSION

Our findings offer several potential directions for future research. *(1) Model updating:* Updates should employ dynamic, hierarchical control such as level- and relation-aware algorithms. Branch-specific strategies can also improve effectiveness: for leaf nodes, updates can use more data for higher accuracy, while root nodes may require less data. Data size should be carefully calibrated for global consistency. Moreover, models exhibit subject-dependent sensitivity, hence, update methods should account for differences across domains. *(2) Evaluation metrics:* The conflict rate offers a more nuanced assessment of models, capturing hidden inconsistencies and ensuring that updates improve the model more holistically rather than just for specific tasks. This mirrors human reasoning in the sense that humans also monitor for contradictions and coherence, but the analogy is descriptive rather than mechanistic. *(3) Foundation models:* Future models could be designed with layer-wise or tensor-wise modularity, enabling finer-grained control when applying updates. By building update-friendly architectures, such models would allow interventions to target specific branches or layers more effectively, improving both efficiency and consistency of knowledge updates.

Our work has several limitations. First, our experiments are based on four domains due to limited compute budget and could be expanded to more domains and multimodal models. Second, our unified framework does not give theoretical bound for propagation and consistency remains open. Third, the analysis is based on recent editing and unlearning approaches, which could be extended to other algorithms to gain more insights.

## 6 CONCLUSION

We introduced KnowledgeSmith to understand the knowledge updating mechanism in LLMs by unifying editing and unlearning. Our experiments highlight fundamental trade-offs, e.g., unlearning prioritizes stability and efficiency but yields modest enforcement, while editing enforces knowledge updates more effectively at the risk of destabilization and higher computational cost. We hope our benchmark and analysis can shed light on future research on LLM knowledge updating.

ACKNOWLEDGMENT

This paper is partially supported by The Commonwealth Cyber Initiative (CCI) program (H-2Q25-020), William & Mary Faculty Research Award, and Modal Academic Compute Award. The authors acknowledge William & Mary Research Computing for providing computational resources and/or technical support that have contributed to the results reported within this paper. URL: https://www.wm.edu/it/rc.

ETHICAL AND REPRODUCIBILITY STATEMENT

ETHICS STATEMENT

This work investigates knowledge editing and unlearning in large language models with the goal of improving our understanding of how models update and forget factual information. Our experiments are restricted to controlled benchmarks, including publicly available datasets and synthetic data that we release. We do not use sensitive, private, or personally identifiable information. While the methods studied could, in principle, be misused to manipulate model knowledge for harmful purposes, our intention is purely scientific, and we have limited our scope to safe, non-sensitive settings. All pretrained models used in this study are publicly available and used in accordance with their licenses. We believe our work contributes to safer, more transparent, and more responsible approaches to model editing and unlearning.

REPRODUCIBILITY STATEMENT

We have made every effort to ensure the reproducibility of our results. All datasets used are publicly available or synthetically generated; details of dataset construction, splits, and preprocessing are provided in Appendix A. Model architectures, and evaluation metrics are fully described. Our implementation builds on open-source frameworks (e.g., PyTorch, HuggingFace Transformers, vLLM), and we will release the configuration files and synthetic benchmark data upon publication.

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

**Appendix**
**KnowledgeSmith: Uncovering Knowledge Updating in LLMs with Model Editing and Unlearning**

CONTENTS

## A  DATA GENERATION EXAMPLE AND PIPELINE

To make our pipeline transparent, we provide an end-to-end example showing how a single knowledge point expands into a large set of evaluation items, emphasizing hierarchical structure and controlled fact editing.

### A.1  KNOWLEDGE POINT AND KNOWLEDGE GRAPH (KG)

We illustrate how a knowledge point can be represented as a triple and anchored at different levels of the knowledge graph. Table 3 shows one example from each domain.

Table 3: Examples of knowledge triples and anchoring across different levels of the KG hierarchy.

| Domain | Example Triple | KG (Root → Intermediate → Leaf) |
|---|---|---|
| Biology | (DNA double helix, discovered_in, 1953) | Root: concept of DNA structure → role in molecular biology and genetics → link to genetics/medicine/biotech applications |
| Economics | (Phillips curve, describes, inflation–unemployment relationship) | Root: economic trade-offs → macroeconomic models of inflation and unemployment → policy debates on stagflation and monetary policy |
| History | (Declaration of Independence, signed_in, 1776) | Root: revolutions and independence movements → American Revolutionary era → specific events such as the Continental Congress or early U.S. governance |
| Physics | (Theory of General Relativity, published_in, 1915) | Root: fundamental physics theories → spacetime and gravitation framework → applications such as black holes, gravitational waves, or GPS corrections |

This fact is anchored at three levels of the knowledge graph:

- **Root:** broad, domain-level understanding.
- **Intermediate:** contextual understanding, including its role and implications.
- **Leaf:** fine-grained, specific questions.

### A.2  TEMPLATE GENERATION

For the selected fact, we generate multiple question templates per KG level, capturing different aspects of the fact (definition, role, context, and application).

- **Root-level templates:** Broad factual or conceptual questions.
- **Intermediate-level templates:** Questions about domain implications, causal relationships, and contextual applications.
- **Leaf-level templates:** Specific, field-dependent scenarios where the fact influences outcomes or knowledge in that domain.

An example of generated templates is shown in Table 4, where leaf-level templates are instantiated with different fields (e.g., genetics, medicine).

### A.3  PROMPTING GPT FOR QUESTION GENERATION

Our pipeline for generating evaluation questions follows these steps:

1. **Knowledge Graph Generation:** GPT is prompted to generate a structured KG for the target domain. Nodes represent root, intermediate, and leaf-level knowledge.
2. **Fact Selection:** From the KG, a single fact is selected (e.g., `(DNA double helix, discovered_in, 1953)`) to anchor all subsequent questions.
3. **Template Generation:** GPT is prompted to produce multiple templated question forms surrounding the fact. Templates vary in phrasing, style, and emphasis, covering definition, context, role, and applications.
4. **Level-Specific Question Generation:** Each template is input to GPT with instructions specifying the desired KG level (root, intermediate, leaf). Example prompts:

Table 4: QA templates for four knowledge points across Biology, Economics, History, and Physics.

| Level | Biology: *DNA double helix* | Economics: *Phillips curve* | History: *Declaration of Independence (1776)* | Physics: *General Relativity (1915)* |
|---|---|---|---|---|
| Root-level | What is the DNA double helix? Who discovered the DNA double helix? When was the DNA double helix discovered? What does the DNA double helix describe? Why is the DNA double helix important in biology? What shape is the DNA double helix? What was learned from the DNA double helix? Which scientists worked on the DNA double helix? | What is the Phillips curve? What relationship does the Phillips curve describe? Who proposed the Phillips curve? When was the Phillips curve introduced? Why is the Phillips curve important in economics? How is the Phillips curve used in macroeconomics? What does the Phillips curve imply about inflation and unemployment? Which countries have applied the Phillips curve concept? | What is the Declaration of Independence? When was the Declaration of Independence signed? Who signed the Declaration of Independence? Why was the Declaration of Independence created? What does the Declaration of Independence proclaim? Which country declared independence in 1776? What historical context led to the Declaration of Independence? Why is the Declaration of Independence important in history? | What is the Theory of General Relativity? Who proposed the Theory of General Relativity? When was the Theory of General Relativity published? Why is the Theory of General Relativity important? What does the Theory of General Relativity describe? How does General Relativity differ from Newtonian physics? What are the key concepts in General Relativity? Which experiments confirmed General Relativity? |
| Intermediate | How did the DNA double helix change molecular biology? What discoveries followed the DNA double helix? What role did the DNA double helix play in genetics? How did the DNA double helix influence medical research? What techniques confirmed the DNA double helix? How is the DNA double helix taught in schools? What reaction did scientists have to the DNA double helix? How did the DNA double helix affect other fields of science? | How does the Phillips curve affect monetary policy? What criticisms exist for the Phillips curve? How did the Phillips curve shape economic thought? How does the Phillips curve relate to inflation targeting? What data supports or contradicts the Phillips curve? How do economists interpret the Phillips curve over time? How does the Phillips curve influence labor market policies? How is the Phillips curve taught in universities? | How did the Declaration of Independence influence the American Revolution? What ideas from the Enlightenment are in the Declaration? How did other countries react to the Declaration? What role did the Declaration play in forming the U.S. government? How was the Declaration received by the British crown? What debates occurred during the drafting of the Declaration? How did the Declaration impact colonial society? How is the Declaration taught in schools? | How did General Relativity influence modern physics? What role does General Relativity play in cosmology? How does General Relativity explain gravity? How was General Relativity received by the scientific community? How does General Relativity relate to black holes? How is General Relativity taught in universities? What mathematical tools are used in General Relativity? How does General Relativity affect GPS technology? |
| Leaf-level | How did the DNA double helix influence research in genetics? What impact did the DNA double helix have in medicine? How was forensic science affected by the DNA double helix? In evolutionary biology, what role did the DNA double helix play? Why did biotechnology change after the DNA double helix? What does public health owe to the DNA double helix? How did the DNA double helix influence research in anthropology? What impact did the DNA double helix have in bioinformatics? How was drug development affected by the DNA double helix? In agriculture, what role did the DNA double helix play? | How does the Phillips curve explain stagflation in the 1970s? How did the Phillips curve influence central bank decisions? How is unemployment measured in relation to the Phillips curve? What role did the Phillips curve play in New Keynesian economics? How do different countries' experiences validate the Phillips curve? What empirical models are used to test the Phillips curve? How does the Phillips curve relate to wage inflation? How did the Phillips curve inform fiscal policy during recessions? How is the Phillips curve applied in modern macroeconomic forecasting? How does the Phillips curve interact with supply shocks? | Which founding fathers were key authors of the Declaration? How did the Declaration affect slavery debates in the U.S.? What role did the Declaration play in the Revolutionary War? How were the colonies mobilized after the Declaration? How did newspapers and pamphlets spread the Declaration? What influence did the Declaration have on other independence movements? How did international law view the Declaration at the time? How did the Declaration inspire subsequent U.S. legislation? How did the Declaration affect Native American relations? How did the Declaration shape early U.S. political parties? | How did General Relativity predict the bending of light? How was General Relativity confirmed during the 1919 solar eclipse? How does General Relativity influence gravitational wave research? How did General Relativity impact quantum theory? How does General Relativity affect modern cosmological models? How do black hole studies rely on General Relativity? How does General Relativity explain time dilation near massive objects? How did General Relativity change our understanding of space-time? How does General Relativity relate to the expansion of the universe? How are relativistic effects measured in particle accelerators? |

### Root-level Prompt

Knowledge fact: "DNA double helix is a fundamental concept in molecular biology."
Generate 3 multiple-choice questions targeting broad, domain-level understanding (root-level). Each question should have 4 answer options (A, B, C, D), one correct answer, and 3 plausible distractors.

### Intermediate-level Prompt

Knowledge fact: "DNA double helix discovery influenced the field of genetics."
Generate 3 multiple-choice questions targeting intermediate-level understanding using the same format.

### Leaf-level Prompt

Knowledge fact: "DNA double helix was discovered in 1953 by Watson and Crick."
Generate 3 multiple-choice questions targeting leaf-level understanding (specific facts). Ensure 4 answer options, one correct answer, and 3 plausible distractors.

## A.4 PROBE TYPES

From each generated question template, we derive six probe types to evaluate different aspects of model behavior:

- **Direct Probe:** Queries the target fact in its canonical direction.
- **Reverse Probe:** Queries the fact in the inverted relation to test bidirectional consistency.
- **Multi-hop Probe:** Tests knowledge propagation by asking indirectly via intermediate nodes.
- **Contextual Probe:** Embeds the fact in a rich or distractor-laden context.
- **Conflict Probe:** Presents contradictory or competing information to assess resolution.
- **Comparison Probe:** Forces a choice between multiple candidates to evaluate selective updating.

Example prompts for the four subjects are shown in Table 5.

Table 5: Example probes across four subject domains, illustrating six probe types.

| Subject | Example Probes |
|---|---|
| Biology (DNA double helix) | **Direct:** When was the DNA double helix discovered?
**Reverse:** Which molecule's structure was determined in 1953 as a double helix?
**Multi-hop:** Who were the key scientists whose discovery of the DNA structure influenced modern genetics?
**Contextual:** The DNA double helix discovery transformed molecular biology. In which year was this breakthrough made?
**Conflict:** Some sources claim 1952, others 1953. Which year is correct?
**Comparison:** Was the DNA double helix discovered in 1953 or 1955? |
| Economics (Phillips curve) | **Direct:** What relationship does the Phillips curve describe?
**Reverse:** Which economic principle captures the link between inflation and unemployment?
**Multi-hop:** Which macroeconomic models rely on understanding the inflation-unemployment trade-off?
**Contextual:** The Phillips curve has shaped monetary policy debates. What relationship does it represent?
**Conflict:** Some argue it holds only short-term, others claim long-term relevance. Which is correct?
**Comparison:** Does the Phillips curve describe inflation-unemployment or wage-productivity trade-offs? |
| History (Declaration of Independence) | **Direct:** In what year was the Declaration of Independence signed?
**Reverse:** Which historical document was signed in 1776?
**Multi-hop:** Which events or congresses led to the signing of the Declaration?
**Contextual:** Amid the Revolutionary era, the Declaration was signed. Which year did this occur?
**Conflict:** Some accounts state July 2, others July 4. Which is correct?
**Comparison:** Was the Declaration signed in 1776 or 1777? |
| Physics (General Relativity) | **Direct:** In what year did Einstein publish the theory of General Relativity?
**Reverse:** Which scientist published General Relativity in 1915?
**Multi-hop:** Which subsequent physics phenomena were explained following Einstein's publication?
**Contextual:** General Relativity transformed our understanding of space-time. When was it published?
**Conflict:** Some sources claim 1915, others 1916. Which is correct?
**Comparison:** Did Einstein publish General Relativity in 1915 or 1920? |

## A.5 MULTIPLE-CHOICE FORMATTING AND DATA RECORDS

All probes are formatted as four-choice QA items consistent with MMLU. Distractors are created via entity substitution and paraphrasing. An example for the four subjects is shown in Table 6

Table 6: Compact multiple-choice probes across four subjects. Correct answers indicated.

| Subject | Example Multiple Choice |
|---|---|
| Biology (DNA double helix) | **Q:** When was the DNA double helix discovered? 
 A. 1953 (Correct)    B. 1955    C. 1962    D. 1947 |
| Economics (Phillips curve) | **Q:** What relationship does the Phillips curve describe? 
 A. Inflation vs. unemployment (Correct)    B. Wage vs. productivity    C. Interest rate vs. investment    D. Savings vs. consumption |
| History (Declaration of Independence) | **Q:** In what year was the Declaration of Independence signed? 
 A. 1776 (Correct)    B. 1775    C. 1777    D. 1781 |
| Physics (General Relativity) | **Q:** In what year did Einstein publish the theory of General Relativity? 
 A. 1915 (Correct)    B. 1920    C. 1912    D. 1918 |

## A.6 QUALITY CONTROL

Items undergo:

1. Format validation (4 options, 1 correct answer)
2. Factual validation against the KG
3. Distractor validation (plausible yet incorrect)

Manual spot checks ensure grammaticality and factual correctness; GPT-generated distractors are cross-checked with encyclopedic sources.

## A.7 DOMAIN AND SAMPLE GRANULARITY

Domains include **Biology**, **History**, **Physics**, and **Economics**, each curated into a structured KG. Our study focuses on modifying one fact at a time; all QA items are anchored on this fact. Multiple templates per node level, probe types, paraphrases, and varying data scales (1, 10, 100, 1,000, 10,000) allow a single fact to generate up to millions of QA items for large-scale evaluation.

## B PROPAGATION ASYMMETRY METRICS AND ALGORITHM

To quantify over- vs. under-spreading rigorously, we define:

$$\text{Collateral Change Ratio (CCR)} = \frac{1}{|\mathcal{Q}_{\text{related}}|} \sum_{x \in \mathcal{Q}_{\text{related}}} d\big(p_{\theta'}(\cdot \mid x), p_\theta(\cdot \mid x)\big), \tag{4}$$

$$\text{Residual Retention (RR)} = \frac{1}{|\mathcal{Q}_{\text{related}}|} \sum_{x \in \mathcal{Q}_{\text{related}}} \mathbf{1}\big[\hat{y}_{\theta'}(x) = y_\theta(x)\big], \tag{5}$$

where $\mathcal{Q}_{\text{related}}$ denotes structurally related probes, $p_\theta$ and $p_{\theta'}$ are predictions before and after intervention, and $d(\cdot, \cdot)$ is a distance metric (KL, label change, etc.).

**Propagation Evaluation Algorithm:**

1. Select a target node at hierarchy level $L$.
2. Apply editing or unlearning to the node.
3. Measure direct accuracy on target node ($Acc_{\text{direct}}$).
4. Measure multi-hop accuracy on related nodes ($Acc_{\text{multi-hop}}$).
5. Compute CCR and RR metrics:
   - Editing: $1 - Acc_{\text{multi-hop}}$ as proxy for over-spreading.
   - Unlearning: $Acc_{\text{multi-hop}}$ as proxy for under-spreading.
6. Repeat for all hierarchy levels and average over domains.

# C    STRESS TESTING

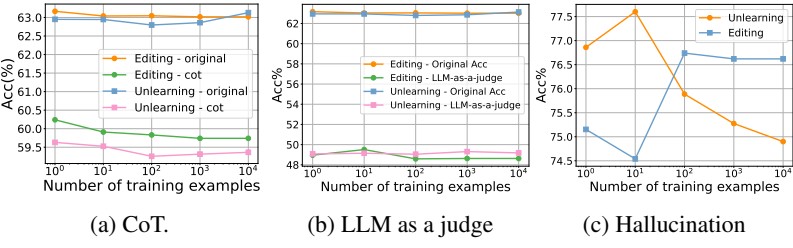

(a) CoT.                 (b) LLM as a judge              (c) Hallucination

Figure 8: Stress testing.

We evaluate *instruction-following* ability (Figure 8b) and *hallucination* on the TruthfulQA (Lin et al., 2022) dataset (Figure 8c), testing whether the parameter update $\theta \to \theta'$ preserves desired behavior when executing complex tasks. These evaluations provide a comprehensive view of how the unified framework constrains model updates, ensuring both local alignment with target distributions and global reliability across diverse scenarios.

For **hallucination**, the average accuracy across data scales for unlearning is 76.0%, and for editing is 76.1%, with standard deviations of 0.87 and 0.91 respectively. This indicates that **both editing and unlearning maintain stable performance** under hallucination tests, with no significant increase in spurious behavior.

For **instruction-following**, when measured using an LLM as a judge, editing accuracy drops from 63.0% (original) to 48.6% on average, while unlearning drops from 62.9% to 49.1%. Although the absolute difference is small, editing shows slightly larger variability (standard deviation 0.12%) compared to unlearning (0.10%). This suggests that **editing is more aggressive** in updating targeted knowledge but may slightly perturb complex reasoning tasks, whereas **unlearning better preserves general instruction-following ability**.

# D    ROBUSTNESS AND FAILURE MODE

## D.1    ADVERSARIAL ROBUSTNESS ANALYSIS

To complement our main text results, we provide a detailed analysis of adversarial robustness for editing and unlearning interventions. Adversarial robustness is evaluated by exposing the model to deliberately misleading or deceptive probes, which combine unrelated or conflicting concepts. This stresses the model's ability to maintain prior knowledge ($\mathcal{Q}^-$) while incorporating updates.

**Experimental Setup**    We vary the number of training examples used for each intervention: 1, 10, 100, 1000, and 10,000. For each data scale, we measure two complementary performance metrics:

- **Original Accuracy:** The model's performance on standard in-domain probes ($\mathcal{Q}^+$), reflecting whether the intended knowledge update was successfully incorporated without disrupting unrelated facts.
- **Adversarial Accuracy:** The model's performance on *conflict probes*, which contain contradictory or misleading information. These probes test the model's *robustness against adversarial perturbations*, i.e., whether it can resist adopting incorrect or conflicting knowledge while maintaining its updated and preserved facts.

By comparing original and adversarial accuracy across training scales and intervention types (editing vs. unlearning), we assess:

- The sensitivity of each method to misleading inputs.
- How stability and resistance to conflicts evolve as more examples are provided.
- Differences in trade-offs between aggressive updates (editing) and conservative updates (unlearning).

This setup allows us to systematically quantify the *adversarial robustness* of interventions, linking conflict probe performance directly to practical model reliability under deceptive or contradictory inputs.

**Observations**   Our observations are:

- **Editing exhibits strong local updates but high adversarial sensitivity:** Original accuracy remains stable around 63% across all data scales. However, adversarial accuracy drops sharply from 36.7% at 1 example to 31.7% at 10,000 examples. This indicates that while editing successfully enforces target updates, it leaves models vulnerable to misleading inputs, with adversarial failure increasing slightly as data scale grows.
- **Unlearning maintains more stable adversarial performance:** Original accuracy is similar to editing. Adversarial accuracy remains relatively constant around 33–35%, showing that unlearning prioritizes preservation over aggressive enforcement, making the model less sensitive to adversarially constructed probes.
- **Trade-off between update intensity and robustness:** Comparing the two interventions, editing maximizes immediate factual incorporation at the cost of susceptibility to adversarial probes, whereas unlearning provides conservative updates that better preserve prior knowledge, yielding higher adversarial robustness.
- **Data scale effects:** Increasing the number of examples slightly improves adversarial robustness for unlearning (e.g., from 33.3% at 1 example to 34.8% at 1,000 examples), but the trend is less pronounced for editing. This suggests that adding more training data does not fully mitigate adversarial vulnerability for aggressive editing strategies.

**Summary**   These results reinforce the broader trade-offs observed in our main text. Editing achieves stronger local adaptation and in-domain gains, but adversarial robustness is compromised. Unlearning is more conservative, achieving lower immediate gains but maintaining stability under adversarial stress. Together, these findings highlight the importance of considering both factual enforcement and robustness when designing knowledge update strategies in LLMs.

## D.2   FAILURE MODE EXAMPLES

We provide examples of failure mode for each subject as shown in Table 7.

Table 7: Representative examples of each failure mode for the four studied subjects. Each subject is listed in a separate row for readability.

| Subject | Failure Mode | Example |
|---|---|---|
| Biology (DNA) | Under-forgetting (RR) | DNA year remains 1953 after update to 1955 |
| | Over-spreading (CCR) | DNA update changes RNA discovery year |
| | Conflict Emergence | DNA reported as 1953 and 1955 |
| | Knowledge Drift | DNA update causes cell structure errors |
| | Instruction-Following Drop | Fails to explain multi-step DNA replication |
| | Hallucination Increase | Invents molecule "X-DNA" |
| Economics (Phillips curve) | Under-forgetting (RR) | Phillips curve still inflation-unemployment after update |
| | Over-spreading (CCR) | Phillips curve update alters Laffer curve |
| | Conflict Emergence | Links both inflation-unemployment and wages-productivity |
| | Knowledge Drift | Update mispredicts supply-demand |
| | Instruction-Following Drop | Misapplies multi-step economic policy reasoning |
| | Hallucination Increase | Fabricates fictional "Y-Index" |
| History (Declaration) | Under-forgetting (RR) | Declaration year still 1776 after update to 1777 |
| | Over-spreading (CCR) | Declaration update changes Constitution year |
| | Conflict Emergence | Declaration signed 1776 and 1777 |
| | Knowledge Drift | Update affects French Revolution facts |
| | Instruction-Following Drop | Struggles with chronological sequencing of events |
| | Hallucination Increase | Claims fake historical figure influenced Declaration |
| Physics (General Relativity) | Under-forgetting (RR) | GR year remains 1915 after update to 1920 |
| | Over-spreading (CCR) | GR update changes Special Relativity year |
| | Conflict Emergence | GR dated 1915 and 1920 |
| | Knowledge Drift | Update reduces quantum mechanics accuracy |
| | Instruction-Following Drop | Cannot solve multi-step relativity problems |
| | Hallucination Increase | Reports spurious physics law "Relativistic Thermodynamics Law" |

# E  ADDITIONAL METHODS

To further validate the generality of the propagation asymmetry phenomena reported in the main paper, we conducted an additional suite of experiments using multiple independent intervention algorithms, spanning both editing and unlearning paradigms. These experiments were performed on the same four subject domains (*biology*, *economic*, *physics*, *history*), and evaluated at root-, intermediate-, and leaf-level nodes in our conceptual hierarchies.

## E.1  UNLEARNING

We applied the gradient ascent method base on Tofu (Maini et al., 2024) framework with varying numbers of updates across four subjects and multiple training set sizes. The results shown in Table 8 replicate the core findings presented in the main paper:

- propagation remains asymmetric across hierarchy levels,
- leaf nodes experience weaker upward transfer,
- root-level deletions continue to exhibit stronger downward effects.

Importantly, these consistency patterns persist regardless of the number of training examples and irrespective of subject domain, suggesting that the structural behaviors we identified are not artifacts of a particular unlearning implementation.

| Subject | Train Size | Root | Intermediate | Leaf |
|---------|-----------|------|--------------|------|
| biology | 1 | 16.67 | 16.67 | 16.67 |
| biology | 10 | 16.67 | 16.67 | 16.67 |
| biology | 100 | 25.00 | 25.00 | 25.00 |
| biology | 1000 | 29.17 | 25.00 | 16.67 |
| biology | 10000 | 16.67 | 29.17 | 25.00 |
| economic | 1 | 29.17 | 29.17 | 29.17 |
| economic | 10 | 29.17 | 29.17 | 33.33 |
| economic | 100 | 20.83 | 16.67 | 25.00 |
| economic | 1000 | 37.50 | 37.50 | 29.17 |
| economic | 10000 | 37.50 | 20.83 | 25.00 |
| physics | 1 | 25.00 | 25.00 | 25.00 |
| physics | 10 | 25.00 | 25.00 | 25.00 |
| physics | 100 | 16.67 | 20.83 | 20.83 |
| physics | 1000 | 12.50 | 16.67 | 20.83 |
| physics | 10000 | 8.33 | 16.67 | 12.50 |
| history | 1 | 16.67 | 16.67 | 16.67 |
| history | 10 | 16.67 | 16.67 | 16.67 |
| history | 100 | 12.50 | 16.67 | 12.50 |
| history | 1000 | 4.17 | 8.33 | 0.00 |
| history | 10000 | 0.00 | 12.50 | 0.00 |

Table 8: Unlearning experiments using Tofu across domains and hierarchy levels.

## E.2  EDITING

We also evaluated the MEND editing method (Mitchell et al., 2021) on the same corpus of subjects, hierarchy depths, and training sizes. The results shown in Table 9 demonstrate that:

- editing accuracy follows the same hierarchy-dependent plasticity structure observed in the main paper,
- root-level edits continue to propagate downward more strongly than bottom-up corrections,
- leaf nodes remain the easiest to modify reliably.

These findings reinforce that the asymmetry patterns we report are algorithm-agnostic, emerging from the structure of the knowledge graph itself rather than any specific intervention technique.

| Dataset | Train Size | Root | Intermediate | Leaf |
|---|---|---|---|---|
| biology | 1 | 35.2 | 32.7 | 42.1 |
| biology | 10 | 36.1 | 33.2 | 43.3 |
| biology | 100 | 37.6 | 34.4 | 44.7 |
| biology | 1000 | 38.9 | 35.1 | 46.2 |
| biology | 10000 | 20.3 | 18.7 | 40.5 |
| economic | 1 | 45.3 | 42.6 | 50.7 |
| economic | 10 | 46.2 | 43.3 | 49.4 |
| economic | 100 | 47.7 | 44.6 | 52.9 |
| economic | 1000 | 41.3 | 45.7 | 54.1 |
| economic | 10000 | 30.2 | 28.3 | 53.2 |
| physics | 1 | 25.3 | 22.7 | 30.2 |
| physics | 10 | 26.1 | 23.1 | 31.3 |
| physics | 100 | 27.4 | 24.6 | 32.6 |
| physics | 1000 | 28.7 | 25.4 | 27.7 |
| physics | 10000 | 15.2 | 13.4 | 30.3 |
| history | 1 | 10.3 | 9.7 | 12.4 |
| history | 10 | 11.2 | 10.3 | 13.3 |
| history | 100 | 11.1 | 11.4 | 14.1 |
| history | 1000 | 12.6 | 12.8 | 15.3 |
| history | 10000 | 6.1 | 5.7 | 14.8 |

Table 9: Editing experiments using MEND across domains and hierarchy levels.

## F  SEQUENTIAL UPDATE

To further validate our claim that **editing and unlearning behave fundamentally differently**, we additionally conducted *multi-step sequential updates* on multiple facts using Qwen3-14B, LLaMA3-8B, and Gemma-7B. This section reports the results to illustrate the phenomenon clearly.

### SEQUENTIAL EDITING BEHAVIOR

Across multiple sequential edits, the model retains previously edited knowledge with only minor drift. Even after five cumulative edits, the performance on earlier edited facts remains largely stable. This supports our claim that **editing operations are robust and localized**, even under sequential updates.

| Acc | 1 | 1 & 2 | 1 & 2 & 3 | 1 & 2 & 3 & 4 | 1 & 2 & 3 & 4 & 5 |
|---|---|---|---|---|---|
| Edit Fact 1 | 55.0% | 54.5% | 54.0% | 53.8% | 53.5% |
| Edit Fact 2 | — | 48.0% | 47.5% | 47.0% | 46.5% |
| Edit Fact 3 | — | — | 62.0% | 61.5% | 61.0% |
| Edit Fact 4 | — | — | — | 50.0% | 49.5% |
| Edit Fact 5 | — | — | — | — | 57.0% |

Table 10: Sequential editing performance.

### SEQUENTIAL UNLEARNING BEHAVIOR

In contrast, unlearning shows **clear cumulative degradation**. When more facts are removed sequentially, the model's performance on earlier unlearned facts, as well as related queries, drops sharply. This supports our central claim: **Unlearning is inherently more disruptive than editing**, because removing information often affects interconnected knowledge.

| Acc | 1 | 1 & 2 | 1 & 2 & 3 | 1 & 2 & 3 & 4 | 1 & 2 & 3 & 4 & 5 |
|---|---|---|---|---|---|
| Unlearn Fact 1 | 54.2% | 37.7% | 40.5% | 34.3% | 27.2% |
| Unlearn Fact 2 | — | 45.1% | 37.8% | 31.5% | 25.2% |
| Unlearn Fact 3 | — | — | 43.2% | 36.0% | 28.8% |
| Unlearn Fact 4 | — | — | — | 40.5% | 31.5% |
| Unlearn Fact 5 | — | — | — | — | 34.2% |

Table 11: Sequential unlearning performance.

## G ACCURACY RESULT

Editing accuracy for the 13 model across llama3, qwen3, qwq, mistral, gemma and deepseek families are lists below in Table 12. Unlearning accuracy for the 13 model across llama3, qwen3, qwq, mistral, gemma and deepseek families are lists below in Table 13.

## H MODEL SIMILARITY RESULT

**Representation Similarity Analysis** Our unified framework models editing and unlearning as optimizing $\mathcal{L}_{\text{task}}$ against $\mathcal{L}_{\text{pres}}$. While probe-based evaluation measures outcomes on $\mathcal{Q}^+$ and $\mathcal{Q}^-$, it does not reveal how the internal representations change during this optimization. To capture these hidden dynamics, we analyze representational shifts from the original (pre-KnowledgeSmith) state to the post-KnowledgeSmith state using Centered Kernel Alignment (CKA) (Kornblith et al., 2019), KL divergence, L2 distance and Fisher score (Zhang et al., 2022).

For unlearning, these metrics expose a sharp phase transition around 1000 samples: below this point, representations remain close to baseline, but beyond it they reorganize abruptly, suggesting a capacity breakpoint where $\mathcal{L}_{\text{pres}}$ is overwhelmed by repeated optimization on $\mathcal{Q}^+$. Editing, in contrast, produces smoother trajectories. KL divergence and Fisher scores increase steadily with training size, indicating progressive local updates to representations rather than wholesale restructuring. For example, biology edits on DeepSeek-8B show KL and Fisher growing from $(\text{KL}\approx 20, \text{Fisher}\approx 9.7)$ with a single sample to $(\text{KL}\approx 172, \text{Fisher}\approx 93.7)$ at 1000 samples, after which growth plateaus as the optimization stabilizes.

These results demonstrate that **unlearning triggers abrupt phase transitions in representation space once data scale crosses a threshold**, while editing produces gradual, localized adjustments, underscoring the need for representation level analysis beyond probe accuracy.

**Computationally Efficiency**. For the same model on a target dataset of $10,000$ examples, unlearning typically completes in about $1.5$ hours on an NVIDIA H100. Knowledge editing is more resource-intensive (roughly 6 hours). This additional cost highlights the heavier computational demands of precise factual editing.

In summary, **unlearning prioritizes stability and low computational cost, while editing maximizes factual enforcement but risks destabilizing other knowledge and requires more resources.** The choice between the two depends on whether minimizing collateral effects or maximizing certainty of change is the primary goal.

Model similarity for llama3, qwen3, qwq, mistral, gemma and deepseek 6 families are lists below in Tables 14 to 19

## I LLM USAGE

We use large language models (LLMs) only for grammar checking and correction.

Table 12: Editing Accuracy

| Branch | Train Size | Test Set | llama3.2-1b-instruct | | | | llama3-8b-instruct | | | |
|---|---|---|---|---|---|---|---|---|---|---|
| | | | Biology | History | Economic | Physics | Biology | History | Economic | Physics |
| Intermediate | 1 | ID | 45.83 | 25 | 37.5 | 20.83 | 20.83 | 4.17 | 29.17 | 12.5 |
| | | OOD | 44.05 | 44.07 | 44.11 | 44.05 | 63.08 | 63.13 | 63.22 | 63.05 |
| | 10 | ID | 20.83 | 25 | 50 | 33.33 | 25 | 12.5 | 25 | 20.83 |
| | | OOD | 44.04 | 44.06 | 44.02 | 42.06 | 63.07 | 63.04 | 63.1 | 63 |
| | 100 | ID | 45.83 | 25 | 45.83 | 0 | 20.83 | 12.5 | 20.83 | 41.67 |
| | | OOD | 23.35 | 43.99 | 44.08 | 26.88 | 63.05 | 63.13 | 63.12 | 24.72 |
| | 1000 | ID | 4.17 | 25 | 45.83 | 0 | 20.83 | 12.5 | 20.83 | 12.5 |
| | | OOD | 25.22 | 43.98 | 44.17 | 26.85 | 63.06 | 63.03 | 63.09 | 24.3 |
| | 10000 | ID | 4.17 | 25 | 45.83 | 0 | 20.83 | 12.5 | 20.83 | 12.5 |
| | | OOD | 25.22 | 43.98 | 44.17 | 26.85 | 63.06 | 63.03 | 63.09 | 24.3 |
| Root | 1 | ID | 41.67 | 25 | 41.67 | 29.17 | 16.67 | 4.17 | 33.33 | 16.67 |
| | | OOD | 44.17 | 44.16 | 43.99 | 44.15 | 63.04 | 62.91 | 63.1 | 63.05 |
| | 10 | ID | 29.17 | 25 | 45.83 | 29.17 | 12.5 | 4.17 | 33.33 | 25 |
| | | OOD | 44.1 | 44.28 | 44.12 | 44.07 | 63.01 | 62.9 | 63.12 | 63.1 |
| | 100 | ID | 29.17 | 25 | 4.17 | 20.83 | 12.5 | 4.17 | 33.33 | 25 |
| | | OOD | 44.12 | 44.22 | 26.24 | 44.2 | 63 | 62.97 | 63.15 | 63 |
| | 1000 | ID | 29.17 | 25 | 0 | 25 | 12.5 | 4.17 | 33.33 | 16.67 |
| | | OOD | 44.15 | 44.26 | 25.4 | 44.09 | 62.98 | 62.98 | 63.11 | 63.11 |
| | 10000 | ID | 29.17 | 25 | 0 | 25 | 12.5 | 4.17 | 33.33 | 16.67 |
| | | OOD | 44.15 | 44.26 | 25.4 | 44.09 | 62.98 | 62.98 | 63.11 | 63.11 |
| Leaf | 1 | ID | 41.67 | 25 | 33.33 | 25 | 16.67 | 4.17 | 37.5 | 16.67 |
| | | OOD | 44.13 | 44.11 | 44.08 | 44.02 | 63.1 | 63.07 | 63.09 | 63.1 |
| | 10 | ID | 25 | 25 | 62.5 | 20.83 | 16.67 | 4.17 | 37.5 | 25 |
| | | OOD | 44.19 | 44.41 | 43.74 | 43.18 | 63.12 | 62.96 | 63.06 | 62.75 |
| | 100 | ID | 4.17 | 4.17 | 4.17 | 0 | 16.67 | 12.5 | 25 | 4.17 |
| | | OOD | 25.45 | 40.55 | 25.53 | 26.88 | 62.78 | 62.47 | 62.6 | 25.41 |
| | 1000 | ID | 25 | 45.83 | 0 | 8.33 | 16.67 | 4.17 | 25 | 16.67 |
| | | OOD | 25.78 | 23.44 | 26.63 | 24.84 | 62.77 | 59.56 | 62.59 | 24.25 |
| | 10000 | ID | 25 | 45.83 | 0 | 8.33 | 16.67 | 4.17 | 25 | 16.67 |
| | | OOD | 25.78 | 23.44 | 26.63 | 24.84 | 62.77 | 59.56 | 62.59 | 24.25 |

| Branch | Train Size | Test Set | llama3.2-3b-instruct | | | | llama3.3-70b-instruct | | | |
|---|---|---|---|---|---|---|---|---|---|---|
| | | | Biology | History | Economic | Physics | Biology | History | Economic | Physics |
| Intermediate | 1 | ID | 25 | 12.5 | 41.67 | 16.67 | 20.83 | 8.33 | 20.83 | 25 |
| | | OOD | 59.17 | 59.24 | 59.36 | 59.22 | 81.44 | 81.42 | 81.39 | 81.42 |
| | 10 | ID | 12.5 | 0 | 37.5 | 37.5 | 20.83 | 62.5 | 41.67 | 29.17 |
| | | OOD | 56.38 | 56.84 | 58.63 | 58.39 | 81.38 | 81.38 | 81.43 | 81.48 |
| | 100 | ID | 29.17 | 20.83 | 54.17 | 12.5 | 20.83 | 58.33 | 50 | 29.17 |
| | | OOD | 23.47 | 26.9 | 23.32 | 25.84 | 81.46 | 81.26 | 81.33 | 81.39 |
| | 1000 | ID | 4.17 | 45.83 | 41.67 | 0 | 25 | 58.33 | 50 | 29.17 |
| | | OOD | 25.45 | 25.12 | 25.31 | 25.08 | 81.39 | 81.35 | 81.38 | 81.31 |
| | 10000 | ID | 4.17 | 45.83 | 41.67 | 0 | 25 | 58.33 | 50 | 29.17 |
| | | OOD | 25.45 | 25.12 | 25.31 | 25.08 | 81.39 | 81.35 | 81.38 | 81.31 |
| Root | 1 | ID | 25 | 4.17 | 29.17 | 12.5 | 20.83 | 4.17 | 20.83 | 25 |
| | | OOD | 59.2 | 59.28 | 59.24 | 59.34 | 81.41 | 81.46 | 81.46 | 81.43 |
| | 10 | ID | 41.67 | 29.17 | 16.67 | 4.17 | 58.33 | 45.83 | 37.5 | 33.33 |
| | | OOD | 58.76 | 58.73 | 58.37 | 58.72 | 81.41 | 81.46 | 81.39 | 81.51 |
| | 100 | ID | 37.5 | 0 | 41.67 | 29.17 | 58.33 | 25 | 41.67 | 33.33 |
| | | OOD | 23.3 | 26.86 | 24.4 | 25.34 | 81.39 | 81.41 | 81.4 | 81.42 |
| | 1000 | ID | 4.17 | 0 | 20.83 | 25 | 58.33 | 20.83 | 41.67 | 33.33 |
| | | OOD | 25.57 | 26.48 | 25.06 | 25.2 | 81.42 | 81.46 | 81.48 | 81.44 |
| | 10000 | ID | 4.17 | 0 | 20.83 | 25 | 58.33 | 20.83 | 41.67 | 33.33 |
| | | OOD | 25.57 | 26.48 | 25.06 | 25.2 | 81.42 | 81.46 | 81.48 | 81.44 |
| Leaf | 1 | ID | 20.83 | 8.33 | 33.33 | 20.83 | 20.83 | 8.33 | 20.83 | 25 |
| | | OOD | 59.23 | 59.25 | 59.3 | 59.24 | 81.43 | 81.37 | 81.43 | 81.45 |
| | 10 | ID | 37.5 | 8.33 | 45.83 | 29.17 | 25 | 25 | 58.33 | 20.83 |
| | | OOD | 59.13 | 59.26 | 58.35 | 56.96 | 81.43 | 81.38 | 81.41 | 81.34 |
| | 100 | ID | 8.33 | 4.17 | 54.17 | 20.83 | 25 | 20.83 | 58.33 | 20.83 |
| | | OOD | 24.49 | 25.52 | 23.21 | 24.55 | 81.41 | 81.44 | 81.33 | 81.5 |
| | 1000 | ID | 20.83 | 4.17 | 54.17 | 4.17 | 25 | 25 | 62.5 | 20.83 |
| | | OOD | 27.23 | 26.24 | 23.19 | 25.42 | 81.43 | 81.37 | 81.29 | 81.44 |
| | 10000 | ID | 20.83 | 4.17 | 54.17 | 4.17 | 25 | 25 | 62.5 | 20.83 |
| | | OOD | 27.23 | 26.24 | 23.19 | 25.42 | 81.43 | 81.37 | 81.29 | 81.44 |

| Branch | Train Size | Test Set | qwen3-1.7b Biology | History | Economic | Physics | qwen3-32b Biology | History | Economic | Physics |
|---|---|---|---|---|---|---|---|---|---|---|
| Intermediate | 1 | ID | 20.83 | 12.5 | 33.33 | 29.17 | 12.5 | 0 | 25 | 12.5 |
| | | OOD | 53 | 53.99 | 54.05 | 54.08 | 75.07 | 75.11 | 75.19 | 75.09 |
| | 10 | ID | 25 | 12.5 | 33.33 | 29.17 | 20.83 | 0 | 20.83 | 8.33 |
| | | OOD | 53.55 | 53.95 | 54.05 | 54.08 | 75.07 | 75.02 | 75.08 | 75.02 |
| | 100 | ID | 20.83 | 12.5 | 33.33 | 37.5 | 20.83 | 0 | 29.17 | 8.33 |
| | | OOD | 53 | 54 | 53.82 | 53.02 | 75.17 | 75.15 | 74.97 | 75.1 |
| | 1000 | ID | 25 | 12.5 | 33.33 | 37.5 | 20.83 | 0 | 29.17 | 8.33 |
| | | OOD | 53.04 | 53.99 | 53.87 | 53.07 | 75.2 | 75.15 | 74.86 | 75.03 |
| | 10000 | ID | 25 | 12.5 | 33.33 | 37.5 | 20.83 | 0 | 29.17 | 8.33 |
| | | OOD | 53.04 | 53.99 | 53.87 | 53.07 | 75.2 | 75.15 | 74.86 | 75.03 |
| Root | 1 | ID | 37.5 | 45.83 | 25 | 12.5 | 20.83 | 16.67 | 45.83 | 8.33 |
| | | OOD | 53.81 | 53.8 | 53.7 | 54 | 75.2 | 75.12 | 75.21 | 75 |
| | 10 | ID | 33.33 | 45.83 | 16.67 | 12.5 | 16.67 | 25 | 33.33 | 16.67 |
| | | OOD | 53.78 | 53.78 | 53.87 | 54 | 75.05 | 75.17 | 75.02 | 75.07 |
| | 100 | ID | 29.17 | 45.83 | 25 | 20.83 | 20.83 | 12.5 | 12.5 | 16.67 |
| | | OOD | 53.74 | 53.8 | 53.7 | 53.65 | 75.02 | 75.15 | 74.98 | 75.04 |
| | 1000 | ID | 29.17 | 45.83 | 25 | 20.83 | 20.83 | 16.67 | 12.5 | 16.67 |
| | | OOD | 53.79 | 53.82 | 53.75 | 53.75 | 75.1 | 75.1 | 75 | 75.06 |
| | 10000 | ID | 29.17 | 45.83 | 25 | 20.83 | 20.83 | 16.67 | 12.5 | 16.67 |
| | | OOD | 53.79 | 53.82 | 53.75 | 53.75 | 75.1 | 75.1 | 75 | 75.06 |
| Leaf | 1 | ID | 16.67 | 37.5 | 37.5 | 25 | 20.83 | 0 | 25 | 8.33 |
| | | OOD | 53.97 | 53.19 | 53.28 | 51.66 | 75.15 | 75.1 | 75.07 | 75.07 |
| | 10 | ID | 16.67 | 29.17 | 8.33 | 16.67 | 20.83 | 25 | 29.17 | 4.17 |
| | | OOD | 53.87 | 53.86 | 53.76 | 53.92 | 74.9 | 75.05 | 75.15 | 75.17 |
| | 100 | ID | 16.67 | 33.33 | 29.17 | 25 | 29.17 | 16.67 | 54.17 | 4.17 |
| | | OOD | 54.2 | 53.2 | 53.33 | 51.66 | 75.1 | 75.12 | 74.88 | 75.13 |
| | 1000 | ID | 16.67 | 37.5 | 37.5 | 33.33 | 25 | 4.17 | 41.67 | 4.17 |
| | | OOD | 53.69 | 53.19 | 53.28 | 39.71 | 75.16 | 74.18 | 74.69 | 75.15 |
| | 10000 | ID | 16.67 | 37.5 | 37.5 | 33.33 | 25 | 4.17 | 41.67 | 4.17 |
| | | OOD | 53.69 | 53.19 | 53.28 | 39.71 | 75.16 | 74.18 | 74.69 | 75.15 |

| Branch | Train Size | Test Set | qwen3-14b Biology | History | Economic | Physics | qwq-32b Biology | History | Economic | Physics |
|---|---|---|---|---|---|---|---|---|---|---|
| Intermediate | 1 | ID | 20.83 | 8.33 | 16.67 | 25 | 16.67 | 4.17 | 70.83 | 12.5 |
| | | OOD | 73.84 | 73.86 | 73.89 | 73.94 | 77.4 | 77.45 | 77.42 | 77.45 |
| | 10 | ID | 20.83 | 0 | 4.17 | 20.83 | 12.5 | 4.17 | 33.33 | 12.5 |
| | | OOD | 73.78 | 73.54 | 73.42 | 73.62 | 77.36 | 77.39 | 77.42 | 77.35 |
| | 100 | ID | 25 | 4.17 | 4.17 | 16.67 | 16.67 | 0 | 37.5 | 12.5 |
| | | OOD | 73.76 | 73.45 | 73.36 | 73.56 | 77.28 | 77.43 | 77.41 | 77.39 |
| | 1000 | ID | 25 | 4.17 | 4.17 | 16.67 | 16.67 | 0 | 37.5 | 12.5 |
| | | OOD | 73.73 | 73.42 | 73.33 | 73.56 | 77.33 | 77.43 | 77.41 | 77.4 |
| | 10000 | ID | 25 | 4.17 | 4.17 | 16.67 | 16.67 | 0 | 37.5 | 12.5 |
| | | OOD | 73.73 | 73.42 | 73.33 | 73.56 | 77.33 | 77.43 | 77.41 | 77.4 |
| Root | 1 | ID | 41.67 | 20.83 | 33.33 | 25 | 12.5 | 20.83 | 20.83 | 12.5 |
| | | OOD | 73.86 | 73.9 | 73.86 | 73.79 | 77.4 | 77.47 | 77.37 | 77.42 |
| | 10 | ID | 20.83 | 8.33 | 33.33 | 16.67 | 20.83 | 12.5 | 16.67 | 12.5 |
| | | OOD | 73.81 | 73.71 | 73.87 | 73.85 | 77.35 | 77.53 | 77.47 | 77.38 |
| | 100 | ID | 16.67 | 12.5 | 37.5 | 16.67 | 16.67 | 12.5 | 16.67 | 12.5 |
| | | OOD | 73.71 | 73.68 | 73.81 | 73.58 | 77.43 | 77.39 | 77.43 | 77.3 |
| | 1000 | ID | 16.67 | 12.5 | 41.67 | 16.67 | 16.67 | 12.5 | 16.67 | 12.5 |
| | | OOD | 73.73 | 73.64 | 73.84 | 73.55 | 77.44 | 77.39 | 77.45 | 77.35 |
| | 10000 | ID | 16.67 | 12.5 | 41.67 | 16.67 | 16.67 | 12.5 | 16.67 | 12.5 |
| | | OOD | 73.73 | 73.64 | 73.84 | 73.55 | 77.44 | 77.39 | 77.45 | 77.35 |
| Leaf | 1 | ID | 25 | 0 | 16.67 | 20.83 | 16.67 | 0 | 33.33 | 12.5 |
| | | OOD | 73.89 | 73.89 | 73.87 | 73.88 | 77.48 | 77.39 | 77.4 | 77.48 |
| | 10 | ID | 20.83 | 4.17 | 33.33 | 12.5 | 25 | 0 | 41.67 | 12.5 |
| | | OOD | 73.91 | 73.63 | 73.63 | 73.69 | 77.5 | 77.37 | 77.33 | 77.42 |
| | 100 | ID | 20.83 | 0 | 29.17 | 16.67 | 29.17 | 0 | 29.17 | 12.5 |
| | | OOD | 73.66 | 73.39 | 73.42 | 73.6 | 77.4 | 77.52 | 77.33 | 77.4 |
| | 1000 | ID | 20.83 | 0 | 8.33 | 16.67 | 20.83 | 0 | 41.67 | 12.5 |
| | | OOD | 65.94 | 73.39 | 39.3 | 73.5 | 77.27 | 77.52 | 68.74 | 77.28 |
| | 10000 | ID | 20.83 | 0 | 8.33 | 16.67 | 20.83 | 0 | 41.67 | 12.5 |
| | | OOD | 65.94 | 73.39 | 39.3 | 73.5 | 77.27 | 77.52 | 68.74 | 77.28 |

| Branch | Train Size | Test Set | mistral-Small-24B-Instruct-2501 | | | | gemma-2b | | | |
|---|---|---|---|---|---|---|---|---|---|---|
| | | | Biology | History | Economic | Physics | Biology | History | Economic | Physics |
| Intermediate | 1 | ID | 16.67 | 50 | 12.5 | 20.83 | 8.33 | 4.17 | 12.5 | 12.5 |
| | | OOD | 73.4 | 73.34 | 73.24 | 73.39 | 30.46 | 30.63 | 30.49 | 30.53 |
| | 10 | ID | 50 | 45.83 | 16.67 | 0 | 20.83 | 12.5 | 29.17 | 4.17 |
| | | OOD | 24.47 | 22.99 | 25.2 | 25.51 | 29.4 | 30.37 | 29.06 | 30.29 |
| | 100 | ID | 29.17 | 45.83 | 54.17 | 58.33 | 16.67 | 45.83 | 37.5 | 29.17 |
| | | OOD | 24.22 | 22.95 | 24.16 | 23 | 25.81 | 24.08 | 26.16 | 26.54 |
| | 1000 | ID | 45.83 | 45.83 | 54.17 | 58.33 | 16.67 | 45.83 | 54.17 | 58.33 |
| | | OOD | 22.87 | 22.95 | 24.16 | 23 | 25.81 | 22.95 | 22.95 | 22.95 |
| | 10000 | ID | 45.83 | 45.83 | 54.17 | 58.33 | 16.67 | 45.83 | 54.17 | 58.33 |
| | | OOD | 22.87 | 22.95 | 24.16 | 23 | 25.81 | 22.95 | 22.95 | 22.95 |
| Root | 1 | ID | 33.33 | 20.83 | 16.67 | 45.83 | 4.17 | 0.0 | 16.67 | 4.17 |
| | | OOD | 73.42 | 73.42 | 73.16 | 73.39 | 30.54 | 30.59 | 30.64 | 30.54 |
| | 10 | ID | 45.83 | 54.17 | 4.17 | 58.33 | 8.33 | 8.33 | 12.5 | 33.33 |
| | | OOD | 22.95 | 25.2 | 25.27 | 23.14 | 30.34 | 27.18 | 30.55 | 25.79 |
| | 100 | ID | 45.83 | 37.5 | 54.17 | 58.33 | 4.17 | 50.0 | 25.0 | 54.17 |
| | | OOD | 22.83 | 24.4 | 23.32 | 22.99 | 29.3 | 24.51 | 29.13 | 23.74 |
| | 1000 | ID | 45.83 | 37.5 | 41.67 | 37.5 | 45.83 | 45.83 | 54.17 | 58.33 |
| | | OOD | 23.14 | 24.4 | 24.76 | 25.12 | 22.95 | 22.95 | 22.95 | 22.95 |
| | 10000 | ID | 45.83 | 37.5 | 41.67 | 37.5 | 45.83 | 45.83 | 54.17 | 58.33 |
| | | OOD | 23.14 | 24.4 | 24.76 | 25.12 | 22.95 | 22.95 | 22.95 | 22.95 |
| Leaf | 1 | ID | 50 | 41.67 | 83.33 | 4.17 | 8.33 | 33.33 | 25.0 | 0.0 |
| | | OOD | 73.32 | 73.24 | 73.14 | 73.42 | 30.25 | 28.98 | 30.64 | 30.44 |
| | 10 | ID | 4.17 | 45.83 | 4.17 | 41.67 | 12.5 | 33.33 | 20.83 | 4.17 |
| | | OOD | 25.47 | 22.95 | 25.54 | 25.19 | 27.7 | 24.6 | 25.28 | 29.08 |
| | 100 | ID | 8.33 | 45.83 | 54.17 | 8.33 | 25.0 | 37.5 | 37.5 | 41.67 |
| | | OOD | 26.63 | 22.96 | 22.95 | 24.61 | 24.94 | 24.19 | 24.9 | 24.07 |
| | 1000 | ID | 41.67 | 45.83 | 54.17 | 8.33 | 45.83 | 45.83 | 54.17 | 58.33 |
| | | OOD | 23.54 | 22.96 | 22.95 | 24.61 | 22.83 | 22.95 | 22.95 | 22.95 |
| | 10000 | ID | 41.67 | 45.83 | 54.17 | 8.33 | 45.83 | 45.83 | 54.17 | 58.33 |
| | | OOD | 23.54 | 22.96 | 22.95 | 24.61 | 22.83 | 22.95 | 22.95 | 22.95 |
| | | | mistral-Large-Instruct-2411 | | | | gemma-7b | | | |
| | | | Biology | History | Economic | Physics | Biology | History | Economic | Physics |
| Intermediate | 1 | ID | 25.0 | 62.5 | 25.0 | 12.5 | 45.83 | 37.5 | 41.67 | 45.83 |
| | | OOD | 82.13 | 82.42 | 82.22 | 82.37 | 59.22 | 58.96 | 56.69 | 57.78 |
| | 10 | ID | 0.0 | 45.83 | 41.67 | 37.5 | 45.83 | 45.83 | 54.17 | 50.0 |
| | | OOD | 26.89 | 22.97 | 24.53 | 24.7 | 22.95 | 22.95 | 22.95 | 23.25 |
| | 100 | ID | 16.67 | 62.5 | 25.0 | 50.0 | 25.0 | 45.83 | 50.0 | 41.67 |
| | | OOD | 23.89 | 25.84 | 25.0 | 23.05 | 24.2 | 22.95 | 23.11 | 23.11 |
| | 1000 | ID | 16.67 | 62.5 | 25.0 | 50.0 | 29.17 | 54.17 | 8.33 | 66.67 |
| | | OOD | 23.89 | 25.84 | 25.0 | 23.05 | 24.9 | 25.59 | 25.22 | 24.68 |
| | 10000 | ID | 16.67 | 62.5 | 25.0 | 50.0 | 29.17 | 54.17 | 8.33 | 66.67 |
| | | OOD | 23.89 | 25.84 | 25.0 | 23.05 | 24.9 | 25.59 | 25.22 | 24.68 |
| Root | 1 | ID | 25.0 | 12.5 | 45.83 | 62.5 | 37.5 | 41.67 | 50.0 | 16.67 |
| | | OOD | 82.25 | 82.22 | 82.24 | 82.25 | 59.7 | 59.56 | 57.63 | 59.74 |
| | 10 | ID | 0.0 | 0.0 | 4.17 | 58.33 | 45.83 | 41.67 | 33.33 | 58.33 |
| | | OOD | 26.19 | 26.89 | 25.41 | 22.95 | 28.43 | 22.97 | 29.24 | 22.97 |
| | 100 | ID | 8.33 | 45.83 | 37.5 | 58.33 | 45.83 | 45.83 | 45.83 | 50.0 |
| | | OOD | 26.86 | 22.95 | 24.64 | 23.0 | 22.95 | 23.07 | 22.95 | 24.13 |
| | 1000 | ID | 8.33 | 45.83 | 37.5 | 58.33 | 33.33 | 45.83 | 20.83 | 54.17 |
| | | OOD | 26.86 | 22.95 | 24.64 | 23.0 | 23.98 | 23.07 | 23.24 | 23.34 |
| | 10000 | ID | 8.33 | 45.83 | 37.5 | 58.33 | 33.33 | 45.83 | 20.83 | 54.17 |
| | | OOD | 26.86 | 22.95 | 24.64 | 23.0 | 23.98 | 23.07 | 23.24 | 23.34 |
| Leaf | 1 | ID | 54.17 | 29.17 | 41.67 | 37.5 | 45.83 | 45.83 | 54.17 | 33.33 |
| | | OOD | 82.19 | 82.25 | 82.07 | 82.08 | 22.82 | 22.97 | 22.95 | 59.29 |
| | 10 | ID | 4.17 | 0.0 | 54.17 | 58.33 | 45.83 | 45.83 | 45.83 | 58.33 |
| | | OOD | 25.47 | 25.51 | 22.95 | 23.07 | 22.95 | 23.07 | 23.33 | 22.87 |
| | 100 | ID | 50.0 | 0.0 | 45.83 | 54.17 | 37.5 | 41.67 | 41.67 | 58.33 |
| | | OOD | 23.05 | 24.6 | 24.69 | 25.55 | 23.96 | 23.78 | 23.38 | 22.94 |
| | 1000 | ID | 50.0 | 0.0 | 45.83 | 54.17 | 4.17 | 54.17 | 4.17 | 4.17 |
| | | OOD | 23.05 | 24.6 | 24.69 | 25.55 | 25.48 | 24.49 | 25.52 | 25.54 |
| | 10000 | ID | 50.0 | 0.0 | 45.83 | 54.17 | 4.17 | 54.17 | 4.17 | 4.17 |
| | | OOD | 23.05 | 24.6 | 24.69 | 25.55 | 25.48 | 24.49 | 25.52 | 25.54 |

| Branch | Train Size | Test Set | DeepSeek-R1-0528-Qwen3-8B | | | |
| | | | Biology | History | Economic | Physics |
|---|---|---|---|---|---|---|
| Intermediate | 1 | ID | 8.33 | 0.0 | 20.83 | 25.0 |
| | | OOD | 65.99 | 66.09 | 66.0 | 66.02 |
| | 10 | ID | 25.0 | 0.0 | 33.33 | 29.17 |
| | | OOD | 65.94 | 65.93 | 66.01 | 65.95 |
| | 100 | ID | 16.67 | 0.0 | 33.33 | 33.33 |
| | | OOD | 65.95 | 65.94 | 66.07 | 65.89 |
| | 1000 | ID | 16.67 | 0.0 | 33.33 | 37.5 |
| | | OOD | 65.9 | 65.94 | 66.07 | 66.02 |
| | 10000 | ID | 16.67 | 0.0 | 33.33 | 37.5 |
| | | OOD | 65.9 | 65.94 | 66.07 | 66.02 |
| Root | 1 | ID | 12.5 | 8.33 | 0.0 | 45.83 |
| | | OOD | 66.07 | 65.96 | 28.17 | 65.94 |
| | 10 | ID | 12.5 | 8.33 | 16.67 | 45.83 |
| | | OOD | 66.02 | 65.99 | 65.98 | 65.93 |
| | 100 | ID | 12.5 | 4.17 | 8.33 | 45.83 |
| | | OOD | 65.97 | 66.1 | 66.02 | 66.0 |
| | 1000 | ID | 12.5 | 4.17 | 8.33 | 45.83 |
| | | OOD | 65.92 | 66.02 | 65.99 | 65.92 |
| | 10000 | ID | 12.5 | 4.17 | 8.33 | 45.83 |
| | | OOD | 65.92 | 66.02 | 65.99 | 65.92 |
| Leaf | 1 | ID | 16.67 | 0.0 | 20.83 | 25.0 |
| | | OOD | 65.9 | 65.96 | 66.09 | 65.92 |
| | 10 | ID | 20.83 | 8.33 | 37.5 | 25.0 |
| | | OOD | 65.83 | 65.84 | 65.92 | 65.92 |
| | 100 | ID | 16.67 | 4.17 | 29.17 | 25.0 |
| | | OOD | 65.95 | 65.77 | 65.92 | 65.76 |
| | 1000 | ID | 16.67 | 4.17 | 25.0 | 20.83 |
| | | OOD | 66.02 | 65.77 | 64.29 | 65.8 |
| | 10000 | ID | 16.67 | 4.17 | 25.0 | 20.83 |
| | | OOD | 66.02 | 65.77 | 64.29 | 65.8 |

Table 13: Unlearning Accuracy

| Branch | Train Size | Test Set | llama3.2-1b-instruct | | | | llama3-8b-instruct | | | |
|---|---|---|---|---|---|---|---|---|---|---|
| | | | Biology | History | Economic | Physics | Biology | History | Economic | Physics |
| Intermediate | 1 | ID | 25.0 | 20.83 | 33.33 | 12.5 | 16.67 | 4.17 | 29.17 | 12.5 |
| | | OOD | 32.69 | 32.69 | 32.69 | 32.69 | 63.01 | 63.01 | 63.01 | 63.01 |
| | 10 | ID | 20.83 | 20.83 | 33.33 | 16.67 | 16.67 | 4.17 | 29.17 | 12.5 |
| | | OOD | 32.75 | 32.74 | 32.74 | 32.6 | 63.01 | 63.0 | 63.0 | 62.98 |
| | 100 | ID | 29.17 | 20.83 | 33.33 | 16.67 | 16.67 | 4.17 | 37.5 | 12.5 |
| | | OOD | 32.55 | 32.93 | 32.76 | 32.77 | 62.85 | 62.8 | 62.84 | 62.75 |
| | 1000 | ID | 33.33 | 4.17 | 37.5 | 16.67 | 16.67 | 4.17 | 33.33 | 12.5 |
| | | OOD | 32.67 | 33.91 | 32.72 | 32.51 | 62.93 | 62.82 | 62.91 | 62.85 |
| | 10000 | ID | 33.33 | 4.17 | 37.5 | 16.67 | 16.67 | 4.17 | 33.33 | 12.5 |
| | | OOD | 32.67 | 33.91 | 32.72 | 32.51 | 62.93 | 62.82 | 62.91 | 62.85 |
| Root | 1 | ID | 25.0 | 20.83 | 33.33 | 12.5 | 16.67 | 4.17 | 29.17 | 12.5 |
| | | OOD | 32.69 | 32.69 | 32.69 | 32.69 | 63.01 | 63.01 | 63.01 | 63.01 |
| | 10 | ID | 20.83 | 20.83 | 33.33 | 16.67 | 16.67 | 4.17 | 29.17 | 12.5 |
| | | OOD | 32.63 | 32.61 | 32.55 | 32.69 | 63.0 | 63.0 | 63.02 | 63.0 |
| | 100 | ID | 29.17 | 16.67 | 33.33 | 37.5 | 16.67 | 4.17 | 33.33 | 12.5 |
| | | OOD | 32.97 | 32.74 | 32.84 | 32.94 | 62.91 | 62.98 | 62.87 | 62.89 |
| | 1000 | ID | 16.67 | 8.33 | 37.5 | 4.17 | 16.67 | 4.17 | 33.33 | 12.5 |
| | | OOD | 32.66 | 32.86 | 33.11 | 33.26 | 63.01 | 62.99 | 62.81 | 62.89 |
| | 10000 | ID | 16.67 | 8.33 | 37.5 | 4.17 | 16.67 | 4.17 | 33.33 | 12.5 |
| | | OOD | 32.66 | 32.86 | 33.11 | 33.26 | 63.01 | 62.99 | 62.81 | 62.89 |
| Leaf | 1 | ID | 25.0 | 20.83 | 33.33 | 12.5 | 16.67 | 4.17 | 29.17 | 12.5 |
| | | OOD | 32.69 | 32.69 | 32.69 | 32.69 | 63.01 | 63.01 | 63.01 | 63.01 |
| | 10 | ID | 25.0 | 20.83 | 37.5 | 12.5 | 16.67 | 4.17 | 29.17 | 12.5 |
| | | OOD | 32.69 | 32.58 | 32.69 | 32.74 | 62.98 | 63.02 | 62.99 | 62.98 |
| | 100 | ID | 25.0 | 20.83 | 25.0 | 16.67 | 16.67 | 4.17 | 33.33 | 12.5 |
| | | OOD | 32.57 | 32.85 | 32.73 | 32.73 | 62.75 | 62.75 | 63.02 | 62.75 |
| | 1000 | ID | 20.83 | 12.5 | 20.83 | 20.83 | 16.67 | 4.17 | 33.33 | 12.5 |
| | | OOD | 32.68 | 33.08 | 32.27 | 31.68 | 62.68 | 62.53 | 62.98 | 62.69 |
| | 10000 | ID | 20.83 | 12.5 | 20.83 | 20.83 | 16.67 | 4.17 | 33.33 | 12.5 |
| | | OOD | 32.68 | 33.08 | 32.27 | 31.68 | 62.68 | 62.53 | 62.98 | 62.69 |

| Branch | Train Size | Test Set | llama3.2-3b-instruct | | | | llama3.3-70b-instruct | | | |
|---|---|---|---|---|---|---|---|---|---|---|
| | | | Biology | History | Economic | Physics | Biology | History | Economic | Physics |
| Intermediate | 1 | ID | 20.83 | 4.17 | 41.67 | 16.67 | 20.83 | 8.33 | 20.83 | 20.83 |
| | | OOD | 59.33 | 59.33 | 59.33 | 59.37 | 81.33 | 81.33 | 81.33 | 81.33 |
| | 10 | ID | 20.83 | 4.17 | 41.67 | 16.67 | 20.83 | 8.33 | 20.83 | 20.83 |
| | | OOD | 59.17 | 59.29 | 59.27 | 59.24 | 81.33 | 81.33 | 81.33 | 81.33 |
| | 100 | ID | 25.0 | 4.17 | 41.67 | 16.67 | 20.83 | 8.33 | 20.83 | 20.83 |
| | | OOD | 59.07 | 59.05 | 59.51 | 59.06 | 81.35 | 81.36 | 81.35 | 81.35 |
| | 1000 | ID | 16.67 | 4.17 | 41.67 | 12.5 | 20.83 | 8.33 | 29.17 | 29.72 |
| | | OOD | 59.14 | 59.22 | 59.13 | 59.3 | 81.38 | 81.47 | 81.41 | 81.37 |
| | 10000 | ID | 16.67 | 4.17 | 41.67 | 12.5 | 20.83 | 8.33 | 29.17 | 29.72 |
| | | OOD | 59.14 | 59.22 | 59.13 | 59.3 | 81.38 | 81.47 | 81.41 | 81.37 |
| Root | 1 | ID | 20.83 | 4.17 | 41.67 | 16.67 | 20.83 | 8.33 | 20.83 | 20.83 |
| | | OOD | 59.33 | 59.33 | 59.33 | 59.37 | 81.33 | 81.33 | 81.33 | 81.33 |
| | 10 | ID | 20.83 | 4.17 | 41.67 | 16.67 | 20.83 | 8.33 | 20.83 | 20.83 |
| | | OOD | 59.29 | 59.3 | 59.34 | 59.2 | 81.33 | 81.33 | 81.33 | 81.33 |
| | 100 | ID | 25.0 | 4.17 | 37.5 | 16.67 | 20.83 | 8.33 | 20.83 | 20.83 |
| | | OOD | 58.94 | 58.98 | 59.51 | 59.07 | 81.38 | 81.35 | 81.37 | 81.35 |
| | 1000 | ID | 16.67 | 4.17 | 41.67 | 12.5 | 20.83 | 8.33 | 25.0 | 23.33 |
| | | OOD | 58.96 | 58.99 | 59.41 | 59.12 | 81.39 | 81.33 | 81.41 | 81.33 |
| | 10000 | ID | 16.67 | 4.17 | 41.67 | 12.5 | 20.83 | 8.33 | 25.0 | 23.33 |
| | | OOD | 58.96 | 58.99 | 59.41 | 59.12 | 81.39 | 81.33 | 81.41 | 81.33 |
| Leaf | 1 | ID | 20.83 | 4.17 | 41.67 | 16.67 | 20.83 | 8.33 | 20.83 | 20.83 |
| | | OOD | 59.33 | 59.33 | 59.33 | 59.37 | 81.33 | 81.33 | 81.33 | 81.33 |
| | 10 | ID | 20.83 | 4.17 | 41.67 | 16.67 | 20.83 | 8.33 | 20.83 | 20.83 |
| | | OOD | 59.29 | 59.27 | 59.16 | 59.26 | 81.33 | 81.33 | 81.33 | 81.33 |
| | 100 | ID | 20.83 | 4.17 | 41.67 | 16.67 | 20.83 | 8.33 | 20.83 | 20.83 |
| | | OOD | 59.12 | 59.09 | 59.46 | 59.08 | 81.37 | 81.37 | 81.35 | 81.39 |
| | 1000 | ID | 12.5 | 4.17 | 45.83 | 16.67 | 20.83 | 8.33 | 25.0 | 20.33 |
| | | OOD | 58.99 | 58.94 | 59.31 | 58.87 | 81.32 | 81.37 | 81.44 | 81.32 |
| | 10000 | ID | 12.5 | 4.17 | 45.83 | 16.67 | 20.83 | 8.33 | 25.0 | 20.33 |
| | | OOD | 58.99 | 58.94 | 59.31 | 58.87 | 81.32 | 81.37 | 81.44 | 81.32 |

| Branch | Train Size | Test Set | qwen3-1.7b | | | | qwen3-32b | | | |
|---|---|---|---|---|---|---|---|---|---|---|
| | | | Biology | History | Economic | Physics | Biology | History | Economic | Physics |
| Intermediate | 1 | ID | 16.67 | 16.67 | 33.33 | 16.67 | 16.67 | 0.0 | 16.67 | 12.5 |
| | | OOD | 53.9 | 53.92 | 53.92 | 53.93 | 75.13 | 75.13 | 75.13 | 75.13 |
| | 10 | ID | 16.67 | 16.67 | 33.33 | 16.67 | 16.67 | 0.0 | 16.67 | 12.5 |
| | | OOD | 53.92 | 53.97 | 54.09 | 53.95 | 75.13 | 75.13 | 75.13 | 75.13 |
| | 100 | ID | 8.33 | 16.67 | 20.83 | 8.33 | 16.67 | 0.0 | 16.67 | 12.5 |
| | | OOD | 53.25 | 53.51 | 54.42 | 53.25 | 75.07 | 75.07 | 75.14 | 75.07 |
| | 1000 | ID | 25.0 | 20.83 | 25.0 | 25.0 | 16.67 | 0.0 | 25.0 | 12.5 |
| | | OOD | 52.66 | 52.36 | 53.6 | 52.64 | 75.21 | 75.07 | 75.26 | 74.98 |
| | 10000 | ID | 25.0 | 20.83 | 25.0 | 25.0 | 16.67 | 0.0 | 25.0 | 12.5 |
| | | OOD | 52.66 | 52.36 | 53.6 | 53.64 | 75.21 | 75.07 | 75.26 | 74.98 |
| Root | 1 | ID | 12.5 | 16.67 | 33.33 | 15.33 | 16.67 | 0.0 | 16.67 | 12.5 |
| | | OOD | 53.92 | 53.92 | 53.92 | 53.92 | 75.13 | 75.13 | 75.13 | 75.13 |
| | 10 | ID | 16.67 | 16.67 | 33.33 | 16.67 | 16.67 | 0.0 | 16.67 | 12.5 |
| | | OOD | 53.99 | 53.85 | 53.83 | 53.89 | 75.13 | 75.13 | 75.13 | 75.13 |
| | 100 | ID | 8.33 | 16.67 | 20.83 | 16.67 | 16.67 | 0.0 | 16.67 | 12.5 |
| | | OOD | 53.55 | 53.0 | 54.01 | 53.35 | 75.18 | 75.16 | 75.12 | 75.15 |
| | 1000 | ID | 33.33 | 29.17 | 37.5 | 29.33 | 16.67 | 0.0 | 25.0 | 12.5 |
| | | OOD | 52.67 | 51.83 | 53.65 | 53.67 | 75.33 | 75.16 | 75.05 | 75.23 |
| | 10000 | ID | 33.33 | 29.17 | 37.5 | 33.33 | 16.67 | 0.0 | 25.0 | 12.5 |
| | | OOD | 52.67 | 51.83 | 53.65 | 52.67 | 75.33 | 75.16 | 75.05 | 75.23 |
| Leaf | 1 | ID | 16.67 | 16.67 | 33.33 | 16.67 | 16.67 | 0.0 | 16.67 | 12.5 |
| | | OOD | 53.9 | 53.9 | 53.92 | 53.9 | 75.13 | 75.13 | 75.13 | 75.09 |
| | 10 | ID | 16.67 | 16.67 | 33.33 | 16.67 | 16.67 | 0.0 | 16.67 | 12.5 |
| | | OOD | 54.0 | 54.02 | 54.0 | 54.05 | 75.13 | 75.13 | 75.13 | 75.13 |
| | 100 | ID | 16.67 | 16.67 | 25.0 | 16.67 | 16.67 | 0.0 | 20.83 | 12.5 |
| | | OOD | 53.82 | 53.68 | 54.56 | 53.88 | 75.07 | 75.1 | 75.05 | 75.14 |
| | 1000 | ID | 25.0 | 29.17 | 25.0 | 25.0 | 16.67 | 0.0 | 25.0 | 12.5 |
| | | OOD | 52.81 | 53.16 | 53.46 | 53.81 | 75.11 | 75.1 | 75.25 | 74.71 |
| | 10000 | ID | 25.0 | 29.17 | 25.0 | 25.0 | 16.67 | 0.0 | 25.0 | 12.5 |
| | | OOD | 52.81 | 53.16 | 53.46 | 53.73 | 75.11 | 75.1 | 75.25 | 74.71 |

| Branch | Train Size | Test Set | qwen3-14b | | | | qwq-32b | | | |
|---|---|---|---|---|---|---|---|---|---|---|
| | | | Biology | History | Economic | Physics | Biology | History | Economic | Physics |
| Intermediate | 1 | ID | 20.83 | 4.17 | 25.0 | 12.5 | 12.5 | 0.0 | 29.17 | 12.5 |
| | | OOD | 73.86 | 73.86 | 73.86 | 73.86 | 77.42 | 77.42 | 77.42 | 77.42 |
| | 10 | ID | 20.83 | 4.17 | 25.0 | 12.5 | 12.5 | 0.0 | 29.17 | 12.5 |
| | | OOD | 73.84 | 73.86 | 73.86 | 73.86 | 77.38 | 77.45 | 77.44 | 77.4 |
| | 100 | ID | 20.83 | 0.0 | 25.0 | 12.5 | 12.5 | 0.0 | 29.17 | 12.5 |
| | | OOD | 73.61 | 73.63 | 73.89 | 73.94 | 77.4 | 77.37 | 77.28 | 77.37 |
| | 1000 | ID | 20.83 | 4.17 | 20.83 | 16.67 | 12.5 | 0.0 | 29.17 | 12.5 |
| | | OOD | 73.15 | 73.23 | 73.83 | 73.59 | 77.35 | 77.37 | 77.27 | 77.42 |
| | 10000 | ID | 20.83 | 4.17 | 20.83 | 16.67 | 12.5 | 0.0 | 29.17 | 12.5 |
| | | OOD | 73.15 | 73.23 | 73.83 | 73.59 | 77.35 | 77.37 | 77.27 | 77.42 |
| Root | 1 | ID | 20.83 | 4.17 | 25.0 | 12.5 | 12.5 | 0.0 | 29.17 | 12.5 |
| | | OOD | 73.86 | 73.86 | 73.86 | 73.86 | 77.42 | 77.42 | 77.42 | 77.42 |
| | 10 | ID | 20.83 | 4.17 | 25.0 | 12.5 | 12.5 | 0.0 | 29.17 | 12.5 |
| | | OOD | 73.88 | 73.86 | 73.86 | 73.85 | 77.45 | 77.47 | 77.44 | 77.48 |
| | 100 | ID | 20.83 | 0.0 | 25.0 | 12.5 | 12.5 | 0.0 | 29.17 | 12.5 |
| | | OOD | 73.84 | 73.66 | 73.86 | 73.62 | 77.35 | 77.3 | 77.38 | 77.45 |
| | 1000 | ID | 20.83 | 4.17 | 25.0 | 12.5 | 12.5 | 0.0 | 29.17 | 12.5 |
| | | OOD | 73.54 | 73.48 | 73.5 | 73.26 | 77.55 | 77.3 | 77.38 | 77.55 |
| | 10000 | ID | 20.83 | 4.17 | 25.0 | 12.5 | 12.5 | 0.0 | 29.17 | 12.5 |
| | | OOD | 73.54 | 73.48 | 73.5 | 73.26 | 77.55 | 77.3 | 77.38 | 77.55 |
| Leaf | 1 | ID | 20.83 | 4.17 | 25.0 | 12.5 | 12.5 | 0.0 | 29.17 | 12.5 |
| | | OOD | 73.86 | 73.86 | 73.86 | 73.86 | 77.42 | 77.42 | 77.42 | 77.42 |
| | 10 | ID | 20.83 | 4.17 | 25.0 | 12.5 | 12.5 | 0.0 | 29.17 | 12.5 |
| | | OOD | 73.86 | 73.86 | 73.86 | 73.86 | 77.42 | 77.47 | 77.43 | 77.45 |
| | 100 | ID | 20.83 | 0.0 | 25.0 | 12.5 | 12.5 | 0.0 | 29.17 | 12.5 |
| | | OOD | 73.64 | 73.91 | 73.84 | 73.83 | 77.39 | 77.4 | 77.38 | 77.26 |
| | 1000 | ID | 20.83 | 0.0 | 25.0 | 16.67 | 12.5 | 0.0 | 29.17 | 12.5 |
| | | OOD | 72.99 | 73.91 | 73.64 | 73.51 | 77.3 | 77.4 | 77.47 | 77.35 |
| | 10000 | ID | 20.83 | 0.0 | 25.0 | 16.67 | 12.5 | 0.0 | 29.17 | 12.5 |
| | | OOD | 72.99 | 73.91 | 73.64 | 73.51 | 77.3 | 77.4 | 77.47 | 77.35 |

| Branch | Train Size | Test Set | mistral-Small-24B-Instruct-2501 | | | | gemma-2b | | | |
|---|---|---|---|---|---|---|---|---|---|---|
| | | | Biology | History | Economic | Physics | Biology | History | Economic | Physics |
| Intermediate | 1 | ID | 16.67 | 50 | 12.5 | 20.83 | 8.33 | 4.17 | 12.5 | 12.5 |
| | | OOD | 73.4 | 73.34 | 73.24 | 73.39 | 30.46 | 30.63 | 30.49 | 30.53 |
| | 10 | ID | 50 | 45.83 | 16.67 | 0 | 20.83 | 12.5 | 29.17 | 4.17 |
| | | OOD | 24.47 | 22.99 | 25.2 | 25.51 | 29.4 | 30.37 | 29.06 | 30.29 |
| | 100 | ID | 29.17 | 45.83 | 54.17 | 58.33 | 16.67 | 45.83 | 37.5 | 29.17 |
| | | OOD | 24.22 | 22.95 | 24.16 | 23 | 25.81 | 24.08 | 26.16 | 26.54 |
| | 1000 | ID | 45.83 | 45.83 | 54.17 | 58.33 | 45.83 | 45.83 | 54.17 | 58.33 |
| | | OOD | 22.87 | 22.95 | 24.16 | 23 | 22.95 | 22.95 | 22.95 | 22.95 |
| | 10000 | ID | 45.83 | 45.83 | 54.17 | 58.33 | 45.83 | 45.83 | 54.17 | 58.33 |
| | | OOD | 22.87 | 22.95 | 24.16 | 23 | 22.95 | 22.95 | 22.95 | 22.95 |
| Root | 1 | ID | 33.33 | 20.83 | 16.67 | 45.83 | 4.17 | 0.0 | 16.67 | 4.17 |
| | | OOD | 73.42 | 73.42 | 73.16 | 73.39 | 30.54 | 30.59 | 30.64 | 30.54 |
| | 10 | ID | 45.83 | 54.17 | 4.17 | 58.33 | 8.33 | 8.33 | 12.5 | 33.33 |
| | | OOD | 22.95 | 25.2 | 25.27 | 23.14 | 30.34 | 27.18 | 30.55 | 25.79 |
| | 100 | ID | 45.83 | 37.5 | 54.17 | 58.33 | 4.17 | 50.0 | 25.0 | 54.17 |
| | | OOD | 22.83 | 24.4 | 23.32 | 22.99 | 29.3 | 24.51 | 29.13 | 23.74 |
| | 1000 | ID | 45.83 | 37.5 | 41.67 | 37.5 | 45.83 | 45.83 | 54.17 | 58.33 |
| | | OOD | 23.14 | 24.4 | 24.76 | 25.12 | 22.95 | 22.95 | 22.95 | 22.95 |
| | 10000 | ID | 45.83 | 37.5 | 41.67 | 37.5 | 45.83 | 45.83 | 54.17 | 58.33 |
| | | OOD | 23.14 | 24.4 | 24.76 | 25.12 | 22.95 | 22.95 | 22.95 | 22.95 |
| Leaf | 1 | ID | 50 | 41.67 | 83.33 | 4.17 | 8.33 | 33.33 | 25.0 | 0.0 |
| | | OOD | 73.32 | 73.24 | 73.14 | 73.42 | 30.25 | 28.98 | 30.64 | 30.44 |
| | 10 | ID | 4.17 | 45.83 | 4.17 | 41.67 | 12.5 | 33.33 | 20.83 | 4.17 |
| | | OOD | 25.47 | 22.95 | 25.54 | 25.19 | 27.7 | 24.6 | 25.28 | 29.08 |
| | 100 | ID | 8.33 | 45.83 | 54.17 | 8.33 | 25.0 | 37.5 | 37.5 | 41.67 |
| | | OOD | 26.63 | 22.96 | 22.95 | 24.61 | 24.94 | 24.19 | 24.9 | 24.07 |
| | 1000 | ID | 41.67 | 45.83 | 54.17 | 8.33 | 45.83 | 45.83 | 54.17 | 58.33 |
| | | OOD | 23.54 | 22.96 | 22.95 | 24.61 | 22.83 | 22.95 | 22.95 | 22.95 |
| | 10000 | ID | 41.67 | 45.83 | 54.17 | 8.33 | 45.83 | 45.83 | 54.17 | 58.33 |
| | | OOD | 23.54 | 22.96 | 22.95 | 24.61 | 22.83 | 22.95 | 22.95 | 22.95 |

| Branch | Train Size | Test Set | mistral-Large-Instruct-2411 | | | | gemma-7b | | | |
|---|---|---|---|---|---|---|---|---|---|---|
| | | | Biology | History | Economic | Physics | Biology | History | Economic | Physics |
| Intermediate | 1 | ID | 25.0 | 62.5 | 25.0 | 12.5 | 45.83 | 37.5 | 41.67 | 45.83 |
| | | OOD | 82.13 | 82.42 | 82.22 | 82.37 | 59.22 | 58.96 | 56.69 | 57.78 |
| | 10 | ID | 0.0 | 45.83 | 41.67 | 37.5 | 45.83 | 45.83 | 54.17 | 50.0 |
| | | OOD | 26.89 | 22.97 | 24.53 | 24.7 | 22.95 | 22.95 | 22.95 | 23.25 |
| | 100 | ID | 16.67 | 62.5 | 25.0 | 50.0 | 25.0 | 45.83 | 50.0 | 41.67 |
| | | OOD | 23.89 | 25.84 | 25.0 | 23.05 | 24.2 | 22.95 | 23.11 | 23.11 |
| | 1000 | ID | 16.67 | 62.5 | 25.0 | 50.0 | 29.17 | 54.17 | 8.33 | 66.67 |
| | | OOD | 23.89 | 25.84 | 25.0 | 23.05 | 24.9 | 25.59 | 25.22 | 24.68 |
| | 10000 | ID | 16.67 | 62.5 | 25.0 | 50.0 | 29.17 | 54.17 | 8.33 | 66.67 |
| | | OOD | 23.89 | 25.84 | 25.0 | 23.05 | 24.9 | 25.59 | 25.22 | 24.68 |
| Root | 1 | ID | 25.0 | 12.5 | 45.83 | 62.5 | 37.5 | 41.67 | 50.0 | 16.67 |
| | | OOD | 82.25 | 82.22 | 82.24 | 82.25 | 59.7 | 59.56 | 57.63 | 59.74 |
| | 10 | ID | 0.0 | 0.0 | 4.17 | 58.33 | 45.83 | 41.67 | 33.33 | 58.33 |
| | | OOD | 26.19 | 26.89 | 25.41 | 22.95 | 28.43 | 22.97 | 29.24 | 22.97 |
| | 100 | ID | 8.33 | 45.83 | 37.5 | 58.33 | 45.83 | 45.83 | 45.83 | 50.0 |
| | | OOD | 26.86 | 22.95 | 24.64 | 23.0 | 22.95 | 23.07 | 22.95 | 24.13 |
| | 1000 | ID | 8.33 | 45.83 | 37.5 | 58.33 | 33.33 | 45.83 | 20.83 | 54.17 |
| | | OOD | 26.86 | 22.95 | 24.64 | 23.0 | 23.98 | 23.07 | 23.24 | 23.34 |
| | 10000 | ID | 8.33 | 45.83 | 37.5 | 58.33 | 33.33 | 45.83 | 20.83 | 54.17 |
| | | OOD | 26.86 | 22.95 | 24.64 | 23.0 | 23.98 | 23.07 | 23.24 | 23.34 |
| Leaf | 1 | ID | 54.17 | 29.17 | 41.67 | 37.5 | 45.83 | 45.83 | 54.17 | 33.33 |
| | | OOD | 82.19 | 82.25 | 82.07 | 82.08 | 22.82 | 22.97 | 22.95 | 59.29 |
| | 10 | ID | 4.17 | 0.0 | 54.17 | 58.33 | 45.83 | 45.83 | 45.83 | 58.33 |
| | | OOD | 25.47 | 25.51 | 22.95 | 23.07 | 22.95 | 23.07 | 23.33 | 22.87 |
| | 100 | ID | 50.0 | 0.0 | 45.83 | 54.17 | 37.5 | 41.67 | 41.67 | 58.33 |
| | | OOD | 23.05 | 24.6 | 24.69 | 25.55 | 23.96 | 23.78 | 23.38 | 22.94 |
| | 1000 | ID | 50.0 | 0.0 | 45.83 | 54.17 | 4.17 | 54.17 | 4.17 | 4.17 |
| | | OOD | 23.05 | 24.6 | 24.69 | 25.55 | 25.48 | 24.49 | 25.52 | 25.54 |
| | 10000 | ID | 50.0 | 0.0 | 45.83 | 54.17 | 4.17 | 54.17 | 4.17 | 4.17 |
| | | OOD | 23.05 | 24.6 | 24.69 | 25.55 | 25.48 | 24.49 | 25.52 | 25.54 |

| | | | DeepSeek-R1-0528-Qwen3-8B | | | |
|---|---|---|---|---|---|---|
| Branch | Train Size | Test Set | Biology | History | Economic | Physics |
| Intermediate | 1 | ID | 12.5 | 0.0 | 12.5 | 25.0 |
| | | OOD | 65.99 | 65.99 | 65.99 | 65.96 |
| | 10 | ID | 12.5 | 0.0 | 12.5 | 25.0 |
| | | OOD | 65.96 | 65.93 | 66.0 | 65.95 |
| | 100 | ID | 12.5 | 0.0 | 12.5 | 25.0 |
| | | OOD | 65.85 | 65.7 | 66.07 | 65.84 |
| | 1000 | ID | 12.5 | 0.0 | 16.67 | 25.0 |
| | | OOD | 66.39 | 65.7 | 66.24 | 65.89 |
| | 10000 | ID | 12.5 | 0.0 | 16.67 | 25.0 |
| | | OOD | 66.39 | 65.7 | 66.24 | 65.89 |
| Root | 1 | ID | 12.5 | 0.0 | 12.5 | 25.0 |
| | | OOD | 65.99 | 65.99 | 65.99 | 65.96 |
| | 10 | ID | 12.5 | 0.0 | 12.5 | 25.0 |
| | | OOD | 65.97 | 65.98 | 65.99 | 65.99 |
| | 100 | ID | 12.5 | 0.0 | 12.5 | 25.0 |
| | | OOD | 65.82 | 65.95 | 65.97 | 65.73 |
| | 1000 | ID | 12.5 | 0.0 | 16.67 | 25.0 |
| | | OOD | 66.14 | 65.95 | 66.13 | 66.16 |
| | 10000 | ID | 12.5 | 0.0 | 16.67 | 25.0 |
| | | OOD | 66.14 | 65.95 | 66.13 | 66.16 |
| Leaf | 1 | ID | 12.5 | 0.0 | 12.5 | 25.0 |
| | | OOD | 65.99 | 65.99 | 65.99 | 65.96 |
| | 10 | ID | 12.5 | 0.0 | 12.5 | 25.0 |
| | | OOD | 65.99 | 66.04 | 66.0 | 65.92 |
| | 100 | ID | 12.5 | 0.0 | 12.5 | 25.0 |
| | | OOD | 65.66 | 65.78 | 66.07 | 65.9 |
| | 1000 | ID | 12.5 | 0.0 | 16.67 | 25.0 |
| | | OOD | 65.68 | 65.78 | 66.03 | 66.01 |
| | 10000 | ID | 12.5 | 0.0 | 16.67 | 25.0 |
| | | OOD | 65.68 | 65.78 | 66.03 | 66.01 |

Table 14: Normalized model similarity scores for Llama3

| Subject | Branch | Train Size | Editing | | | | Unlearning | | | |
|---|---|---|---|---|---|---|---|---|---|---|
| | | | CKA | Fisher | KL | L2 | CKA | Fisher | KL | L2 |
| biology | root | 1 | 0.999 | 0.000 | 0.017 | 0.006 | 1.000 | 0.000 | 0.000 | 0.000 |
| | | 10 | 0.999 | 0.150 | 0.129 | 0.137 | 0.993 | 0.392 | 0.368 | 0.304 |
| | | 100 | 0.999 | 0.150 | 0.129 | 0.137 | 0.001 | 0.949 | 0.984 | 0.741 |
| | | 1000 | 0.999 | 0.150 | 0.129 | 0.137 | 0.623 | 0.895 | 0.811 | 0.770 |
| | | 10000 | 0.999 | 0.150 | 0.129 | 0.137 | 0.825 | 0.958 | 0.874 | 0.977 |
| | intermediate | 1 | 0.999 | 0.012 | 0.080 | 0.014 | 1.000 | 0.000 | 0.000 | 0.000 |
| | | 10 | 0.999 | 0.087 | 0.119 | 0.091 | 0.994 | 0.390 | 0.355 | 0.302 |
| | | 100 | 0.999 | 0.147 | 0.142 | 0.142 | 0.277 | 0.919 | 0.944 | 0.727 |
| | | 1000 | 0.999 | 0.147 | 0.142 | 0.142 | 0.612 | 0.921 | 0.909 | 0.795 |
| | | 10000 | 0.999 | 0.147 | 0.142 | 0.142 | 0.784 | 0.988 | 0.896 | 0.982 |
| | leaf | 1 | 0.999 | 0.010 | 0.079 | 0.017 | 1.000 | 0.000 | 0.000 | 0.000 |
| | | 10 | 0.999 | 0.216 | 0.263 | 0.229 | 0.994 | 0.371 | 0.358 | 0.285 |
| | | 100 | 0.999 | 0.479 | 0.695 | 0.494 | 0.277 | 0.917 | 0.948 | 0.729 |
| | | 1000 | 0.379 | 0.982 | 0.997 | 0.992 | 0.687 | 0.877 | 0.920 | 0.803 |
| | | 10000 | 0.379 | 0.982 | 0.997 | 0.992 | 0.909 | 0.903 | 0.876 | 0.989 |
| economics | root | 1 | 0.999 | 0.018 | 0.087 | 0.003 | 1.000 | 0.000 | 0.000 | 0.000 |
| | | 10 | 0.999 | 0.018 | 0.087 | 0.003 | 0.996 | 0.388 | 0.329 | 0.306 |
| | | 100 | 0.999 | 0.018 | 0.087 | 0.003 | 0.000 | 0.948 | 0.983 | 0.741 |
| | | 1000 | 0.999 | 0.018 | 0.087 | 0.003 | 0.623 | 0.893 | 0.805 | 0.767 |
| | | 10000 | 0.999 | 0.018 | 0.087 | 0.003 | 0.801 | 1.000 | 0.820 | 0.921 |
| | intermediate | 1 | 0.999 | 0.020 | 0.052 | 0.004 | 1.000 | 0.000 | 0.000 | 0.000 |
| | | 10 | 0.999 | 0.194 | 0.159 | 0.199 | 0.997 | 0.378 | 0.334 | 0.298 |
| | | 100 | 0.999 | 0.316 | 0.269 | 0.322 | 0.502 | 0.940 | 0.950 | 0.747 |
| | | 1000 | 0.999 | 0.316 | 0.269 | 0.322 | 0.686 | 0.895 | 0.828 | 0.788 |
| | | 10000 | 0.999 | 0.316 | 0.269 | 0.322 | 0.807 | 0.909 | 0.825 | 0.966 |
| | leaf | 1 | 0.999 | 0.022 | 0.048 | 0.018 | 1.000 | 0.000 | 0.000 | 0.000 |
| | | 10 | 0.999 | 0.323 | 0.338 | 0.326 | 0.998 | 0.366 | 0.284 | 0.289 |
| | | 100 | 0.999 | 0.363 | 0.362 | 0.371 | 0.394 | 0.912 | 0.915 | 0.724 |
| | | 1000 | 0.999 | 0.371 | 0.365 | 0.380 | 0.666 | 0.891 | 0.818 | 0.787 |
| | | 10000 | 0.999 | 0.371 | 0.365 | 0.380 | 0.851 | 0.891 | 0.815 | 0.942 |
| history | root | 1 | 0.999 | 0.006 | 0.038 | 0.003 | 1.000 | 0.000 | 0.000 | 0.000 |
| | | 10 | 0.999 | 0.124 | 0.152 | 0.134 | 0.994 | 0.390 | 0.365 | 0.298 |
| | | 100 | 0.999 | 0.124 | 0.152 | 0.134 | 0.282 | 0.950 | 0.954 | 0.730 |
| | | 1000 | 0.999 | 0.124 | 0.152 | 0.134 | 0.687 | 0.917 | 0.920 | 0.774 |
| | | 10000 | 0.999 | 0.124 | 0.152 | 0.134 | 0.895 | 0.957 | 0.878 | 0.981 |
| | intermediate | 1 | 0.999 | 0.011 | 0.067 | 0.004 | 1.000 | 0.000 | 0.000 | 0.000 |
| | | 10 | 0.999 | 0.138 | 0.161 | 0.154 | 0.995 | 0.391 | 0.354 | 0.300 |
| | | 100 | 0.999 | 0.138 | 0.161 | 0.154 | 0.230 | 0.925 | 0.970 | 0.723 |
| | | 1000 | 0.999 | 0.138 | 0.161 | 0.154 | 0.738 | 0.868 | 0.850 | 0.770 |
| | | 10000 | 0.999 | 0.138 | 0.161 | 0.154 | 0.888 | 0.991 | 0.842 | 0.973 |
| | leaf | 1 | 1 | 0.001 | 0.070 | 0.000 | 1.000 | 0.000 | 0.000 | 0.000 |
| | | 10 | 0.999 | 0.217 | 0.235 | 0.230 | 0.994 | 0.361 | 0.370 | 0.269 |
| | | 100 | 0.999 | 0.439 | 0.523 | 0.454 | 0.243 | 0.919 | 1.000 | 0.722 |
| | | 1000 | 0.232 | 0.963 | 0.985 | 0.965 | 0.673 | 0.900 | 0.988 | 0.805 |
| | | 10000 | 0.232 | 0.963 | 0.985 | 0.965 | 0.895 | 0.899 | 0.942 | 1.000 |
| physics | root | 1 | 0.999 | 0.011 | 0.000 | 0.010 | 1.000 | 0.000 | 0.000 | 0.000 |
| | | 10 | 0.999 | 0.158 | 0.127 | 0.162 | 0.994 | 0.396 | 0.359 | 0.309 |
| | | 100 | 0.999 | 0.158 | 0.127 | 0.162 | 0.437 | 0.920 | 0.940 | 0.723 |
| | | 1000 | 0.999 | 0.158 | 0.127 | 0.162 | 0.749 | 0.896 | 0.892 | 0.774 |
| | | 10000 | 0.999 | 0.158 | 0.127 | 0.162 | 0.909 | 0.932 | 0.847 | 0.978 |
| | intermediate | 1 | 0.999 | 0.008 | 0.027 | 0.007 | 1.000 | 0.000 | 0.000 | 0.000 |
| | | 10 | 0.999 | 0.214 | 0.192 | 0.223 | 0.997 | 0.381 | 0.337 | 0.300 |
| | | 100 | 0.425 | 0.751 | 1.000 | 0.755 | 0.469 | 0.945 | 0.980 | 0.745 |
| | | 1000 | 0 | 1.000 | 0.990 | 1.000 | 0.701 | 0.911 | 0.869 | 0.801 |
| | | 10000 | 0 | 1.000 | 0.990 | 1.000 | 0.910 | 0.904 | 0.831 | 0.910 |
| | leaf | 1 | 0.999 | 0.023 | 0.043 | 0.020 | 1.000 | 0.000 | 0.000 | 0.000 |
| | | 10 | 0.999 | 0.249 | 0.240 | 0.255 | 0.994 | 0.379 | 0.360 | 0.283 |
| | | 100 | 0.45 | 0.743 | 0.987 | 0.741 | 0.959 | 0.497 | 0.483 | 0.504 |
| | | 1000 | 0.154 | 0.976 | 0.985 | 0.985 | 0.934 | 0.710 | 0.692 | 0.748 |
| | | 10000 | 0.154 | 0.976 | 0.985 | 0.985 | 0.907 | 0.899 | 0.879 | 0.986 |

Table 15: Normalized model similarity scores for DeepSeek

| Subject | Branch | Train Size | Editing CKA | Fisher | KL | L2 | Unlearning CKA | Fisher | KL | L2 |
|---------|--------|-----------|-----|--------|-----|-----|-----|--------|-----|-----|
| biology | root | 1 | 1.000 | 0.150 | 0.199 | 0.054 | 1.000 | 0.000 | 0.000 | 0.000 |
| | | 10 | 0.998 | 0.315 | 0.247 | 0.238 | 0.975 | 0.302 | 0.261 | 0.276 |
| | | 100 | 0.997 | 0.366 | 0.417 | 0.275 | 0.457 | 0.739 | 0.895 | 0.745 |
| | | 1000 | 0.997 | 0.366 | 0.426 | 0.275 | 0.481 | 0.752 | 0.914 | 0.827 |
| | | 10000 | 0.997 | 0.366 | 0.426 | 0.275 | 0.674 | 0.777 | 0.713 | 0.995 |
| | intermediate | 1 | 0.999 | 0.144 | 0.115 | 0.061 | 1.000 | 0.000 | 0.000 | 0.000 |
| | | 10 | 0.996 | 0.405 | 0.389 | 0.271 | 0.988 | 0.292 | 0.298 | 0.271 |
| | | 100 | 0.994 | 0.461 | 0.468 | 0.352 | 0.649 | 0.733 | 0.693 | 0.747 |
| | | 1000 | 0.994 | 0.461 | 0.470 | 0.352 | 0.579 | 0.781 | 0.769 | 0.829 |
| | | 10000 | 0.994 | 0.461 | 0.470 | 0.352 | 0.727 | 0.775 | 0.792 | 0.916 |
| | leaf | 1 | 0.999 | 0.131 | 0.011 | 0.033 | 1.000 | 0.000 | 0.000 | 0.000 |
| | | 10 | 0.991 | 0.523 | 0.682 | 0.385 | 0.986 | 0.250 | 0.501 | 0.248 |
| | | 100 | 0.966 | 0.738 | 0.828 | 0.608 | 0.772 | 0.635 | 0.824 | 0.731 |
| | | 1000 | 0.960 | 0.758 | 0.877 | 0.638 | 0.641 | 0.708 | 0.777 | 0.856 |
| | | 10000 | 0.960 | 0.758 | 0.877 | 0.638 | 0.781 | 0.764 | 0.756 | 0.972 |
| economics | root | 1 | 0.995 | 0.000 | 0.000 | 0.000 | 1.000 | 0.000 | 0.000 | 0.000 |
| | | 10 | 0.993 | 0.219 | 0.186 | 0.194 | 0.968 | 0.320 | 0.334 | 0.276 |
| | | 100 | 0.992 | 0.261 | 0.216 | 0.246 | 0.543 | 0.733 | 0.645 | 0.744 |
| | | 1000 | 0.992 | 0.261 | 0.221 | 0.246 | 0.408 | 0.819 | 0.730 | 0.819 |
| | | 10000 | 0.992 | 0.261 | 0.221 | 0.246 | 0.000 | 0.980 | 1.000 | 0.880 |
| | intermediate | 1 | 0.992 | 0.102 | 0.250 | 0.053 | 1.000 | 0.000 | 0.000 | 0.000 |
| | | 10 | 0.976 | 0.480 | 0.615 | 0.387 | 0.988 | 0.308 | 0.264 | 0.265 |
| | | 100 | 0.972 | 0.500 | 0.634 | 0.412 | 0.675 | 0.746 | 0.857 | 0.754 |
| | | 1000 | 0.972 | 0.500 | 0.634 | 0.412 | 0.646 | 0.787 | 0.877 | 0.836 |
| | | 10000 | 0.972 | 0.500 | 0.634 | 0.412 | 0.645 | 0.869 | 0.919 | 0.917 |
| | leaf | 1 | 0.996 | 0.028 | 0.045 | 0.030 | 1.000 | 0.000 | 0.000 | 0.000 |
| | | 10 | 0.982 | 0.378 | 0.399 | 0.320 | 0.976 | 0.264 | 0.500 | 0.251 |
| | | 100 | 0.953 | 0.546 | 0.561 | 0.505 | 0.716 | 0.711 | 0.910 | 0.733 |
| | | 1000 | 0.000 | 1.000 | 1.000 | 1.000 | 0.582 | 0.744 | 0.805 | 0.847 |
| | | 10000 | 0.000 | 1.000 | 1.000 | 1.000 | 0.460 | 1.000 | 0.890 | 0.897 |
| history | root | 1 | 0.999 | 0.164 | 0.199 | 0.073 | 1.000 | 0.000 | 0.000 | 0.000 |
| | | 10 | 0.997 | 0.299 | 0.394 | 0.197 | 0.980 | 0.315 | 0.370 | 0.276 |
| | | 100 | 0.993 | 0.427 | 0.682 | 0.357 | 0.594 | 0.692 | 0.659 | 0.734 |
| | | 1000 | 0.993 | 0.427 | 0.679 | 0.357 | 0.686 | 0.731 | 0.898 | 0.814 |
| | | 10000 | 0.993 | 0.427 | 0.679 | 0.357 | 0.550 | 0.798 | 0.965 | 1.000 |
| | intermediate | 1 | 0.998 | 0.127 | 0.085 | 0.054 | 1.000 | 0.000 | 0.000 | 0.000 |
| | | 10 | 0.998 | 0.224 | 0.138 | 0.158 | 0.979 | 0.309 | 0.575 | 0.273 |
| | | 100 | 0.998 | 0.247 | 0.164 | 0.195 | 0.418 | 0.730 | 0.969 | 0.741 |
| | | 1000 | 0.998 | 0.247 | 0.168 | 0.195 | 0.732 | 0.724 | 0.762 | 0.811 |
| | | 10000 | 0.998 | 0.247 | 0.168 | 0.195 | 0.702 | 0.775 | 0.949 | 0.992 |
| | leaf | 1 | 0.999 | 0.146 | 0.101 | 0.075 | 1.000 | 0.000 | 0.000 | 0.000 |
| | | 10 | 0.987 | 0.515 | 0.620 | 0.395 | 0.983 | 0.267 | 0.484 | 0.232 |
| | | 100 | 0.971 | 0.644 | 0.691 | 0.571 | 0.698 | 0.671 | 0.835 | 0.722 |
| | | 1000 | 0.968 | 0.658 | 0.699 | 0.590 | 0.106 | 0.739 | 0.833 | 0.868 |
| | | 10000 | 0.968 | 0.658 | 0.699 | 0.590 | 0.610 | 0.743 | 0.853 | 0.991 |
| physics | root | 1 | 0.999 | 0.182 | 0.087 | 0.076 | 1.000 | 0.000 | 0.000 | 0.000 |
| | | 10 | 0.998 | 0.287 | 0.231 | 0.157 | 0.980 | 0.300 | 0.320 | 0.250 |
| | | 100 | 0.998 | 0.287 | 0.232 | 0.157 | 0.600 | 0.700 | 0.800 | 0.750 |
| | | 1000 | 0.998 | 0.287 | 0.239 | 0.157 | 0.650 | 0.740 | 0.850 | 0.820 |
| | | 10000 | 0.998 | 0.287 | 0.239 | 0.157 | 0.550 | 0.800 | 0.950 | 0.950 |
| | intermediate | 1 | 0.999 | 0.172 | 0.174 | 0.031 | 1.000 | 0.000 | 0.000 | 0.000 |
| | | 10 | 0.986 | 0.451 | 0.480 | 0.323 | 0.970 | 0.280 | 0.300 | 0.250 |
| | | 100 | 0.984 | 0.498 | 0.485 | 0.368 | 0.680 | 0.720 | 0.780 | 0.760 |
| | | 1000 | 0.983 | 0.498 | 0.482 | 0.368 | 0.630 | 0.750 | 0.820 | 0.830 |
| | | 10000 | 0.983 | 0.498 | 0.482 | 0.368 | 0.600 | 0.770 | 0.850 | 0.900 |
| | leaf | 1 | 0.998 | 0.221 | 0.144 | 0.076 | 1.000 | 0.000 | 0.000 | 0.000 |
| | | 10 | 0.989 | 0.447 | 0.441 | 0.360 | 0.980 | 0.250 | 0.450 | 0.250 |
| | | 100 | 0.969 | 0.604 | 0.669 | 0.542 | 0.700 | 0.670 | 0.820 | 0.720 |
| | | 1000 | 0.965 | 0.625 | 0.660 | 0.567 | 0.650 | 0.720 | 0.850 | 0.850 |
| | | 10000 | 0.965 | 0.625 | 0.660 | 0.567 | 0.600 | 0.740 | 0.870 | 0.920 |

Table 16: Normalized model similarity scores for Qwen3

| Subject | Branch | Train Size | Editing | | | | Unlearning | | | |
|---|---|---|---|---|---|---|---|---|---|---|
| | | | CKA | Fisher | KL | L2 | CKA | Fisher | KL | L2 |
| biology | root | 1 | 0.999 | 0.102 | 0.060 | 0.085 | 1.000 | 0.000 | 0.000 | 0.000 |
| | | 10 | 0.996 | 0.461 | 0.331 | 0.424 | 0.995 | 0.347 | 0.280 | 0.278 |
| | | 100 | 0.996 | 0.482 | 0.341 | 0.442 | 0.773 | 0.882 | 0.744 | 0.745 |
| | | 1000 | 0.996 | 0.482 | 0.341 | 0.442 | 0.626 | 0.941 | 0.905 | 0.861 |
| | | 10000 | 0.996 | 0.482 | 0.341 | 0.442 | 0.836 | 0.999 | 0.931 | 0.990 |
| | intermediate | 1 | 0.999 | 0.057 | 0.015 | 0.052 | 1.000 | 0.000 | 0.000 | 0.000 |
| | | 10 | 0.999 | 0.313 | 0.171 | 0.274 | 0.992 | 0.340 | 0.285 | 0.276 |
| | | 100 | 0.999 | 0.379 | 0.227 | 0.311 | 0.546 | 0.910 | 0.774 | 0.766 |
| | | 1000 | 0.999 | 0.379 | 0.227 | 0.311 | 0.304 | 0.949 | 0.854 | 0.864 |
| | | 10000 | 0.999 | 0.379 | 0.227 | 0.311 | 0.742 | 0.934 | 0.866 | 0.933 |
| | leaf | 1 | 0.999 | 0.092 | 0.061 | 0.111 | 1.000 | 0.000 | 0.000 | 0.000 |
| | | 10 | 0.998 | 0.422 | 0.308 | 0.419 | 0.994 | 0.330 | 0.290 | 0.256 |
| | | 100 | 0.996 | 0.527 | 0.364 | 0.516 | 0.561 | 0.900 | 0.792 | 0.755 |
| | | 1000 | 0.843 | 0.888 | 0.759 | 0.889 | 0.386 | 0.962 | 0.895 | 0.883 |
| | | 10000 | 0.843 | 0.888 | 0.759 | 0.889 | 0.719 | 0.982 | 0.902 | 0.986 |
| economics | root | 1 | 0.999 | 0.002 | 0.072 | 0.000 | 1.000 | 0.000 | 0.000 | 0.000 |
| | | 10 | 0.998 | 0.292 | 0.236 | 0.268 | 0.995 | 0.342 | 0.280 | 0.283 |
| | | 100 | 0.998 | 0.311 | 0.245 | 0.290 | 0.754 | 0.894 | 0.743 | 0.755 |
| | | 1000 | 0.998 | 0.311 | 0.246 | 0.290 | 0.542 | 0.924 | 0.796 | 0.840 |
| | | 10000 | 0.998 | 0.311 | 0.246 | 0.290 | 0.667 | 0.946 | 0.820 | 0.936 |
| | intermediate | 1 | 0.999 | 0.048 | 0.101 | 0.019 | 1.000 | 0.000 | 0.000 | 0.000 |
| | | 10 | 0.996 | 0.428 | 0.311 | 0.389 | 0.993 | 0.341 | 0.290 | 0.267 |
| | | 100 | 0.996 | 0.454 | 0.326 | 0.417 | 0.720 | 0.904 | 0.785 | 0.772 |
| | | 1000 | 0.996 | 0.454 | 0.326 | 0.417 | 0.668 | 0.943 | 0.825 | 0.864 |
| | | 10000 | 0.996 | 0.454 | 0.326 | 0.417 | 0.741 | 0.945 | 0.844 | 0.946 |
| | leaf | 1 | 0.999 | 0.038 | 0.078 | 0.012 | 1.000 | 0.000 | 0.000 | 0.000 |
| | | 10 | 0.998 | 0.376 | 0.273 | 0.340 | 0.993 | 0.334 | 0.277 | 0.260 |
| | | 100 | 0.986 | 0.635 | 0.461 | 0.620 | 0.798 | 0.879 | 0.721 | 0.746 |
| | | 1000 | 0.000 | 1.000 | 1.000 | 1.000 | 0.677 | 0.960 | 0.858 | 0.871 |
| | | 10000 | 0.000 | 1.000 | 1.000 | 1.000 | 0.748 | 0.979 | 0.858 | 0.968 |
| history | root | 1 | 0.999 | 0.037 | 0.094 | 0.015 | 1.000 | 0.000 | 0.000 | 0.000 |
| | | 10 | 0.998 | 0.415 | 0.345 | 0.393 | 0.987 | 0.337 | 0.291 | 0.271 |
| | | 100 | 0.998 | 0.423 | 0.347 | 0.401 | 0.656 | 0.890 | 0.765 | 0.754 |
| | | 1000 | 0.998 | 0.423 | 0.347 | 0.401 | 0.608 | 0.900 | 0.773 | 0.832 |
| | | 10000 | 0.998 | 0.423 | 0.347 | 0.401 | 0.731 | 0.979 | 0.827 | 0.954 |
| | intermediate | 1 | 0.999 | 0.014 | 0.098 | 0.002 | 1.000 | 0.000 | 0.000 | 0.000 |
| | | 10 | 0.997 | 0.445 | 0.368 | 0.419 | 0.991 | 0.341 | 0.281 | 0.272 |
| | | 100 | 0.996 | 0.500 | 0.389 | 0.482 | 0.301 | 0.910 | 0.765 | 0.764 |
| | | 1000 | 0.996 | 0.500 | 0.390 | 0.482 | 0.000 | 0.924 | 0.797 | 0.841 |
| | | 10000 | 0.996 | 0.500 | 0.390 | 0.482 | 0.691 | 0.979 | 0.847 | 0.971 |
| | leaf | 1 | 1.000 | 0.041 | 0.122 | 0.049 | 1.000 | 0.000 | 0.000 | 0.000 |
| | | 10 | 0.999 | 0.401 | 0.325 | 0.406 | 0.987 | 0.323 | 0.299 | 0.238 |
| | | 100 | 0.997 | 0.510 | 0.386 | 0.515 | 0.655 | 0.886 | 0.819 | 0.732 |
| | | 1000 | 0.997 | 0.510 | 0.386 | 0.515 | 0.036 | 0.994 | 1.000 | 0.897 |
| | | 10000 | 0.997 | 0.510 | 0.386 | 0.515 | 0.520 | 0.982 | 0.980 | 1.000 |
| physics | root | 1 | 0.997 | 0.083 | 0.045 | 0.063 | 1.000 | 0.000 | 0.000 | 0.000 |
| | | 10 | 0.997 | 0.407 | 0.280 | 0.363 | 0.987 | 0.335 | 0.290 | 0.278 |
| | | 100 | 0.996 | 0.462 | 0.329 | 0.428 | 0.688 | 0.887 | 0.767 | 0.751 |
| | | 1000 | 0.996 | 0.462 | 0.329 | 0.428 | 0.517 | 0.921 | 0.858 | 0.848 |
| | | 10000 | 0.996 | 0.462 | 0.329 | 0.428 | 0.764 | 0.993 | 0.871 | 0.938 |
| | intermediate | 1 | 0.999 | 0.000 | 0.000 | 0.003 | 1.000 | 0.000 | 0.000 | 0.000 |
| | | 10 | 0.996 | 0.424 | 0.310 | 0.407 | 0.986 | 0.339 | 0.310 | 0.269 |
| | | 100 | 0.993 | 0.493 | 0.363 | 0.485 | 0.733 | 0.897 | 0.803 | 0.755 |
| | | 1000 | 0.993 | 0.493 | 0.363 | 0.485 | 0.583 | 0.975 | 0.853 | 0.872 |
| | | 10000 | 0.993 | 0.493 | 0.363 | 0.485 | 0.723 | 0.891 | 0.827 | 0.906 |
| | leaf | 1 | 0.999 | 0.051 | 0.053 | 0.048 | 1.000 | 0.000 | 0.000 | 0.000 |
| | | 10 | 0.996 | 0.402 | 0.325 | 0.375 | 0.995 | 0.335 | 0.289 | 0.259 |
| | | 100 | 0.992 | 0.509 | 0.384 | 0.493 | 0.740 | 0.892 | 0.787 | 0.739 |
| | | 1000 | 0.992 | 0.509 | 0.384 | 0.493 | 0.526 | 0.966 | 0.904 | 0.889 |
| | | 10000 | 0.992 | 0.509 | 0.384 | 0.493 | 0.642 | 1.000 | 0.853 | 0.984 |

Table 17: Normalized model similarity scores for QwQ

| Subject | Branch | Train Size | Editing | | | | Unlearning | | | |
|---|---|---|---|---|---|---|---|---|---|---|
| | | | CKA | Fisher | KL | L2 | CKA | Fisher | KL | L2 |
| biology | root | 1 | 0.741 | 0.208 | 0.000 | 0.000 | 0.493 | 0.000 | 0.000 | 0.000 |
| | | 10 | 0.736 | 0.550 | 0.492 | 0.414 | 0.440 | 0.321 | 0.348 | 0.286 |
| | | 100 | 0.997 | 0.613 | 0.521 | 0.478 | 0.490 | 0.799 | 0.826 | 0.737 |
| | | 1000 | 0.760 | 0.613 | 0.521 | 0.478 | 0.433 | 0.788 | 0.887 | 0.803 |
| | | 10000 | 0.760 | 0.613 | 0.521 | 0.478 | 0.491 | 0.934 | 0.977 | 0.989 |
| | intermediate | 1 | 0.766 | 0.111 | 0.048 | 0.022 | 0.493 | 0.000 | 0.000 | 0.000 |
| | | 10 | 1.000 | 0.543 | 0.433 | 0.408 | 0.440 | 0.308 | 0.356 | 0.275 |
| | | 100 | 0.561 | 0.547 | 0.434 | 0.415 | 0.489 | 0.790 | 0.850 | 0.744 |
| | | 1000 | 0.739 | 0.547 | 0.435 | 0.415 | 0.000 | 0.845 | 0.829 | 0.810 |
| | | 10000 | 0.739 | 0.547 | 0.435 | 0.415 | 0.437 | 0.889 | 0.887 | 0.943 |
| | leaf | 1 | 0.741 | 0.103 | 0.019 | 0.021 | 0.493 | 0.000 | 0.000 | 0.000 |
| | | 10 | 0.764 | 0.555 | 0.435 | 0.437 | 0.440 | 0.300 | 0.342 | 0.249 |
| | | 100 | 0.760 | 0.675 | 0.551 | 0.583 | 0.438 | 0.789 | 0.797 | 0.733 |
| | | 1000 | 0.976 | 0.803 | 0.699 | 0.758 | 0.050 | 0.886 | 0.964 | 0.846 |
| | | 10000 | 0.976 | 0.803 | 0.699 | 0.758 | 0.435 | 1.000 | 0.932 | 0.983 |
| economics | root | 1 | 0.741 | 0.063 | 0.097 | 0.066 | 0.493 | 0.000 | 0.000 | 0.000 |
| | | 10 | 0.765 | 0.303 | 0.305 | 0.274 | 0.493 | 0.310 | 0.348 | 0.297 |
| | | 100 | 0.765 | 0.329 | 0.320 | 0.292 | 0.490 | 0.810 | 0.823 | 0.733 |
| | | 1000 | 0.996 | 0.329 | 0.320 | 0.292 | 0.433 | 0.883 | 0.869 | 0.808 |
| | | 10000 | 0.996 | 0.329 | 0.320 | 0.292 | 0.487 | 0.844 | 0.865 | 0.844 |
| | intermediate | 1 | 0.741 | 0.079 | 0.089 | 0.100 | 0.440 | 0.000 | 0.000 | 0.000 |
| | | 10 | 0.763 | 0.403 | 0.391 | 0.370 | 0.440 | 0.301 | 0.351 | 0.266 |
| | | 100 | 0.735 | 0.532 | 0.502 | 0.500 | 0.490 | 0.797 | 0.895 | 0.753 |
| | | 1000 | 0.766 | 0.000 | 0.032 | 0.016 | 0.430 | 0.862 | 0.937 | 0.832 |
| | | 10000 | 0.738 | 0.427 | 0.402 | 0.389 | 0.435 | 0.878 | 0.914 | 0.917 |
| | leaf | 1 | 0.766 | 0.000 | 0.032 | 0.016 | 0.440 | 0.000 | 0.000 | 0.000 |
| | | 10 | 0.738 | 0.427 | 0.402 | 0.389 | 0.493 | 0.288 | 0.346 | 0.253 |
| | | 100 | 0.755 | 0.610 | 0.576 | 0.576 | 0.438 | 0.788 | 0.787 | 0.722 |
| | | 1000 | 0.000 | 1.000 | 1.000 | 1.000 | 0.487 | 0.886 | 0.962 | 0.834 |
| | | 10000 | 0.000 | 1.000 | 1.000 | 1.000 | 0.434 | 0.981 | 0.951 | 0.932 |
| history | root | 1 | 0.766 | 0.194 | 0.195 | 0.148 | 0.434 | 0.793 | 0.848 | 0.728 |
| | | 10 | 0.739 | 0.508 | 0.451 | 0.386 | 0.487 | 0.805 | 0.877 | 0.793 |
| | | 100 | 0.736 | 0.604 | 0.535 | 0.478 | 0.434 | 0.793 | 0.848 | 0.728 |
| | | 1000 | 0.761 | 0.604 | 0.535 | 0.478 | 0.487 | 0.805 | 0.877 | 0.793 |
| | | 10000 | 0.761 | 0.604 | 0.535 | 0.478 | 0.490 | 0.982 | 0.925 | 0.935 |
| | intermediate | 1 | 0.563 | 0.210 | 0.205 | 0.140 | 0.493 | 0.000 | 0.000 | 0.000 |
| | | 10 | 0.999 | 0.500 | 0.473 | 0.406 | 0.493 | 0.315 | 0.345 | 0.279 |
| | | 100 | 0.762 | 0.569 | 0.527 | 0.474 | 0.491 | 0.789 | 0.791 | 0.740 |
| | | 1000 | 0.992 | 0.569 | 0.527 | 0.474 | 0.490 | 0.802 | 0.839 | 0.797 |
| | | 10000 | 0.992 | 0.569 | 0.527 | 0.474 | 0.492 | 0.852 | 0.901 | 0.930 |
| | leaf | 1 | 0.766 | 0.235 | 0.244 | 0.183 | 0.493 | 0.000 | 0.000 | 0.000 |
| | | 10 | 0.994 | 0.568 | 0.501 | 0.462 | 0.493 | 0.290 | 0.347 | 0.239 |
| | | 100 | 0.748 | 0.740 | 0.666 | 0.658 | 1.000 | 0.765 | 0.829 | 0.723 |
| | | 1000 | 0.717 | 0.782 | 0.706 | 0.713 | 0.486 | 0.854 | 1.000 | 0.842 |
| | | 10000 | 0.717 | 0.782 | 0.706 | 0.713 | 0.488 | 0.878 | 0.955 | 1.000 |
| physics | root | 1 | 0.998 | 0.092 | 0.044 | 0.050 | 1.000 | 0.000 | 0.000 | 0.000 |
| | | 10 | 0.998 | 0.284 | 0.213 | 0.227 | 0.987 | 0.344 | 0.323 | 0.279 |
| | | 100 | 0.998 | 0.302 | 0.229 | 0.249 | 0.575 | 0.836 | 0.836 | 0.741 |
| | | 1000 | 0.998 | 0.302 | 0.232 | 0.249 | 0.639 | 0.852 | 0.867 | 0.814 |
| | | 10000 | 0.998 | 0.302 | 0.232 | 0.249 | 0.741 | 0.908 | 0.889 | 0.955 |
| | intermediate | 1 | 0.999 | 0.060 | 0.067 | 0.014 | 1.000 | 0.000 | 0.000 | 0.000 |
| | | 10 | 0.994 | 0.363 | 0.327 | 0.318 | 0.984 | 0.333 | 0.316 | 0.273 |
| | | 100 | 0.736 | 0.548 | 0.525 | 0.522 | 0.627 | 0.854 | 0.854 | 0.753 |
| | | 1000 | 0.761 | 0.548 | 0.525 | 0.522 | 0.638 | 0.879 | 0.847 | 0.834 |
| | | 10000 | 0.761 | 0.548 | 0.525 | 0.522 | 0.744 | 0.855 | 0.836 | 0.905 |
| | leaf | 1 | 0.999 | 0.098 | 0.080 | 0.048 | 1.000 | 0.000 | 0.000 | 0.000 |
| | | 10 | 0.995 | 0.366 | 0.335 | 0.330 | 0.990 | 0.321 | 0.366 | 0.264 |
| | | 100 | 0.804 | 0.619 | 0.680 | 0.592 | 0.800 | 0.686 | 0.697 | 0.654 |
| | | 1000 | 0.704 | 0.703 | 0.676 | 0.682 | 0.703 | 0.799 | 0.815 | 0.829 |
| | | 10000 | 0.704 | 0.703 | 0.676 | 0.682 | 0.716 | 0.880 | 0.867 | 0.963 |

Table 18: Normalized model similarity scores for Mistral

| Subject | Branch | Train Size | Editing | | | | Unlearning | | | |
|---|---|---|---|---|---|---|---|---|---|---|
| | | | CKA | Fisher | KL | L2 | CKA | Fisher | KL | L2 |
| biology | root | 1 | 0.999 | 0.023 | 0.000 | 0.022 | 1.000 | 0.000 | 0.000 | 0.000 |
| | | 10 | 0.480 | 0.547 | 0.977 | 0.532 | 0.988 | 0.347 | 0.376 | 0.000 |
| | | 100 | 0.351 | 0.757 | 0.958 | 0.749 | 0.410 | 0.857 | 1.000 | 0.000 |
| | | 1000 | 0.313 | 0.955 | 0.963 | 0.958 | 0.577 | 0.863 | 0.856 | 0.410 |
| | | 10000 | 0.313 | 0.955 | 0.963 | 0.958 | 0.778 | 0.911 | 0.837 | 0.978 |
| | intermediate | 1 | 0.986 | 0.028 | 0.010 | 0.023 | 1.000 | 0.000 | 0.000 | 0.000 |
| | | 10 | 0.232 | 0.575 | 0.966 | 0.564 | 0.991 | 0.341 | 0.366 | 0.000 |
| | | 100 | 0.249 | 0.758 | 0.977 | 0.754 | 0.491 | 0.854 | 0.908 | 0.000 |
| | | 1000 | 0.385 | 0.972 | 0.994 | 0.972 | 0.498 | 0.884 | 0.820 | 0.319 |
| | | 10000 | 0.385 | 0.972 | 0.994 | 0.972 | 0.751 | 0.899 | 0.800 | 1.000 |
| | leaf | 1 | 1.000 | 0.026 | 0.030 | 0.022 | 1.000 | 0.000 | 0.000 | 0.000 |
| | | 10 | 0.479 | 0.560 | 0.958 | 0.549 | 0.991 | 0.317 | 0.396 | 0.000 |
| | | 100 | 0.683 | 0.762 | 0.957 | 0.739 | 0.537 | 0.817 | 0.932 | 0.000 |
| | | 1000 | 0.578 | 0.952 | 0.963 | 0.953 | 0.571 | 0.849 | 0.873 | 0.327 |
| | | 10000 | 0.578 | 0.952 | 0.963 | 0.953 | 0.803 | 0.883 | 0.967 | 0.988 |
| economics | root | 1 | 0.986 | 0.012 | 0.139 | 0.013 | 1.000 | 0.000 | 0.000 | 0.000 |
| | | 10 | 0.351 | 0.543 | 1.000 | 0.526 | 0.986 | 0.350 | 0.314 | 0.288 |
| | | 100 | 0.147 | 0.736 | 0.982 | 0.719 | 0.432 | 0.858 | 0.790 | 0.747 |
| | | 1000 | 0.197 | 0.898 | 0.966 | 0.894 | 0.524 | 0.879 | 0.777 | 0.809 |
| | | 10000 | 0.197 | 0.898 | 0.966 | 0.894 | 0.489 | 0.975 | 0.880 | 0.912 |
| | intermediate | 1 | 0.999 | 0.027 | 0.087 | 0.017 | 1.000 | 0.000 | 0.000 | 0.000 |
| | | 10 | 0.242 | 0.552 | 0.981 | 0.541 | 0.993 | 0.342 | 0.425 | 0.000 |
| | | 100 | 0.199 | 0.754 | 0.981 | 0.744 | 0.632 | 0.863 | 0.938 | 0.000 |
| | | 1000 | 0.143 | 0.971 | 0.995 | 0.973 | 0.667 | 0.875 | 0.774 | 0.329 |
| | | 10000 | 0.143 | 0.971 | 0.995 | 0.973 | 0.731 | 0.908 | 0.762 | 0.951 |
| | leaf | 1 | 0.986 | 0.015 | 0.124 | 0.012 | 1.000 | 0.000 | 0.000 | 0.000 |
| | | 10 | 0.523 | 0.569 | 0.949 | 0.556 | 0.989 | 0.321 | 0.354 | 0.267 |
| | | 100 | 0.373 | 0.788 | 0.964 | 0.773 | 0.636 | 0.834 | 0.849 | 0.734 |
| | | 1000 | 0.324 | 0.989 | 0.974 | 0.991 | 0.642 | 0.865 | 0.827 | 0.835 |
| | | 10000 | 0.324 | 0.989 | 0.974 | 0.991 | 0.686 | 0.957 | 0.854 | 0.936 |
| history | root | 1 | 0.999 | 0.044 | 0.214 | 0.019 | 1.000 | 0.000 | 0.000 | 0.000 |
| | | 10 | 0.285 | 0.560 | 0.960 | 0.546 | 0.987 | 0.347 | 0.342 | 0.282 |
| | | 100 | 0.163 | 0.767 | 0.969 | 0.760 | 0.511 | 0.844 | 0.793 | 0.739 |
| | | 1000 | 0.185 | 0.929 | 0.959 | 0.930 | 0.660 | 0.849 | 0.864 | 0.807 |
| | | 10000 | 0.185 | 0.929 | 0.959 | 0.930 | 0.725 | 0.911 | 0.890 | 0.978 |
| | intermediate | 1 | 0.999 | 0.035 | 0.238 | 0.021 | 1.000 | 0.000 | 0.000 | 0.000 |
| | | 10 | 0.448 | 0.568 | 0.956 | 0.551 | 0.988 | 0.347 | 0.403 | 0.282 |
| | | 100 | 0.136 | 0.751 | 0.957 | 0.743 | 0.316 | 0.855 | 0.901 | 0.743 |
| | | 1000 | 0.000 | 1.000 | 0.963 | 1.000 | 0.490 | 0.839 | 0.803 | 0.807 |
| | | 10000 | 0.000 | 1.000 | 0.963 | 1.000 | 0.760 | 0.915 | 0.879 | 0.979 |
| | leaf | 1 | 0.986 | 0.051 | 0.211 | 0.034 | 1.000 | 0.000 | 0.000 | 0.000 |
| | | 10 | 0.626 | 0.552 | 0.954 | 0.537 | 0.988 | 0.317 | 0.384 | 0.246 |
| | | 100 | 0.560 | 0.776 | 0.957 | 0.757 | 0.532 | 0.825 | 0.885 | 0.725 |
| | | 1000 | 0.575 | 0.960 | 0.954 | 0.959 | 0.272 | 0.878 | 0.940 | 0.857 |
| | | 10000 | 0.575 | 0.960 | 0.954 | 0.959 | 0.675 | 0.875 | 0.925 | 0.997 |
| physics | root | 1 | 0.999 | 0.000 | 0.018 | 0.000 | 1.000 | 0.000 | 0.000 | 0.000 |
| | | 10 | 0.376 | 0.566 | 0.960 | 0.551 | 0.987 | 0.344 | 0.323 | 0.279 |
| | | 100 | 0.234 | 0.775 | 0.969 | 0.763 | 0.575 | 0.836 | 0.836 | 0.741 |
| | | 1000 | 0.117 | 0.971 | 0.971 | 0.975 | 0.639 | 0.852 | 0.867 | 0.814 |
| | | 10000 | 0.117 | 0.971 | 0.971 | 0.975 | 0.741 | 0.908 | 0.889 | 0.955 |
| | intermediate | 1 | 0.999 | 0.029 | 0.042 | 0.024 | 1.000 | 0.000 | 0.000 | 0.000 |
| | | 10 | 0.443 | 0.591 | 0.964 | 0.573 | 0.984 | 0.333 | 0.316 | 0.273 |
| | | 100 | 0.349 | 0.788 | 0.962 | 0.780 | 0.627 | 0.854 | 0.854 | 0.753 |
| | | 1000 | 0.211 | 0.985 | 0.970 | 0.983 | 0.638 | 0.879 | 0.847 | 0.834 |
| | | 10000 | 0.211 | 0.985 | 0.970 | 0.983 | 0.744 | 0.855 | 0.836 | 0.905 |
| | leaf | 1 | 0.986 | 0.065 | 0.048 | 0.025 | 1.000 | 0.000 | 0.000 | 0.000 |
| | | 10 | 0.333 | 0.566 | 0.967 | 0.548 | 0.990 | 0.321 | 0.366 | 0.264 |
| | | 100 | 0.647 | 0.775 | 0.967 | 0.764 | 0.800 | 0.686 | 0.697 | 0.654 |
| | | 1000 | 0.285 | 0.974 | 0.965 | 0.977 | 0.703 | 0.799 | 0.815 | 0.829 |
| | | 10000 | 0.285 | 0.974 | 0.965 | 0.977 | 0.716 | 0.880 | 0.867 | 0.963 |

Table 19: Normalized model similarity scores for Gemma

| Subject | Branch | Train Size | edit | | | | Unlearning | | | |
|---|---|---|---|---|---|---|---|---|---|---|
| | | | CKA | Fisher | KL | L2 | CKA | Fisher | KL | L2 |
| biology | root | 1 | 1.000 | 0.152 | 0.000 | 0.000 | 0.791 | 0.000 | 0.013 | 0.052 |
| | | 10 | 0.961 | 0.377 | 0.652 | 0.299 | 0.805 | 0.461 | 0.561 | 0.394 |
| | | 100 | 0.962 | 0.462 | 0.787 | 0.575 | 0.949 | 0.651 | 0.645 | 0.452 |
| | | 1000 | 0.653 | 0.692 | 0.954 | 0.813 | 0.875 | 0.757 | 0.725 | 0.547 |
| | | 10000 | 0.653 | 0.692 | 0.954 | 0.813 | 0.904 | 0.947 | 0.889 | 0.876 |
| | intermediate | 1 | 1.000 | 0.000 | 0.197 | 0.040 | 1.000 | 0.000 | 0.000 | 0.000 |
| | | 10 | 0.225 | 0.686 | 0.944 | 0.452 | 0.588 | 0.370 | 0.550 | 0.354 |
| | | 100 | 0.183 | 0.809 | 0.981 | 0.673 | 0.596 | 0.542 | 0.642 | 0.462 |
| | | 1000 | 0.196 | 0.730 | 0.999 | 0.925 | 0.859 | 0.837 | 0.741 | 0.530 |
| | | 10000 | 0.196 | 0.730 | 0.999 | 0.925 | 0.888 | 1.000 | 0.905 | 0.859 |
| | leaf | 1 | 0.544 | 0.673 | 0.932 | 0.446 | 0.853 | 0.000 | 0.000 | 0.000 |
| | | 10 | 0.169 | 0.795 | 0.959 | 0.586 | 0.544 | 0.466 | 0.610 | 0.435 |
| | | 100 | 0.115 | 0.877 | 0.954 | 0.699 | 0.508 | 0.561 | 0.666 | 0.511 |
| | | 1000 | 0.158 | 0.859 | 1.000 | 0.936 | 0.710 | 0.807 | 0.761 | 0.590 |
| | | 10000 | 0.158 | 0.859 | 1.000 | 0.936 | 0.739 | 0.997 | 0.925 | 0.920 |
| economics | root | 1 | 0.994 | 0.273 | 0.548 | 0.041 | 0.870 | 0.000 | 0.000 | 0.000 |
| | | 10 | 0.943 | 0.378 | 0.753 | 0.330 | 0.874 | 0.441 | 0.545 | 0.347 |
| | | 100 | 0.703 | 0.504 | 0.813 | 0.595 | 0.889 | 0.645 | 0.639 | 0.438 |
| | | 1000 | 0.152 | 0.726 | 0.964 | 0.897 | 0.822 | 0.828 | 0.737 | 0.534 |
| | | 10000 | 0.152 | 0.726 | 0.964 | 0.897 | 0.851 | 1.000 | 0.901 | 0.863 |
| | intermediate | 1 | 0.999 | 0.191 | 0.324 | 0.009 | 0.886 | 0.000 | 0.000 | 0.000 |
| | | 10 | 0.283 | 0.661 | 0.873 | 0.435 | 0.578 | 0.397 | 0.562 | 0.382 |
| | | 100 | 0.174 | 0.798 | 0.970 | 0.638 | 0.580 | 0.515 | 0.630 | 0.451 |
| | | 1000 | 0.238 | 0.886 | 0.969 | 0.931 | 0.671 | 0.809 | 0.775 | 0.626 |
| | | 10000 | 0.238 | 0.886 | 0.969 | 0.931 | 0.699 | 1.000 | 0.939 | 0.956 |
| | leaf | 1 | 0.300 | 0.635 | 0.944 | 0.440 | 0.866 | 0.000 | 0.000 | 0.000 |
| | | 10 | 0.154 | 0.700 | 0.967 | 0.544 | 0.631 | 0.449 | 0.578 | 0.370 |
| | | 100 | 0.254 | 0.887 | 0.955 | 0.664 | 0.495 | 0.521 | 0.660 | 0.518 |
| | | 1000 | 0.000 | 1.000 | 0.979 | 1.000 | 0.552 | 0.839 | 0.804 | 0.673 |
| | | 10000 | 0.000 | 1.000 | 0.979 | 1.000 | 0.581 | 1.000 | 0.968 | 1.000 |
| history | root | 1 | 1.000 | 0.061 | 0.074 | 0.036 | 0.878 | 0.189 | 0.182 | 0.173 |
| | | 10 | 0.149 | 0.762 | 0.924 | 0.489 | 0.501 | 0.394 | 0.577 | 0.405 |
| | | 100 | 0.108 | 0.657 | 0.974 | 0.639 | 0.738 | 0.559 | 0.606 | 0.375 |
| | | 1000 | 0.106 | 0.870 | 0.970 | 0.925 | 0.668 | 0.807 | 0.764 | 0.603 |
| | | 10000 | 0.106 | 0.870 | 0.970 | 0.925 | 0.696 | 0.998 | 0.928 | 0.933 |
| | intermediate | 1 | 0.999 | 0.057 | 0.317 | 0.063 | 1.000 | 0.000 | 0.000 | 0.000 |
| | | 10 | 0.511 | 0.759 | 0.931 | 0.474 | 0.550 | 0.376 | 0.585 | 0.432 |
| | | 100 | 0.354 | 0.883 | 0.964 | 0.687 | 0.533 | 0.541 | 0.670 | 0.528 |
| | | 1000 | 0.401 | 0.869 | 0.955 | 0.942 | 0.713 | 0.833 | 0.788 | 0.646 |
| | | 10000 | 0.401 | 0.869 | 0.955 | 0.942 | 0.742 | 1.000 | 0.952 | 0.976 |
| | leaf | 1 | 0.325 | 0.753 | 0.942 | 0.463 | 0.745 | 0.000 | 0.000 | 0.000 |
| | | 10 | 0.424 | 0.759 | 0.934 | 0.499 | 0.555 | 0.400 | 0.590 | 0.429 |
| | | 100 | 0.317 | 0.863 | 0.974 | 0.625 | 0.518 | 0.479 | 0.639 | 0.490 |
| | | 1000 | 0.276 | 0.863 | 0.989 | 0.874 | 0.678 | 0.747 | 0.746 | 0.590 |
| | | 10000 | 0.276 | 0.863 | 0.989 | 0.874 | 0.707 | 0.938 | 0.910 | 0.920 |
| physics | root | 1 | 1.000 | 0.094 | 0.106 | 0.025 | 1.000 | 0.000 | 0.000 | 0.000 |
| | | 10 | 0.135 | 0.847 | 0.946 | 0.535 | 0.440 | 0.403 | 0.600 | 0.447 |
| | | 100 | 0.120 | 0.880 | 0.950 | 0.711 | 0.511 | 0.574 | 0.672 | 0.519 |
| | | 1000 | 0.059 | 0.880 | 0.959 | 0.907 | 0.632 | 0.791 | 0.759 | 0.603 |
| | | 10000 | 0.059 | 0.880 | 0.959 | 0.907 | 0.661 | 0.982 | 0.923 | 0.933 |
| | intermediate | 1 | 0.998 | 0.087 | 0.342 | 0.063 | 1.000 | 0.000 | 0.000 | 0.000 |
| | | 10 | 0.759 | 0.414 | 0.787 | 0.363 | 0.844 | 0.446 | 0.545 | 0.337 |
| | | 100 | 0.562 | 0.429 | 0.851 | 0.600 | 0.980 | 0.655 | 0.614 | 0.369 |
| | | 1000 | 0.159 | 0.675 | 0.995 | 0.846 | 0.867 | 0.775 | 0.702 | 0.477 |
| | | 10000 | 0.159 | 0.675 | 0.995 | 0.846 | 0.896 | 0.966 | 0.866 | 0.806 |
| | leaf | 1 | 1.000 | 0.185 | 0.220 | 0.036 | 0.860 | 0.000 | 0.000 | 0.000 |
| | | 10 | 0.258 | 0.837 | 0.922 | 0.498 | 0.433 | 0.381 | 0.596 | 0.456 |
| | | 100 | 0.267 | 0.701 | 0.964 | 0.649 | 0.710 | 0.560 | 0.626 | 0.421 |
| | | 1000 | 0.142 | 0.882 | 0.976 | 0.908 | 0.651 | 0.783 | 0.759 | 0.604 |
| | | 10000 | 0.142 | 0.882 | 0.976 | 0.908 | 0.680 | 0.973 | 0.923 | 0.933 |

