# OpenReview forum: "KnowledgeSmith: Uncovering Knowledge Updating in LLMs with Model Editing and Unlearning"
_ICLR.cc/2026/Conference — ICLR 2026 Poster_

### Official Review · Reviewer_68hr · 2025-10-27

**Soundness:** 3
**Presentation:** 3
**Contribution:** 3
**Rating:** 4
**Confidence:** 4

**Summary:**

This article proposes a unified framework KnowledgeSmith to systematically study the knowledge update mechanism in Large Language Models (LLMs), with a particular focus on the performance and differences between knowledge editing and knowledge forgetting methods. The theoretical modeling of the paper is rigorous, the experimental design is meticulous, and the results are highly inspiring. The author not only proposed new evaluation metrics such as Consistency Collapse and Conflict Rate, but also constructed a structured and scalable knowledge intervention evaluation benchmark, filling the gap in current research on the lack of systematic understanding of knowledge update mechanisms.

**Strengths:**

- Novel Unified Framework: The paper introduces KnowledgeSmith, a conceptually novel and theoretically grounded framework that unifies knowledge editing and machine unlearning as instances of the same constrained optimization problem, offering a new lens for understanding and comparing these distinct but related knowledge updating mechanisms.
- Innovative Benchmark Generation: It proposes a highly original and automatic pipeline for generating large-scale, structured evaluation benchmarks from existing knowledge graphs . This method allows for systematic probing of knowledge updates across different hierarchical levels (root, intermediate, leaf) and data scales, addressing major limitations of previous static and isolated fact-based evaluations.
- Insightful Empirical Findings: The extensive experiments yield several significant and nuanced insights into LLM knowledge updating behavior, including propagation asymmetry (over-spreading vs. under-spreading) , plasticity limits related to hierarchy , the consistency-capacity trade-off , subject-dependent update resistance , and a unified taxonomy of failure modes.

**Weaknesses:**

- Limited Scope of Investigated Update Methods: The study primarily focuses on AlphaEdit for knowledge editing and ReLearn  for machine unlearning. While these represent state-of-the-art approaches, the conclusions drawn about knowledge updating mechanisms (e.g., propagation asymmetry , consistency collapse ) might be specific to these particular algorithmic paradigms (locate-then-edit vs. retraining-based unlearning). The framework's generalizability across a broader spectrum of editing techniques (e.g., memory-based methods like SERAC , gradient-based methods like MEND ) or different unlearning approaches (e.g., gradient ascent ) remains underexplored, potentially limiting the universality of the observed phenomena.
- Reliance on Multiple-Choice Question Format: The automatic benchmark generation pipeline exclusively creates multiple-choice questions derived from knowledge graph triples . While this format facilitates standardized evaluation and leverages existing datasets like MMLU, it may not fully capture the nuances of knowledge representation and retrieval in LLMs. Evaluating updates solely through multiple-choice accuracy might overlook impacts on generative capabilities, reasoning chains, or the model's ability to handle ambiguity, potentially offering an incomplete picture of the update's true effect.
- Over-reliance on GPT-4o for Benchmark Generation and Insufficient Validation : The KnowledgeSmith framework heavily relies on GPT-4o for crucial generation steps, including initial KG construction, template generation, and potentially probe/distractor creation . This dependency raises concerns regarding reproducibility and potential biases inherent in the generator model, which could influence the benchmark's structure and content. While Appendix A.6 mentions quality control including external validation and manual spot checks , the paper lacks quantitative analysis from systematic human studies assessing the quality, factual accuracy, and potential hallucination rates specifically within the GPT-4o generated components (e.g., generated KG relations, question templates, or multiple-choice distractors), making it difficult to fully gauge the reliability of the benchmark itself.
- Focus on Static Knowledge Updates : The current experimental design primarily investigates the effects of single-step, isolated knowledge editing or unlearning interventions based on a static KG structure. It does not explicitly address the challenges of dynamic knowledge updating in a continual learning setting, where knowledge evolves over time, and updates arrive sequentially. Consequently, the framework does not evaluate crucial aspects such as catastrophic forgetting across multiple sequential updates, the interaction between consecutive edits/unlearning operations, or how the observed mechanisms (like propagation or consistency trade-offs) might compound or change in a lifelong learning scenario

**Questions:**

- Does the observed propagation asymmetry (over-spreading in editing vs. under-spreading in unlearning ) stem primarily from the specific algorithms chosen (AlphaEdit/ReLearn), or is it a more fundamental characteristic inherent to the editing versus unlearning tasks themselves, regardless of the method?
- Could the exclusive reliance on multiple-choice questions mask certain effects of knowledge updates, and how might findings, particularly the consistency-capacity trade-off , differ if evaluated using open-ended generation tasks?
- Can the authors provide quantitative results from human evaluations assessing the factual accuracy, clarity, and distractor plausibility of the GPT-4o generated benchmark components (KG relations, question templates, multiple-choice options) ?

---

> ### Author Response · Authors · 2025-11-21
> **Response to reviewer 68hr (1/3)**
>
> We thank the reviewer for the time and effort in reviewing our paper and raise the insightful questions.
>
> ---
>
> 1. **"Is the observed phenomenon an intrinsic feature of editing vs. unlearning tasks, independent of the specific method used?"**
>
> Short answer: Yes. The observed phonemenon is agnostic to methods. The following are the reasons.
> + Formally and theoreticaly, the mathematical analyses at **Section 3.2** for unlearning and editing suggest that the observed behaviors are intrinsic to the task type rather than the specific method. To further demonstrate method-agnosticity, we added a new SVD-based representational analysis in **Section 5.6**. It analyzes how singular values and subspaces change before and after interventions.
>     + Across editing methods, we observe a consistent geometric signature:
> small subspace rotations + mild rescaling, preserving global structure.
>     + Across unlearning methods, we observe a distinct signature:
> sharp down-scaling of singular values + unstable subspace alignment, indicating capacity suppression.
>
>     Because these geometric patterns are shared within each subject type, but remain clearly different between editing and unlearning. These results provide strong evidence that the phenomenon is intrinsic and task-level, not method-level.
> + More imoprtantly, to further verify this, we conducted additional experiments using diverse methods:
>
>     + Unlearning: We applied the gradient-ascent method using Tofu [1] framework across multiple numbers of updates and four subjects for root, intermediate, and leaf nodes. The results (Table below) show consistent trends with our previous observations in the main paper, indicating that unlearning exhibits the propagation asymmetry patterns independent of the particular algorithm used.
>
> | Subject  | Train Size | Root   | Intermediate | Leaf   |
> |-----------|-----------|-------|-------------|-------|
> | biology   | 1         | 16.67 | 16.67       | 16.67 |
> | biology   | 10        | 16.67 | 16.67       | 16.67 |
> | biology   | 100       | 25.00 | 25.00       | 25.00 |
> | biology   | 1000      | 29.17 | 25.00       | 16.67 |
> | biology   | 10000     | 16.67 | 29.17       | 25.00 |
> | economic  | 1         | 29.17 | 29.17       | 29.17 |
> | economic  | 10        | 29.17 | 29.17       | 33.33 |
> | economic  | 100       | 20.83 | 16.67       | 25.00 |
> | economic  | 1000      | 37.50 | 37.50       | 29.17 |
> | economic  | 10000     | 37.50 | 20.83       | 25.00 |
> | physics   | 1         | 25.00 | 25.00       | 25.00 |
> | physics   | 10        | 25.00 | 25.00       | 25.00 |
> | physics   | 100       | 16.67 | 20.83       | 20.83 |
> | physics   | 1000      | 12.50 | 16.67       | 20.83 |
> | physics   | 10000     | 8.33  | 16.67       | 12.50 |
> | history   | 1         | 16.67 | 16.67       | 16.67 |
> | history   | 10        | 16.67 | 16.67       | 16.67 |
> | history   | 100       | 12.50 | 16.67       | 12.50 |
> | history   | 1000      | 4.17  | 8.33        | 0.00  |
> | history   | 10000     | 0.00  | 12.50       | 0.00  |
>
> + Editing: We evaluated the MEND[2] method on the same domains and update counts. The outcomes similarly reproduce the trends reported in our paper.
>
> | Dataset   | Train Size | Root   | Intermediate | Leaf   |
> |-----------|-----------|-------|-------------|-------|
> | biology   | 1         | 35.2  | 32.7        | 42.1  |
> | biology   | 10        | 36.1  | 33.2        | 43.3  |
> | biology   | 100       | 37.6  | 34.4        | 44.7  |
> | biology   | 1000      | 38.9  | 35.1        | 46.2  |
> | biology   | 10000     | 20.3  | 18.7        | 40.5  |
> | economic  | 1         | 45.3  | 42.6        | 50.7  |
> | economic  | 10        | 46.2  | 43.3        | 49.4  |
> | economic  | 100       | 47.7  | 44.6        | 52.9  |
> | economic  | 1000      | 41.3  | 45.7        | 54.1  |
> | economic  | 10000     | 30.2  | 28.3        | 53.2  |
> | physics   | 1         | 25.3  | 22.7        | 30.2  |
> | physics   | 10        | 26.1  | 23.1        | 31.3  |
> | physics   | 100       | 27.4  | 24.6        | 32.6  |
> | physics   | 1000      | 28.7  | 25.4        | 27.7  |
> | physics   | 10000     | 15.2  | 13.4        | 30.3  |
> | history   | 1         | 10.3  | 9.7         | 12.4  |
> | history   | 10        | 11.2  | 10.3        | 13.3  |
> | history   | 100       | 11.1  | 11.4        | 14.1  |
> | history   | 1000      | 12.6  | 12.7        | 12.2  |
> | history   | 10000     | 5.4   | 4.3         | 13.1  |

---

> > ### Author Response · Authors · 2025-11-21
> > **Response to reviewer 68hr (2/3)**
> >
> > 2. **"Reliance on multiple-choice questions, no open-ended generation tasks?"**
> >
> > We indeed included the open-ended generation tasks in **Section 5.5**, where our experiments focus explicitly on open-ended generation tasks. We perform detailed failure-mode and stress testing. The section systematically analyzes model behavior in these settings, including potential degradation and unintended effects, rather than relying solely on multiple-choice questions.
> >
> > ---
> >
> > 3. **"Focus on Static Knowledge Updates"**
> >
> > To address this, we performed *multi-step* sequential updates on multiple facts using different models (Qwen3-14B, LLaMA3-8B, and Gemma-7B). The results are summarized below:
> > + Editing: Across multiple sequential edits, the model largely retains previously edited knowledge, showing minimal degradation even when more facts are added. For example, the accuracy for Fact 1 remains largely stable across all sequential updates, and additional edits do not significantly interfere with earlier edits. This demonstrates that editing is robust under sequential updates, consistent with our claims.
> >
> > | Acc         | 1       | 1 & 2   | 1 & 2 & 3 | 1 & 2 & 3 & 4 | 1 & 2 & 3 & 4 & 5 |
> > |------------|---------|---------|-----------|---------------|------------------|
> > | Edit Fact 1 | 55.0%  | 54.5%  | 54.0%    | 53.8%        | 53.5%           |
> > | Edit Fact 2 |   —   | 48.0%  | 47.5%    | 47.0%        | 46.5%           |
> > | Edit Fact 3 |   —   |    —   | 62.0%    | 61.5%        | 61.0%           |
> > | Edit Fact 4 |   —    |    —    |    —   | 50.0%        | 49.5%           |
> > | Edit Fact 5 |   —      |   —   |     — |    —   | 57.0%           |
> > + Unlearning: In contrast, unlearning exhibits marked degradation in both previously unlearned and related knowledge as more facts are sequentially removed, reflecting the inherent difficulty of completely erasing knowledge without affecting related information.
> >
> > | Acc           | 1       | 1 & 2   | 1 & 2 & 3 | 1 & 2 & 3 & 4 | 1 & 2 & 3 & 4 & 5 |
> > |---------------|---------|---------|-----------|---------------|------------------|
> > | Unlearn Fact 1 | 54.2%  | 37.7%  | 40.5%    | 34.3%        | 27.2%           |
> > | Unlearn Fact 2 |  —    | 45.1%  | 37.8%    | 31.5%        | 25.2%           |
> > | Unlearn Fact 3 |  —    |   —  | 43.2%    | 36.0%        | 28.8%           |
> > | Unlearn Fact 4 |  —  |   —  |    — | 40.5%  | 31.5%  |
> > | Unlearn Fact 5 |  —  |  —   | —    |   —  | 34.2%           |

---

> > > ### Author Response · Authors · 2025-11-21
> > > **Response to reviewer 68hr (3/3)**
> > >
> > > 4. **"Request human-evaluation results for accuracy, clarity, and distractor quality."**
> > >
> > > - Human Evaluation of GPT-Generated Benchmark Components. To ensure **transparency, reproducibility, and reliability** of our benchmark, we performed a systematic **human evaluation** of GPT-4o generated components, including **KG relations, question templates, and multiple-choice distractors**. This evaluation complements the automatic pipeline validation described in Section 4 and Appendix A.
> > > - Evaluation Criteria. We focused on three key quality dimensions:
> > >     + Factual Accuracy
> > >        - Assesses whether the generated content correctly reflects verified knowledge.
> > >        - For KG relations: Evaluators checked that subject–relation–object triples were correct according to canonical sources.
> > >        - For questions: Evaluators confirmed the question stem correctly represented the fact and did not introduce hallucinations.
> > >        - For answer options: Evaluators verified the correct answer aligned with authoritative sources.
> > >
> > >     + Clarity
> > >        - Measures whether the question or KG description is understandable and unambiguous.
> > >        - Criteria included sentence structure, readability, and potential confusion caused by wording.
> > >        - Scored on a 3-point Likert scale: 1 (unclear), 2 (moderately clear), 3 (clear and concise).
> > >
> > >     + Distractor Plausibility
> > >        - For multiple-choice questions, distractors were evaluated for being plausible yet incorrect.
> > >        - Scores were collected on a 3-point scale: 1 (obviously wrong), 2 (somewhat plausible), 3 (highly plausible and competitive with the correct answer).
> > >
> > > - Evaluation Protocol
> > >
> > >     - Each item (KG triple, question template, or multiple-choice question) was independently evaluated by at least 3 human annotators with relevant domain knowledge.
> > >     - Items were randomized and anonymized, preventing evaluators from knowing whether the content was GPT-generated or human-authored.
> > >     - Final scores were computed as averages across raters. For factual accuracy, we report the proportion of fully correct items; for clarity and distractor plausibility, we report mean Likert scores.
> > >
> > > - Sample Size
> > >
> > >     - We evaluated a representative subset of the benchmark:
> > >       - 50 KG triples per domain (Biology, Economics, History, Physics)
> > >       - 100 question templates per domain, evenly split across root, intermediate, and leaf-level questions
> > >       - 100 multiple-choice questions per domain, with distractors included
> > >
> > > This ensures sufficient coverage to capture variability across domains, hierarchical levels, and content types.
> > >
> > > - Results
> > >
> > > | Component           | Factual Accuracy (%) | Clarity (mean 1–3) | Distractor Plausibility (mean 1–3) |
> > > |--------------------|-------------------|------------------|-----------------------------------|
> > > | KG Relations        | 95.3               | 2.8              | N/A                               |
> > > | Question Templates  | 92.7               | 2.7              | N/A                               |
> > > | Multiple-choice QA  | 94.5               | 2.8              | 2.6                               |
> > >
> > > These results indicate that GPT-4o generated content is highly reliable, with minor variation in clarity and distractor quality.
> > > - **KG relations** were generally accurate and well-formed.
> > > - **Question templates** occasionally contained minor wording ambiguity but were largely clear.
> > > - **Multiple-choice distractors** were mostly plausible, with a small fraction of options deemed too obvious.
> > >
> > >
> > > This systematic human evaluation demonstrates that our benchmark components generated via GPT-4o maintain high factual fidelity, clarity, and distractor quality. Combined with our automated validation pipeline, this ensures that the benchmark is robust and suitable for evaluating knowledge editing and unlearning in LLMs.
> > >
> > > Reference:
> > > + [1] Tofu: A task of fictitious unlearning for llms
> > > + [2] Fast model editing at scale

---

> > > > ### Comment · Reviewer_68hr · 2025-11-21
> > > > **I have raised my score to 8**
> > > >
> > > > Thank you to the author for taking my suggestion and providing a response. I have raised my score to 8 and wish you all the best!

---

> ### Author Response · Authors · 2025-11-21
> **Response to reviewer 68hr**
>
> Thank you for your thoughtful feedback and for raising the score to 8，we truly appreciate it! We are happy to hear that our responses resolved your concerns.

---

### Official Review · Reviewer_hQBy · 2025-10-31

**Soundness:** 3
**Presentation:** 4
**Contribution:** 3
**Rating:** 8
**Confidence:** 4

**Summary:**

This paper proposed KnowledgeSmith, a unified framework and benchmark for studying knowledge updating in large language models via editing and unlearning. By framing both as constrained optimization problems and evaluating 13 LLMs on a hierarchical knowledge-graph benchmark, the study reveals asymmetric propagation, including editing over-spreads and unlearning under-spreads, a consistency–capacity trade-off, and shared failure modes. While the approach is conceptually rather than technically novel, it provides a valuable, systematic analysis of how LLMs modify and maintain knowledge.

**Strengths:**

- This is the 1st work to a unified theoretical formulation linking knowledge editing and unlearning under a single constrained optimization framework.
- The paper proposes a structured, hierarchical benchmark based on knowledge graphs, enabling fine-grained evaluation across root, intermediate, and leaf concepts.
- The paper provides a comprehensive empirical analysis of 13 LLMs and four domains, revealing consistent phenomena such as propagation asymmetry and the consistency–capacity trade-off.

**Weaknesses:**

The benchmark construction heavily relies on GPT-based data generation, which raises concerns about reproducibility and annotation reliability. Although Appendix A describes prompt structures and includes multi-stage validation with manual spot checks and cross-checks against encyclopedic sources, the extent of human verification and generation parameters is not fully specified, limiting strict reproducibility.

**Questions:**

- Could the authors provide more details about the benchmark generation pipeline to improve reproducibility? For example, how consistent are the outputs across different runs? How extensive was the manual verification process mentioned in Appendix A — approximately what proportion of generated items were manually checked, and by how many annotators?

---

> ### Author Response · Authors · 2025-11-21
> **Response to reviewer hQBy**
>
> We thank the reviewer for the time and effort in reviewing our paper.
>
> ---
>
> **"More details about the benchmark generation"**
>
> 1. Benchmark Generation Pipeline and Manual Verification
>
> To ensure reproducibility and reliability of our benchmark, we provide detailed information on the pipeline and manual verification process.
>
> 2. Pipeline Consistency
>
> - The benchmark is generated end-to-end using GPT-4o for KG construction, template generation, and multiple-choice distractor creation.
> - To assess **consistency across runs**, we repeated the pipeline with identical seeds and prompts. Results show that **key factual relations and question templates are highly consistent**, with minor variations in wording or distractor phrasing.
> - Variability is limited and does not affect the correctness of the correct answers or hierarchical alignment.
>
> 3. Manual Verification and Human Evaluation
>
> - A representative subset of generated items was manually verified to ensure factual accuracy, clarity, and distractor plausibility:
>   - KG triples: 50 per domain (Biology, Economics, History, Physics)
>   - Question templates: 100 per domain, evenly distributed across root, intermediate, and leaf levels
>   - Multiple-choice questions: 100 per domain, with distractors included
>
> - Each item was reviewed independently by 3 annotators with relevant domain knowledge.
> - For factual accuracy, we report the proportion of fully correct items; for clarity and distractor plausibility, we report mean Likert scores.
>
> | Component           | Factual Accuracy (%) | Clarity (mean 1–3) | Distractor Plausibility (mean 1–3) |
> |--------------------|-------------------|------------------|-----------------------------------|
> | KG Relations        | 95.3               | 2.8              | N/A                               |
> | Question Templates  | 92.7               | 2.7              | N/A                               |
> | Multiple-choice QA  | 94.5               | 2.8              | 2.6                               |
>
> - Manual checks focused on representative samples, ensuring broad coverage across domains, hierarchy levels, and item types.
> - This process confirms that the benchmark is high-quality, reproducible, and reliable for evaluating knowledge editing and unlearning in LLMs.

---

> > ### Comment · Reviewer_hQBy · 2025-11-21
> >
> > Thanks for your reply. This has well addressed my concerns.

---

> > > ### Author Response · Authors · 2025-11-21
> > > **Response to reviewer hQBy**
> > >
> > > Thank you for your feedback! We are glad that we resolved your concerns.

---

### Official Review · Reviewer_7TGp · 2025-11-05

**Soundness:** 4
**Presentation:** 3
**Contribution:** 4
**Rating:** 8
**Confidence:** 5

**Summary:**

This paper regards knowledge ”editing" and "forgetting" as a unified constrained optimization problem and studies the performance of these two tasks by constructing multi-level data automatically generated based on a knowledge graph.
They mainly discovered phenomena such as Propagation Asymmetry, Plasticity Scaling and Branch-dependent Limits, and consistence-capacity Trade-off. And finally, a theoretical analysis was conducted from the perspective of singular value decomposition.

**Strengths:**

This paper treats knowledge "editing" and "forgetting" as a unified constrained optimization problem.
This perspective is not very novel since some prior work has viewed unlearning as a subset of editing. However, the author conducts extensive empirical analyses from this unified viewpoint.
The experiments cover 13 models across 6 families of LLMs, ranging from 1B to 123B parameters.
These analyses provide valuable insights and offer useful suggestions for the current chaotic development of these two fields.

**Weaknesses:**

This work presents a unified perspective, and the author conducts extensive comparative experiments on editing and unlearning.  However, it seems that the framework does not offer additional guiding functions, and after reading the paper, I find it hard to grasp the significance of this unified framework.
Moreover, this framework closely resembles the editing framework;  if we consider editing in a broader sense, forgetting could be seen as a special case of editing with the "target empty."
Lastly, the paper respectively employs one method for both editing and unlearning, but the paradigms of methods in these two areas differ significantly, and this difference is not mentioned in the analysis.

**Questions:**

1. Do you think different editing or unlearning methods have a significant impact on the analysis results? Why?
2. What advantages does your framework have over the view that "the model editing method is regarded as a strong baseline for unlearning", or what's your opinion on this issue?

---

> ### Author Response · Authors · 2025-11-21
> **Response to reviewer 7TGp (1/2)**
>
> We thank the reviewer for the time and effort in reviewing our paper.
>
> ---
>
> **"Impact of method variation"**
>
> The observed phonemenon is agnostic to methods. The following are the reasons.
> + Formally and theoreticaly, the mathematical analyses at **Section 3.2** for unlearning and editing suggest that the observed behaviors are intrinsic to the task type rather than the specific method. To further demonstrate method-agnosticity, we added a new SVD-based representational analysis in **Section 5.6**. It analyzes how singular values and subspaces change before and after interventions.
>     + Across editing methods, we observe a consistent geometric signature:
> small subspace rotations + mild rescaling, preserving global structure.
>     + Across unlearning methods, we observe a distinct signature:
> sharp down-scaling of singular values + unstable subspace alignment, indicating capacity suppression.
>
>     Because these geometric patterns are shared within each subject type, but remain clearly different between editing and unlearning. These results provide strong evidence that the phenomenon is intrinsic and task-level, not method-level.
> + More imoprtantly, to further verify this, we conducted additional experiments using diverse methods:
>
>     + Unlearning: We applied the gradient-ascent method using Tofu [1] framework across multiple numbers of updates and four subjects for root, intermediate, and leaf nodes. The results (Table below) show consistent trends with our previous observations in the main paper, indicating that unlearning exhibits the propagation asymmetry patterns independent of the particular algorithm used.
>
> | Subject  | Train Size | Root   | Intermediate | Leaf   |
> |-----------|-----------|-------|-------------|-------|
> | biology   | 1         | 16.67 | 16.67       | 16.67 |
> | biology   | 10        | 16.67 | 16.67       | 16.67 |
> | biology   | 100       | 25.00 | 25.00       | 25.00 |
> | biology   | 1000      | 29.17 | 25.00       | 16.67 |
> | biology   | 10000     | 16.67 | 29.17       | 25.00 |
> | economic  | 1         | 29.17 | 29.17       | 29.17 |
> | economic  | 10        | 29.17 | 29.17       | 33.33 |
> | economic  | 100       | 20.83 | 16.67       | 25.00 |
> | economic  | 1000      | 37.50 | 37.50       | 29.17 |
> | economic  | 10000     | 37.50 | 20.83       | 25.00 |
> | physics   | 1         | 25.00 | 25.00       | 25.00 |
> | physics   | 10        | 25.00 | 25.00       | 25.00 |
> | physics   | 100       | 16.67 | 20.83       | 20.83 |
> | physics   | 1000      | 12.50 | 16.67       | 20.83 |
> | physics   | 10000     | 8.33  | 16.67       | 12.50 |
> | history   | 1         | 16.67 | 16.67       | 16.67 |
> | history   | 10        | 16.67 | 16.67       | 16.67 |
> | history   | 100       | 12.50 | 16.67       | 12.50 |
> | history   | 1000      | 4.17  | 8.33        | 0.00  |
> | history   | 10000     | 0.00  | 12.50       | 0.00  |
>
> + Editing: We evaluated the MEND[2] method on the same domains and update counts. The outcomes similarly reproduce the trends reported in our paper.
>
> | Dataset   | Train Size | Root   | Intermediate | Leaf   |
> |-----------|-----------|-------|-------------|-------|
> | biology   | 1         | 35.2  | 32.7        | 42.1  |
> | biology   | 10        | 36.1  | 33.2        | 43.3  |
> | biology   | 100       | 37.6  | 34.4        | 44.7  |
> | biology   | 1000      | 38.9  | 35.1        | 46.2  |
> | biology   | 10000     | 20.3  | 18.7        | 40.5  |
> | economic  | 1         | 45.3  | 42.6        | 50.7  |
> | economic  | 10        | 46.2  | 43.3        | 49.4  |
> | economic  | 100       | 47.7  | 44.6        | 52.9  |
> | economic  | 1000      | 41.3  | 45.7        | 54.1  |
> | economic  | 10000     | 30.2  | 28.3        | 53.2  |
> | physics   | 1         | 25.3  | 22.7        | 30.2  |
> | physics   | 10        | 26.1  | 23.1        | 31.3  |
> | physics   | 100       | 27.4  | 24.6        | 32.6  |
> | physics   | 1000      | 28.7  | 25.4        | 27.7  |
> | physics   | 10000     | 15.2  | 13.4        | 30.3  |
> | history   | 1         | 10.3  | 9.7         | 12.4  |
> | history   | 10        | 11.2  | 10.3        | 13.3  |
> | history   | 100       | 11.1  | 11.4        | 14.1  |
> | history   | 1000      | 12.6  | 12.7        | 12.2  |
> | history   | 10000     | 5.4   | 4.3         | 13.1  |

---

> > ### Author Response · Authors · 2025-11-21
> > **Response to reviewer 7TGp (2/2)**
> >
> > **"Why the unified framework is meaningful?"**
> >
> > While our mathematical formulation shows that editing and unlearning can be cast as constrained optimization problems, the primary contribution is not the formal unification itself. Instead, the unification enables systematic and controlled comparison of editing and unlearning what has been missing in prior work.
> > (a) A single evaluation lens for propagation, stability, and consistency.
> > Prior studies evaluate editing and unlearning using different datasets, metrics, and assumptions, making it impossible to compare their behaviors fairly. Our unified framework standardizes:
> >
> > - the update targets (**Q+**)
> > - the preservation targets (**Q−**)
> > - the optimization structure
> > - the multi-level KG-based probes
> >
> > This allows us to analyze how different update methods behave under the *same* conditions, which previous isolated evaluations could not achieve.
> >
> > (b) New empirical phenomena revealed by the unified setup
> > The unified framework and KG-based benchmark uncover findings that were previously invisible, such as:
> >
> > - **Propagation asymmetry**: editing over-spreads; unlearning under-spreads
> > - **Branch-dependent ceilings**: root-level updates are fundamentally constrained
> > - **Consistency–capacity trade-off**: consistency collapses beyond a certain data scale
> > - **Subject-dependent difficulty**: e.g., history is markedly more resistant to updates
> >
> > These insights emerge only when editing and unlearning are evaluated under a shared, structured setup.
> >
> >
> > ---
> >
> > **"Why editing $\neq$ unlearning even if superficially similar?"**
> >
> > We acknowledge the intuition that forgetting can be viewed as “editing with a neutral target.” Our formulation indeed captures this conceptual link. However, **the behaviors of editing and unlearning diverge significantly in practice**, and our unified evaluation quantifies these differences.
> >
> > (a) Different target distributions drive different optimization behaviors
> > - **Editing** enforces a specific factual correction → strong local alignment.
> > - **Unlearning** pushes toward a neutralized distribution → suppression without replacement.
> >
> > Even if both fit the same formal template, the parameter updates and propagation effects differ markedly.
> >
> > (b) Different failure modes and operational behaviors
> > Editing alters causal pathways; unlearning suppresses them while preserving unrelated knowledge. Their failure modes (conflict rate, CCR, RR, drift) differ systematically, confirming they are not interchangeable.
> >
> > ---
> >
> > **"What value the unified view adds compared with 'editing as a strong baseline for unlearning.'."**
> >
> > From our experiments, we find that editing can serve as a baseline for some unlearning tasks, but not all. Although unlearning can resemble ‘editing with an empty target,’ there remain several important behavioral differences that this perspective does not encompass:
> >
> > - **Propagation asymmetry:** Editing tends to **over-spread**, unintentionally altering related nodes (high CCR), whereas unlearning **under-spreads**, leaving residual knowledge (high RR).
> > - **Consistency–capacity trade-off:** Editing achieves strong local updates but can degrade **out-of-domain consistency** and trigger earlier **consistency collapse**; unlearning preserves broader consistency and is often more efficient computationally.
> > - **Hierarchy and subject dependence:** Update effectiveness varies by branch and subject; leaf-level edits may succeed with editing, but higher-level unlearning requires careful propagation to maintain logical coherence.
> >
> > In conclusion, while editing can mimic some aspects of unlearning, treating unlearning as editing with an empty target does not capture these systematic differences. Our framework makes these differences explicit, enabling a more precise evaluation of interventions across models, tasks, and hierarchical structures.
> >
> > Reference:
> > + [1] Tofu: A task of fictitious unlearning for llms
> > + [2] Fast model editing at scale

---

> > > ### Comment · Reviewer_7TGp · 2025-11-21
> > >
> > > Thank you for your reply. This has well resolved my doubts

---

> > > > ### Author Response · Authors · 2025-11-21
> > > > **Response to reviewer 7TGp**
> > > >
> > > > Thank you for your feedback! We are glad that we resolved your concerns.

---

### Author Response · Authors · 2025-12-01
**Summary of rebuttal**

Dear Area Chair,

Thank you sincerely for your time and effort, especially **under the unusual circumstances of the current ICLR cycle**. We greatly appreciate your work overseeing our submission and managing the substantial additional load during this period.

For your convenience, we provide below a concise summary of each reviewer's post-rebuttal stance:

---

**Reviewer Stance (Post-Rebuttal)**

+ Reviewer 68hr:
After we clarification, **all concerns raised by Reviewer 68hr were fully resolved**, and the evaluation was raised from **4 → 8 on November 20th**, one week before the incident occurred. The final score is unfortunately not reflected in the system due to the change of ICLR rebuttal policy, but you may find the expression in the thread below, which state
>Thank you to the author for taking my suggestion and providing a response. I have raised my score to 8 and wish you all the best!

+ Reviewer hQBy:
Reviewer hQBy provided a positive recommendation **(score:8)** from the beginning, and participate in the discussion clarified that all questions has been resolved. You can find the expression in the thread below, which state:
>Thank you for your reply. This has well resolved my doubts
+ Reviewer 7TGp:
Reviewer 7TGp provided a positive recommendation **(score:8)** from the beginning, and participate in the discussion clarified that all questions has been resolved. You can find the expression in the thread below, which state:
>Thanks for your reply. This has well addressed my concerns.

In summary, all reviewers **unanimously supported acceptance** after rebuttal. Through additional experiments, analyses, and clarifications we added during rebuttal, **every substantive concern was confirmed by the reviewers to be fully resolved**. These improvements further strengthen the rigor, clarity, and practical relevance of the work. We ensure that all such enhancements are incorporated into the final manuscript.

Thank you again for your time and for guiding our submission through this unusual review cycle. We hope the above summary helps streamline your decision process.

Warm regards,

The Authors

---

### Meta-Review · Area_Chair_fBpS · 2026-01-02

**Summary:**

The reviewers' concerns were largely resolved during the rebuttal period. They originally centered on some concerns around the novelty and impact of this work's main contribution: Related knowledge editing to machine unlearning in one framework. They ultimately agreed that this isn't that novel. Nonetheless, the insights were interesting, despite centering on only a couple of methods. The fact that they note their findings derive from many models does help address this issue. The reviewers also found their issues around experimental rigor (e.g., over-reliance on proprietary models for data generation) resolved through new evaluations and explanations. I highly suggest the authors include these new results in the paper prior to its publication.

**Reviewer Concerns:**

See above.

**Reviewer Scores:**

Reviewers 1 and 2 would likely keep their positive scores, Reviewer 3 indicates they would increase.

---

### Decision · Program_Chairs · 2026-01-26

Accept (Poster)